# Branches: Efficiently Seeking Optimal Sparse Decision Trees via AO*

**Ayman Chaouki** [1]   **Jesse Read** [1]   **Albert Bifet** [2][3]

## Abstract

Decision Tree (DT) Learning is a fundamental problem in Interpretable Machine Learning, yet it poses a formidable optimisation challenge. Practical algorithms have recently emerged, primarily leveraging Dynamic Programming and Branch & Bound. However, most of these approaches rely on a Depth-First-Search strategy, which is inefficient when searching for DTs at high depths and requires the definition of a maximum depth hyperparameter. Best-First-Search was also employed by other methods to circumvent these issues. The downside of this strategy is its higher memory consumption, as such, it has to be designed in a fully efficient manner that takes full advantage of the problem's structure. We formulate the problem within an AND/OR graph search framework and we solve it with a novel AO*-type algorithm called BRANCHES. We prove both optimality and complexity guarantees for BRANCHES and we show that it is more efficient than the state of the art theoretically and on a variety of experiments. Furthermore, BRANCHES supports nonbinary features unlike the other methods, we show that this property can further induce larger gains in computational efficiency.

## 1. Introduction

Black-box models are ill-suited for contexts where decisions carry substantial ramifications. In healthcare for example, misdiagnoses can delay crucial treatments and lead to severe outcomes. Likewise, in the criminal justice system, black-box models can obscure biases and result in discriminatory rulings. Such risks highlight the necessity of adopting interpretable models in sensitive domains.

Decision Trees (DTs) are highly interpretable due to their

---

[1]LIX, Ecole Polytechnique, IP Paris [2]AI Institute, University of Waikato [3]LTCI, Télécom Paris, IP Paris. Correspondence to: Ayman Chaouki <chaoukiayman2@gmail.com>.

*Proceedings of the 42$^{nd}$ International Conference on Machine Learning*, Vancouver, Canada. PMLR 267, 2025. Copyright 2025 by the author(s).

simple decision rules (splits). However, this interpretability weakens as the number of splits increases, which makes the joint optimisation of accuracy and sparsity (minimising the number of splits) a fundamental problem in Interpretable Machine Learning. We refer to this problem as the *sparsity* problem. This optimisation task is particularly difficult due to its NP-completeness (Hyafil & Rivest, 1976). Consequently, greedy approaches, such as C4.5 (Quinlan, 2014) and CART (Breiman et al., 1984), have been historically favoured. While these methods are fast and scalable, their greedy nature often yields suboptimal and overly complex DTs.

This issue spurred a large research effort into investigating alternatives, mainly focusing on *Mathematical Programming* (Bennett, 1992; 1994; Bennett & Blue, 1996; Bessiere et al., 2009; Norouzi et al., 2015; Bertsimas & Dunn, 2017; Narodytska et al., 2018; Verwer & Zhang, 2019; Hu et al., 2020; Blanquero et al., 2021; Günlük et al., 2021). However, the number of variables in these Mathematical Programs depends strongly on the dataset size and thus induces poor scalability. Moreover, these methods often fix the DT structure and only optimise the internal splits and leaf predictions, overlooking the sparsity portion of the problem.

Recently, Dynamic Programming (DP) (Bellman & Dreyfus, 2015) and Branch & Bound (B&B) (Lawler & Wood, 1966) led to breakthroughs in runtimes, with most approaches employing a Depth-First-Search (DFS) strategy (Nijssen & Fromont, 2007; 2010; Aglin et al., 2020; Demirović et al., 2022; van der Linden et al., 2023). While DFS is appealing from a storage economy perspective, its uninformed nature makes it inefficient for large problems (Pearl, 1984, p.36). On the other hand, informed strategies were also used through Best-First-Search (BFS) (Hu et al., 2019; Lin et al., 2020), albeit in a sub-efficient manner that does not take full advantage of the problem's AND/OR structure.

Montanari et al. (1975) showed that DP problems can be formulated within an AND/OR graph search framework, in which they can be solved efficiently with powerful heuristic search algorithms (Pearl, 1984). We follow this approach because the sparsity problem can be framed within DP, then upon defining an adequate heuristic, which we call the *Purification Bound*, we solve the AND/OR graph search problem with the celebrated AO* approach (Nilsson, 2014;

Martelli & Montanari, 1978). The induced algorithm, called BRANCHES, is guaranteed to return an optimal solution when it terminates. In addition, BRANCHES also satisfies complexity guarantees in the form of an upper bound on the number of evaluated branches before termination. To the best of our knowledge, such analysis was only previously conducted in (Hu et al., 2019, Theorem E.2). We show numerically that BRANCHES' complexity bound is significantly smaller than the bound in (Hu et al., 2019, Theorem E.2). Empirically, BRANCHES always finds an optimal solution in substantially fewer iterations than the state of the art, and despite its current Python implementation, it also displays better runtimes than its C++ competitors. Furthermore, upon reaching timeout, some methods can still propose a solution, albeit with no quality guarantees. This property is called the *anytime behaviour* and it is satisfied by BRANCHES. On this front, our experiments show that BRANCHES proposes better solutions than the state of the art.

## 2. Related Work

In this section, we mainly survey the proposed **Depth First Search (DFS)** and **Best First Search (BFS)** algorithms.

**DFS:** Nijssen & Fromont (2007; 2010) formulate a search space, called the lattice of itemsets, from which DTs can be mined. This powerful idea induced the DL8 algorithm and is at the basis of many subsequent works. DL8 explores the lattice in a DFS fashion seeking an optimal DT that satisfies a certain set of constraints. Moreover, DL8 exhibits enough flexibility to solve the sparsity problem, as evident by the lexicographical objective $\text{Argmin}_T\{\text{error}(T), \text{size}(T)\}$ considered in (Nijssen & Fromont, 2010, Section 2.3). However, the immense size of the lattice, which grows exponentially with the number of features, renders DL8 impractical on many real-world applications. A decade later, Aglin et al. (2020) improved DL8 with a B&B component that prunes the lattice based on the current best found solution. The induced DL8.5 algorithm is faster than DL8 on a broader range of applications. However, it only considers constraints on the maximum depth and minimum number of data per leaf, prohibiting it from solving the sparsity problem. In a subsequent work, Demirović et al. (2022) improved the computational complexity of DL8.5 through the use of a specialised technique for handling DTs of depth 2, which resulted in the MurTree algorithm. Recently, MURTREE was generalised by van der Linden et al. (2023) to handling a wider range of objectives within the STREED framework.

**BFS:** Hu et al. (2019) introduce OSDT, which considers the objective $\text{Argmin}_T\{\text{error}(T) + \lambda\text{leaves}(T)\}$ where $0 < \lambda < 1$ is a soft penalty on the number of leaves. A similar objective has been considered in prior works, e.g. (Bert-

simas & Dunn, 2017). Unlike the DFS methods, OSDT employs analytical bounds and a priority queue to prioritise regions with better bounds, resulting in a more aggressive pruning of the search space. On the other hand, OSDT operates on the space of DTs instead of the lattice of itemsets, which greatly slows it down. To alleviate this issue, Lin et al. (2020) developed GOSDT, a BFS algorithm operating on the lattice of itemsets. GOSDT can be considered the state of the art for the sparsity problem, its DP is as efficient as the DFS methods while its B&B prunes the lattice more efficiently. Furthermore, GOSDT generalises OSDT to other objectives including weighted accuracy, balanced accuracy, F-score, AUC and partial area under the ROC convex hull.

**DFS vs BFS:** (Pearl, 1984, Chapter 2) provides an excellent discussion on the differences between DFS and BFS, with a brief summary in (Pearl, 1984, Section 2.5). The main advantage of DFS is its storage economy. However, it necessitates a maximum depth parameter to avoid long searches in one region of the search space. Moreover, the uninformed nature of DFS makes it inefficient for large problems. On the other hand, the informed nature of BFS allows it to find solutions more quickly than DFS without the need for a maximum depth parameter. Nevertheless, this judicious paradigm comes at the cost of high memory consumption, hence the necessity to devise BFS algorithms that find an optimal solution as quickly as possible.

To achieve this, we frame the DP problem of sparsity as an AND/OR graph search (Martelli & Montanari, 1978). (Nilsson, 2014, Section 3.1) and (Pearl, 1984, Section 1.2.4) provide detailed overviews on AND/OR graphs. In this context, we can apply the popular and efficient AO* algorithm, which was introduced in (Martelli & Montanari, 1978) and (Nilsson, 2014, Section 3.2). We note that Martelli & Montanari (1978) employed AO* (with the early name HS) to seeking optimal DTs, but in a cost-sensitivity context (Lomax & Vadera, 2013) that is distinct from the sparsity problem. Another difference is that the authors sought DTs that perfectly classify a dataset while we seek DTs on the pareto front jointly maximising accuracy and minimising the number of splits. Verhaeghe et al. (2020) also employ an AND/OR formulation, but within a Constraint-Programming (CP) paradigm. The induced CP algorithm does not solve the sparsity problem but rather a similar problem to DL8.5 where the maximum DT depth is constrained. In an empirical comparison, Aglin et al. (2020) thoroughly showed that DL8.5 outperforms CP.

## 3. Problem Formulation

We consider classification problems with categorical features $X = (X^{(1)}, \dots, X^{(q)})$ and class $Y \in \{1, \dots, K\}$:

$$\forall i \in \{1, \dots q\} : X^{(i)} \in \{1, \dots, C_i\}, \ C_i \geq 2$$

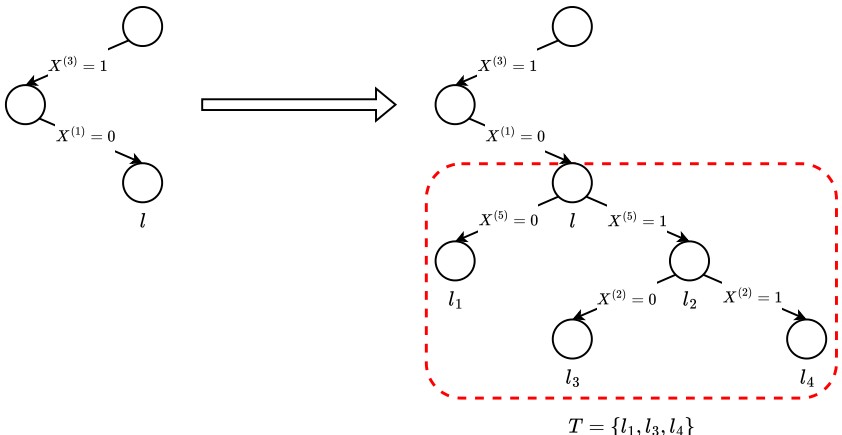

$$T = \{l_1, l_3, l_4\}$$

**Figure 1.** Consider a feature space with five binary features $X^{(1)}, X^{(2)}, X^{(3)}, X^{(4)}, X^{(5)} \in \{0, 1\}$. The figure provides an example of a sub-DT $T = \{l_1, l_3, l_4\}$ rooted in $l$ that stems from splitting branch $l$ with respect to feature $X^{(5)}$ and splitting branch $l_2$ with respect to feature $X^{(2)}$, the red perimeter emphasises the fact that $T$ is rooted in $l$. Here $\mathcal{S}(T) = 2, l = \mathbb{I}\{X^{(3)} = 1\} \wedge \mathbb{I}\{X^{(1)} = 0\}, l_1 = l \wedge \mathbb{I}\{X^{(5)} = 0\}, l_2 = l \wedge \mathbb{I}\{X^{(5)} = 1\}, l_3 = l_2 \wedge \mathbb{I}\{X^{(2)} = 0\}, l_4 = l_2 \wedge \mathbb{I}\{X^{(2)} = 1\}$.

where $q \geq 2$ and $K \geq 2$. We are provided with a dataset $\mathcal{D} = \{(X_m, Y_m)\}_{m=1}^n$ of $n \geq 1$ examples. In the following sections, we define the notions of branches and sub-DTs that are key to our formulation.

### 3.1. Branches

A branch $l$ is a conjunction of clauses on the features of the following form (where $\mathbb{I}$ is the indicator function):

$$l = \bigwedge_{v=1}^{\mathcal{S}(l)} \mathbb{I}\{X^{(i_v)} = j_v\}$$

such that $\forall 1 \leq v \leq \mathcal{S}(l) : 1 \leq i_v \leq q, \ 1 \leq j_v \leq C_{i_v}$ and:

$$\forall v, v' \in \{1, \dots \mathcal{S}(l)\} : v \neq v' \implies i_v \neq i_{v'} \quad (1)$$

This condition ensures that no feature is used in more than one clause within $l$. We refer to these clauses as *rules* or *splits*. $\mathcal{S}(l)$ is the number of splits in $l$.

For any datum $X = (X^{(1)}, \dots, X^{(q)})$, the valuation of $l$ for $X$ is denoted $l(X) \in \{0, 1\}$ and defined as follows:

$$l(X) = 1 \iff \bigwedge_{v=1}^{\mathcal{S}(l)} \mathbb{I}\{X^{(i_v)} = j_v\} = 1$$

When $l(X) = 1$, we say that $X$ is in $l$ or that $l$ contains $X$. The branch containing all possible data is called the root and is denoted $\Omega$. Since the valuation of $l$ for any datum remains invariant when reordering the splits, we represent $l$ uniquely by ordering its splits from the smallest feature index to the highest, i.e. we impose $1 \leq i_1 < \dots < i_{\mathcal{S}(l)} \leq q$. This unique representation is at the core of our memoisation.

In the following, we define the notion of splitting a branch. Let $i \in \{1, \dots, q\} \setminus \{i_1, \dots, i_{\mathcal{S}(l)}\}$ be an unused feature in the splits of $l$. We define the children of $l$ that stem from splitting $l$ with respect to $i$ as the set $\mathrm{Ch}(l, i) = \{l_1, \dots, l_{C_i}\}$ where:

$$\forall j \in \{1, \dots, C_i\} : l_j = l \wedge \mathbb{I}\{X^{(i)} = j\} \quad (2)$$

The dataset $\mathcal{D} = \{(X_m, Y_m)\}_{m=1}^n$ provides an empirical distribution of the data. The probability that a datum is in $l$ is as follows:

$$\mathbb{P}[l(X) = 1] = \frac{n(l)}{n}$$

where $n(l) = \sum_{m=1}^n l(X_m)$ is the number of data in $l$ and $n$ is the total number of data. Likewise, we want to define the probability that a datum is in $l$ and correctly classified. For this purpose, we define the predicted class in $l$ as:

$$k^*(l) = \mathrm{Argmax}_{1 \leq k \leq K} \{n_k(l)\}$$

where $n_k(l) = \sum_{m=1}^n l(X_m) \mathbb{I}\{Y_m = k\}$ is the number of data in $l$ that are of class $k$. In other words, $k^*(l)$ is the majority class in $l$. Then the probability that a datum is in $l$ and correctly classified is:

$$\mathcal{H}(l) = \mathbb{P}[l(X) = 1, k^*(l) = Y] = \frac{n_{k^*(l)}(l)}{n} \quad (3)$$

### 3.2. Sub-DTs

Let $l$ be a branch, a sub-DT rooted in $l$ is a collection of branches $T = \{l_1, \dots, l_{|T|}\}$ that stems from successive splits starting from from $l$, we denote $\mathcal{S}(T)$ the number of these splits. Intuitively, $\mathcal{S}(T)$ can be see as the number of *internal nodes* in $T$, and $l_1, \dots, l_{|T|}$ as the *leaves* of $T$.

Figure 1 provides an example of a sub-DT. $T$ partitions $l$ in the following sense:

$$\begin{cases} l = \bigvee_{u=1}^{|T|} l_u \\ \forall u, u' \in \{1, \ldots, |T|\} : u \neq u' \implies l_u \wedge l_{u'} = 0 \end{cases}$$

For any datum $X$ in $l$, $T$ predicts the majority class of the branch $l_u \in T$ containing $X$:

$$T(X) = \sum_{u=1}^{|T|} l_u(X) k^*(l_u) \in \{1, \ldots, K\} \qquad (4)$$

Now we can define the probability that a datum is in $l$ and correctly classified by $T$:

$$\mathcal{H}(T) = \mathbb{P}[l(X) = 1, T(X) = Y]$$
$$= \sum_{u=1}^{|T|} \mathbb{P}[l_u(X) = 1, k^*(l_u) = Y] = \sum_{u=1}^{|T|} \mathcal{H}(l_u)$$

The additivity property is due to $\{l_1, \ldots, l_{|T|}\}$ forming a partition of $l$, then the result stems from the definitions (4) and (3).

We define a DT as a sub-DT that is rooted in the root $\Omega$. Let $T$ be a DT, since $\Omega(X) = 1$ for any datum $X$ then:

$$\mathcal{H}(T) = \mathbb{P}[\Omega(X) = 1, T(X) = Y] = \mathbb{P}[T(X) = Y]$$

which is the accuracy of $T$. To solve the sparsity problem, we seek the DT $T^*$ maximising the following objective:

$$\mathcal{H}_\lambda(T) = -\lambda \mathcal{S}(T) + \mathcal{H}(T) \qquad (5)$$

where $0 < \lambda < 1$ is a penalty parameter penalising DTs with a large number of splits. This objective is employed by CART during the pruning phase, it was also considered by Bertsimas & Dunn (2017) and recently by Chaouki et al. (2024). Hu et al. (2019) and Lin et al. (2020) use a slightly different version, where the total number of leaves is penalised instead. In a setting with binary features, the number of leaves of a DT is always equal to the number of splits plus 1 and the two objectives are equivalent in this case.

### 3.3. AND/OR Graph Representation

To formulate our AND/OR graph, we define the following state space model $(\mathcal{S}, \mathcal{T}, \mathcal{A}, F, r)$ where $\mathcal{S}$ is the set of states, $\mathcal{T} \subset \mathcal{S}$ the set of terminal (or goal) states, $\mathcal{A}$ the set of actions with $\mathcal{A}(l)$ the set of permissible actions at a non-terminal state $l \in \mathcal{S} \setminus \mathcal{T}$, $F : \mathcal{S} \times \mathcal{A} \mapsto \mathcal{P}(\mathcal{S})$ the transition function (with $\mathcal{P}(\mathcal{S})$ the power set of $\mathcal{S}$) and $r : \mathcal{S} \times \mathcal{A} \mapsto \mathbb{R}$ the reward function.

**States $\mathcal{S}$:** Our state space is the set of all branches along with the set of all terminal states $\mathcal{T}$, which we define below. The root $\Omega$ is always our initial state.

**Terminal states:** For every branch $l$ we assign a unique terminal state $\bar{l} \in \mathcal{T}$, which signals that we should not consider any further splits (end of search).

**Actions $\mathcal{A}$ and transitions $F$:** Consider a non-terminal state (a branch) $l = \bigwedge_{v=1}^{\mathcal{S}(l)} \mathbb{I}\{X^{(i_v)} = j_v\} \in \mathcal{S} \setminus \mathcal{T}$, there are two types of actions in $\mathcal{A}(l)$:

- The terminal action $\bar{a}$, which transitions $l$ into its corresponding terminal state $\bar{l} \in \mathcal{T}$. Thus $F(l, \bar{a}) = \{\bar{l}\}$.

- Split actions, which is the set of all unused features $\{1, \ldots, q\} \setminus \{i_1, \ldots, i_{\mathcal{S}(l)}\}$ by $l$. Let $i$ be a split action, taking $i$ transitions $l$ to the set of children $\text{Ch}(l, i)$, defined in (2). Therefore $F(l, i) = \text{Ch}(l, i)$.

Thus $\mathcal{A}(l) = \{\bar{a}\} \cup \{1, \ldots, q\} \setminus \{i_1, \ldots, i_{\mathcal{S}(l)}\}$. When $\mathcal{S}(l) = q$, then $\mathcal{A}(l) = \{\bar{a}\}$ and we can only transition to $\bar{l}$.

**Reward function $r$:** Let $l \in \mathcal{S} \setminus \mathcal{T}$ be a non-terminal state and $a \in \mathcal{A}(l)$, we define the reward $r(l, a)$ as follows:

- If $a$ is a split action, then $r(l, a) = -\lambda$ regardless of $l$. $\lambda$ is the penalty parameter defined in Equation (5).

- If $a = \bar{a}$, then $r(l, \bar{a}) = \mathcal{H}(l)$ as per Equation (3).

We represent our state space model as an AND/OR graph following the hypergraph convention in (Nilsson, 2014, Section 3.1). The nodes of the hypergraph represent states and its connectors represent actions. A connector is a generalisation of the notion of edge, it connects a parent node to a set of successor nodes. In our formulation, a node (state) $l$ has $|\mathcal{A}(l)|$ outgoing connectors, each corresponding to an action. The connector of action $a \in \mathcal{A}(l)$ links node $l$ to the set of successor nodes $F(l, a)$ and has a weight $r(l, a)$. See Figure 2 for an example. In the following, we introduce the notion of policy and we transform the problem into seeking an optimal policy.

A policy $\pi$ maps each non-terminal state $l \in \mathcal{S} \setminus \mathcal{T}$ to an action $\pi(l) \in \mathcal{A}(l)$. We define the trajectory of $\pi$ from $l$ as the sequence $\left(T_{l,t}^\pi\right)_{t \geq 0}$ where $T_{l,0}^\pi = \{l\}$ and for all $t \geq 0$:

$$T_{l,t+1}^\pi = \bigcup_{l_u \in T_{l,t}^\pi \setminus \mathcal{T}} F(l_u, \pi(l_u)) \bigcup_{\overline{l_u} \in T_{l,t}^\pi \cap \mathcal{T}} \{\overline{l_u}\}, t \geq 0$$

The term $\bigcup_{\overline{l_u} \in T_{l,t}^\pi \cap \mathcal{T}} \{\overline{l_u}\}$ is the set of terminal states in $T_{l,t}^\pi$ while the term $\bigcup_{l_u \in T_{l,t}^\pi \setminus \mathcal{T}} F(l_u, \pi(l_u))$ is the set of induced states from taking the actions dictated by policy $\pi$ in each non-terminal state in $T_{l,t}^\pi$. As such, the trajectory $\left(T_{l,t}^\pi\right)_{t \geq 0}$ stems from following $\pi$ from $l$ by applying it, each time, at the non-terminal states and retaining the terminal states.

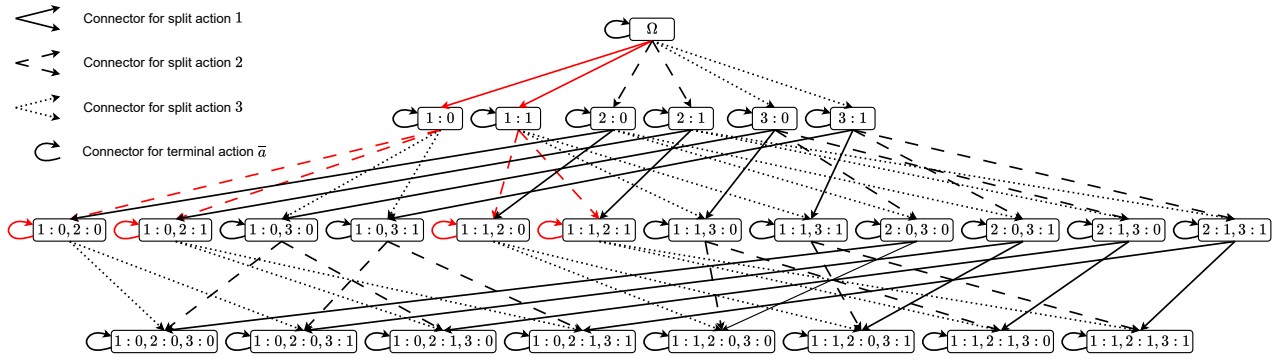

*Figure 2.* AND/OR graph for a classification problem with three binary features $X^{(1)}, X^{(2)}, X^{(3)}$. To make the notation lighter, we represent any branch $l = \bigwedge_{v=1}^{\mathcal{S}(l)} \mathbb{I}\{X^{(i_v)} = j_v\}$ with $i_1 : j_1, \ldots, i_{\mathcal{S}(l)} : j_{\mathcal{S}(l)}$, for example $1 : 0, 2 : 1$ represents the branch $\mathbb{I}\{X^{(1)} = 0\} \wedge \mathbb{I}\{X^{(2)} = 1\}$. We colour in red the actions taken by the policy: $\pi(\Omega) = 1, \pi(1 : 0) = \pi(1 : 1) = 2, \pi(1 : 0, 2 : 0) = \pi(1 : 0, 2 : 1) = \pi(1 : 1, 2 : 0) = \pi(1 : 1, 2 : 1) = \bar{a}$, which also depicts the DT $T^\pi$ of $\pi$. Note that, although the curved connector associated with the terminal action $\bar{a}$ connects to the same node, it transitions to a terminal state from which no action can be taken. We represent it like this to avoid overloading the figure with additional nodes corresponding to the terminal states.

**Proposition 3.1.** *Let $\pi$ be a policy and $l \in \mathcal{S} \setminus \mathcal{T}$, then there exists a minimum $\tau_l^\pi \geq 1$ such that for any $t \geq \tau_l^\pi$, $T_{l,t}^\pi = \{\overline{l_1}, \ldots, \overline{l_{|T_{\tau_l^\pi}|}}\}$ is composed of terminal states only.*

Proposition 3.1 shows that for any branch $l \in \mathcal{S} \setminus \mathcal{T}$, any policy $\pi$ arrives at a final set of states $\{\overline{l_1}, \ldots, \overline{l_{|T_{\tau_l^\pi}|}}\}$, in which case we define $T_l^\pi = \{l_1, \ldots, l_{|T_{\tau_l^\pi}|}\}$ as the **sub-DT of $\pi$ rooted in $l$**. $T_l^\pi$ is indeed a sub-DT rooted in $l$ by the definition in Section 3.2 because it stems from successive splits from $l$. For the root $\Omega$, we simplify the notation $T_\Omega^\pi \equiv T^\pi$ and call $T^\pi$ the **DT of $\pi$**. Figure 2 provides an example of the DT of a policy. This definition allows us to evaluate policies as follows: we define the value of a policy $\pi$ from a non-terminal state $l \in \mathcal{S} \setminus \mathcal{T}$ with:

$$\mathcal{V}^\pi(l) = \sum_{t=0}^{\tau_l^\pi - 1} \sum_{l_u \in T_{l,t}^\pi \setminus \mathcal{T}} r(l_u, \pi(l_u)) \qquad (6)$$

$\mathcal{V}^\pi(l)$ is the cumulative reward incurred by following policy $\pi$ from $l$ until we end up in the sub-DT $T_l^\pi$. In the example of Figure 2, $\mathcal{V}^\pi(\Omega)$ is the sum over all the weights of the red connectors. Policies are evaluated and compared with respect to their value from $\Omega$, an optimal policy being $\pi^* \in \mathrm{Argmax}_\pi \mathcal{V}^\pi(\Omega)$. In the following, we justify why seeking $\pi^*$ is equivalent to seeking $T^* = \mathrm{Argmax}_T \mathcal{H}_\lambda(T)$.

**Proposition 3.2.** *Let $\pi$ be a policy and $l \in \mathcal{S} \setminus \mathcal{T}$ a non-terminal state, then $\mathcal{V}^\pi(l)$ satisfies the following:*

$$\mathcal{V}^\pi(l) = \mathcal{H}_\lambda(T_l^\pi) = -\lambda \mathcal{S}(T_l^\pi) + \mathcal{H}(T_l^\pi)$$

*Moreover, the optimal DT $T^*$ is the DT of an optimal policy $\pi^*$, in other words $T^* = T^{\pi^*}$.*

Proposition 3.2 implies that we can seek $\pi^*$ and then deduce $T^*$ by following $\pi^*$ from $\Omega$. In the terminology of (Nilsson, 2014, Section 3.1), $T^*$ is an optimal solution graph. Our task can be transcribed as seeking a policy that induces an optimal solution graph. This can be achieved with AO*.

## 4. The Algorithm: BRANCHES

To describe BRANCHES, we first derive the Bellman optimality equation satisfied by the problem in Proposition 4.1. To do this conveniently, we introduce the state-action values. For any policy $\pi$, branch $l \in \mathcal{S} \setminus \mathcal{T}$ and action $a \in \mathcal{A}(l)$, the state-action value $\mathcal{Q}^\pi(l, a)$ is the cumulative reward of taking action $a$ first and then following $\pi$:

$$\mathcal{Q}^\pi(l, a) = r(l, a) + \sum_{l_u \in F(l,a) \setminus \mathcal{T}} \mathcal{V}^\pi(l_u) \qquad (7)$$

We abbreviate the notation for all quantities that are related to $\pi^*$ with $T_{l,t}^* \equiv T_{l,t}^{\pi^*}, \tau_l^* \equiv \tau_l^{\pi^*}, \mathcal{V}^* \equiv \mathcal{V}^{\pi^*}, \mathcal{Q}^* \equiv \mathcal{Q}^{\pi^*}$.

**Proposition 4.1** (Bellman Optimality Equations). *Let $\pi^*$ be an optimal policy, i.e. $\pi^* \in \mathrm{Argmax}_\pi \mathcal{V}^\pi(\Omega)$ and consider the set of non-terminal states in its trajectory from $\Omega$:*

$$\mathcal{S}^* = \bigcup_{t=0}^{\tau_\Omega^* - 1} T_{\Omega,t}^* \setminus \mathcal{T}$$

*Now consider a policy $\pi$ and suppose that for all $l \in \mathcal{S}^*$:*

$$\mathcal{V}^\pi(l) = \max_{a \in \mathcal{A}(l)} \mathcal{Q}^\pi(l, a) \qquad (8)$$

*Then $\pi$ is optimal and we also have:*

$$\pi(l) = \mathrm{Argmax}_{a \in \mathcal{A}(l)} \mathcal{Q}^\pi(l, a) \qquad (9)$$

Note that Proposition 4.1 establishes that any optimal policy $\pi^* \in \text{Argmax}_\pi \mathcal{V}^\pi(\Omega)$ only has to satisfy the Bellman optimality equations (8) and (9) for a subset of few *relevant* states $\mathcal{S}^*$ regardless of the remaining states of $\mathcal{S}$. As such, it suffices to seek $\pi^*$ as a partial policy defined on these states, without the need to define it elsewhere. This implies that an efficient search strategy should focus the search effort on the relevant part of the state space to find $T^*$ quickly, this is exactly the purpose of AO* and its advantage over Dynamic Programming (Hansen & Zilberstein, 2001, p.2, 11, 12).

To adopt an AO* approach, we need to define adequate heuristic estimates in the form of upper bounds $\mathcal{V}(l)$ and $\mathcal{Q}(l, a)$ on $\mathcal{V}^*(l)$ and $\mathcal{Q}^*(l, a)$ respectively.

### 4.1. Heuristic estimates $\mathcal{V}(l)$ and $\mathcal{Q}(l, a)$

Let $l \in \mathcal{S} \setminus \mathcal{T}$ be a non-terminal state. For the terminal action $\overline{a}$, according to definition (7), we have direct access to $\mathcal{Q}^*(l, \overline{a}) = r(l, \overline{a}) = \mathcal{H}(l)$ and thus we can define $\mathcal{Q}(l, \overline{a})$ straightforwardly using Equation (3) as follows:

$$\mathcal{Q}(l, \overline{a}) = \mathcal{Q}^*(l, \overline{a}) = r(l, \overline{a}) = \mathcal{H}(l) = \frac{n_{k^*(l)}(l)}{n} \quad (10)$$

Now consider a split action $a \in \mathcal{A}(l) \setminus \{\overline{a}\}$ and let us define $\mathcal{Q}(l, a)$ and $\mathcal{V}(l)$. The definition (7) and the Bellman equation (8) suggest the following recursive definition:

$$\mathcal{Q}(l, a) = -\lambda + \sum_{l_u \in F(l, a)} \mathcal{V}(l_u) \quad (11)$$

$$\mathcal{V}(l) = \max_{a \in \mathcal{A}(l)} \mathcal{Q}(l, a) \quad (12)$$

We note that $\mathcal{Q}(l, a)$ in Equation (11) can only be calculated if the heuristic estimates $\mathcal{V}(l_u)$ are available. Thus to complete this recursive definition, we need to initialise $\mathcal{V}(l_u)$ adequately.

**Proposition 4.2** (Purification Bound)**.** *Let $l \in \mathcal{S} \setminus \mathcal{T}$, we define the Purification Bound as follows:*

*If $\mathcal{A}(l) \setminus \{\overline{a}\} \neq \emptyset$:*

$$\mathcal{V}(l) = \max\{\mathcal{H}(l), -\lambda + \mathbb{P}[l(X) = 1]\}$$
$$= \max\left\{\frac{n_{k^*(l)}(l)}{n}, -\lambda + \frac{n(l)}{n}\right\} \quad (13)$$

*Otherwise:*

$$\mathcal{V}(l) = \mathcal{V}^*(l) = \mathcal{H}(l) = \frac{n_{k^*(l)}(l)}{n} \quad (14)$$

*The bounds $\mathcal{V}(l)$ are initialised with (13) or (14), then they are recursively backpropagated to the ancestors of $l$ in the AND/OR graph through (11) and (12). The resulting heuristic estimates $\mathcal{Q}(l, a)$ and $\mathcal{V}(l)$ are upper bounds on the true optimal values $\mathcal{Q}^*(l, a)$ and $\mathcal{V}^*(l)$ respectively.*

Equation (14) is straightforward. Indeed, $\mathcal{A}(l) \setminus \{\overline{a}\} = \emptyset$ means that no split action can be taken at $l$ because we have already exhausted them all, which happens when $\mathcal{S}(l) = q$, i.e. when $l$ employs all the features in its splits. In this case, Equation (8) implies that $\mathcal{V}^*(l) = \mathcal{Q}^*(l, \overline{a}) = \mathcal{H}(l)$. On the other hand, when $\mathcal{A}(l) \setminus \{\overline{a}\} \neq \emptyset$, the following provides the intuitive reasoning behind the Purification Bound. We want to initialise an upper bound $\mathcal{V}(l) \geq \mathcal{V}^*(l)$. Proposition 3.2 states that $\mathcal{V}^*(l) = \mathcal{H}_\lambda(T_l^*)$ which is the objective of the best possible sub-DT rooted in $l$. Initially, the only information we have about $l$ is $\mathcal{H}(l)$, which is the objective of the sub-DT $\{l\}$. Furthermore, we know that any other sub-DT $T$ rooted in $l$ employs at least one split and has an accuracy $\mathcal{H}(T)$ at most equal $\mathbb{P}[l(X) = 1]$ as shown below

$$\mathcal{H}(T) = \mathbb{P}[l(X) = 1, T(X) = Y] \leq \mathbb{P}[l(X) = 1]$$

where equality only happens if all the branches in $T$ are *pure*, i.e. each branch contains only one class. As a consequence, all sub-DTs $T$ rooted in $l$, including $T_l^*$, satisfy:

$$\mathcal{H}_\lambda(T) \leq \max\left\{\mathcal{H}(l), -\lambda + \mathbb{P}[l(X) = 1]\right\}$$

**Remark:** We emphasize that we do not initialise $\mathcal{V}(l)$ for all tip nodes (nodes with no successors, also called leaves) of the AND/OR graph and then run the recursive updates (11) and (12) up the AND/OR graph. This would be a purely DP approach and it would indeed find $T^*$. However, such an approach defies the purpose of focused search as it would require very expensive computational resources. We need a search strategy to carefully choose the branches to evaluate in order to find $T^*$ as quickly as possible. The next section is dedicated to describing this search strategy.

**Summary:** For a non-terminal state $l \in \mathcal{S} \setminus \mathcal{T}$, $\mathcal{Q}(l, \overline{a})$ is known in advance and calculated with (10). For any split action $a \in \mathcal{A}(l) \setminus \{\overline{a}\}$, $\mathcal{Q}(l, a)$ is calculated with (11). $\mathcal{V}(l)$ are first initialised with (13) (for branches $l$ that are specifically chosen by the search strategy) and later updated with (12).

### 4.2. The Search strategy

We initialise a *memo* and a search graph $G$ that consist solely of the root $\Omega$. Throughout the algorithm's execution, we label a node $l$ (in $G$) as SOLVED when we know its optimal action $\pi^*(l) = \text{Argmax}_{a \in \mathcal{A}(l)} \mathcal{Q}^*(l, a)$, in which case $\mathcal{V}^*(l) = \mathcal{V}(l)$ and $\mathcal{Q}^*(l, \pi^*(l)) = \mathcal{Q}(l, \pi^*(l))$. Until $\Omega$ is SOLVED perform the following steps at each iteration:

**Selection:** Starting from the root $l = \Omega$, until $l$ is a leaf of $G$, descend $G$ by following the selection policy:

$$\widetilde{\pi}(l) = \text{Argmax}_{a \in \mathcal{A}(l)} \mathcal{Q}(l, a) \quad (15)$$

Store the connector (action) $\widetilde{\pi}(l)$ in a list *path*. If all the states in $F(l, \widetilde{\pi}(l))$ are SOLVED, choose one of them arbi-

trarily, stop the Selection step and move to the Backpropagation step. Otherwise, choose an UNSOLVED state from $F(l, \widetilde{\pi}(l))$ and make it the current state $l$, Appendix B provides the details of how we conduct this choice. Repeat the process until reaching a leaf $l$ of $G$.

**Expansion:** The purpose of this step is to grow $G$ with the successor nodes of $l$ and update $\mathcal{V}(l)$ and $\mathcal{Q}(l, a)$ for all actions $a \in \mathcal{A}(l)$. For the terminal action, calculate $\mathcal{Q}(l, \overline{a}) = \mathcal{H}(l)$ as per (10). Generate all the successor nodes of $l$ and add them to $G$ as follows: for all split actions $a \in \mathcal{A}(l) \setminus \{\overline{a}\}$ and all successors $l_u \in F(l, a)$, if $l_u \in memo$ add a link between $l$ and $l_u$ as part of the connector $a$, otherwise create a node $l_u$ in $G$ and add it as a successor of $l$ stemming from the connector $a$, store $l_u$ in $memo$, and initialise $\mathcal{V}(l_u)$ with the Purification Bound in Proposition 4.2. If $\mathcal{V}(l_u) = \frac{n_{k^*(l_u)}(l_u)}{n}$ then label $l_u$ as SOLVED because we would know that $\overline{a}$ is optimal at $l_u$ as shown below:

$$\mathcal{Q}^*(l_u, \overline{a}) = \mathcal{Q}(l_u, \overline{a}) = \mathcal{V}(l_u) = \max_{a \in \mathcal{A}(l_u)} \mathcal{Q}(l_u, a)$$
$$\geq \max_{a \in \mathcal{A}(l_u)} \mathcal{Q}^*(l_u, a)$$

For all split actions $a \in \mathcal{A}(l) \setminus \{\overline{a}\}$ calculate $\mathcal{Q}(l, a)$ with (11), then deduce $\mathcal{V}(l)$ with (12). Update the selection policy at $l$ with Equation (15). If all the successors in $F(l, \widetilde{\pi}(l))$ are SOLVED then label $l$ as SOLVED.

**Backpropagation:** Update the heuristic estimates upwards in $G$ through the list $path$ of selected connectors, which we stored back in the Selection step. For $j = length(path) - 1, \ldots, 0$: $\widetilde{\pi}(l) = path[j]$, update $\mathcal{Q}(l, \widetilde{\pi}(l)), \mathcal{V}(l)$ and $\widetilde{\pi}(l)$ with (11), (12) and (15) respectively. If $\widetilde{\pi}(l) = \overline{a}$ or all the successors in $F(l, \widetilde{\pi}(l))$ are SOLVED then label $l$ as SOLVED.

We provide implementation details and a pseudocode in Appendix D and Algorithm 1. Theorem 4.3 proves the optimality of BRANCHES.

**Theorem 4.3** (Optimality)**.** *Upon termination, the selection policy $\widetilde{\pi}$ becomes optimal. In other words:*

$$\mathcal{V}^{\widetilde{\pi}}(\Omega) = \mathcal{V}^*(\Omega) = \max_{\pi} \mathcal{V}^{\pi}(\Omega)$$

To accurately assess the search efficiency of BRANCHES, we analyse in Theorem 4.4 the number of branch evaluations, i.e. calculations of $\mathcal{H}(l)$, it performs before terminating.

**Theorem 4.4** (Complexity)**.** *Let $\Gamma(q, C, \lambda)$ denote the total number of branch evaluations performed by BRANCHES for a classification problem with a number of features $q \geq 2$, a penalty parameter $0 < \lambda < 1$, and a number of categories per feature $C \geq 2$. Then, $\Gamma(q, C, \lambda)$ satisfies:*

$$\Gamma(q, C, \lambda) \leq \sum_{h=0}^{\kappa} (q - h) C^{h+1} \binom{q}{h}$$

*Table 1.* Comparing orders of magnitude of the complexity bounds of BRANCHES and OSDT for binary features and for different values of $\lambda$ and $q$.

| $\lambda$ | $q = 10$ | | $q = 15$ | | $q = 20$ | |
|---|---|---|---|---|---|---|
| | BRANCHES | OSDT | BRANCHES | OSDT | BRANCHES | OSDT |
| 0.1 | $10^4$ | $10^{13}$ | $10^5$ | $10^{16}$ | $10^6$ | $10^{18}$ |
| 0.05 | $10^5$ | $10^{271}$ | $10^7$ | $10^{473}$ | $10^9$ | $10^{576}$ |
| 0.01 | $10^5$ | $10^{392}$ | $10^8$ | INF | $10^{10}$ | INF |

*where $\kappa = \min\left\{\left\lfloor \frac{1}{K\lambda}\right\rfloor - 1, q\right\}$.*

To our knowledge, only (Hu et al., 2019, Theorem E.2) performs a similar analysis for OSDT. Due to the difficulty of comparing the two bounds analytically, we rather compare them numerically. Table 1 shows that our complexity bound is significantly smaller than the bound in (Hu et al., 2019, Theorem E.2) for different settings.

## 5. Experiments

We employ 11 datasets from the UCI repository. For each dataset we use different types of encodings: suffix -o indicates ordinal encoding, suffix -f indicates binary (one-hot) encoding where the first category of each feature is dropped and no suffix designates a binary encoding retaining all categories. An additional binary encoding dropping the last category is considered in monk1-l, the reason being to showcase that different dropping options can lead to widely differing solutions. Moreover, our motivation behind dropping a category in the first place pertains to reducing the number of resulting binary features, this in turn can greatly simplify the task of finding an optimal sparse DT and help the algorithms be more scalable as we observe in Table 2. The downside however is that dropping a category can also yield more complex DT solutions than keeping all the categories during one-hot encoding. We note that the algorithms we compare with exclusively consider binary features, thus necessitating a preliminary binary encoding. This seemingly benign detail can significantly harm performance by introducing a large amount of unnecessary splits as we explain in Appendix E. BRANCHES can sidestep this issue because it supports non-binary DTs on ordinal encodings of the data, these types of DT structures are commonly known under the name multi-way splits.

We set a time limit of 5 minutes for all algorithms and we run all the experiments on a personal Machine (Processor: 2,6 GHz 6-Core Intel Core i7; Memory: 16 GB). Moreover, since it is necessary to fix a maximum depth for DFS methods, we set it to 20 for MurTree and STreeD while the BFS methods GOSDT and BRANCHES run with infinite maximum depth. The aim being to analyse performance at a large maximum depth in these experiments. The results are summarised in Table 2. For a detailed comparison across

*Table 2.* Comparing BRANCHES with the state of the art for a large maximum depth 20. BRANCHES is the only method that is applicable to ordinal encoding, hence why we put ⎯ for the remaining methods. Furthermore, we ran into memory issues with MurTree that kill the kernel, in these cases also we put ⎯. For STreeD, when it reaches timeout, it does not propose a solution due to its lack of anytime behaviour, we indicate those cases with ⎯ as well. The APIs of STreeD and MurTree do not provide the number of iterations.

| Dataset | MurTree | | | | STreeD | | | | GOSDT | | | | | BRANCHES | | | | |
|---|---|---|---|---|---|---|---|---|---|---|---|---|---|---|---|---|---|---|
| | objective | accuracy | splits | time (s) | objective | accuracy | splits | time (s) | objective | accuracy | splits | time (s) | iterations | objective | accuracy | splits | time (s) | iterations |
| monk1 | **0.940** | **1** | **6** | **0.04** | **0.940** | **1** | **6** | 3.28 | **0.940** | **1** | **6** | 3.05 | 83928 | **0.940** | **1** | **6** | **0.05** | **146** |
| monk1-l | **0.930** | **1** | **7** | **0.03** | **0.930** | **1** | **7** | 2.80 | **0.930** | **1** | **7** | 0.87 | 29770 | **0.930** | **1** | **7** | **0.02** | **117** |
| monk1-f | **0.983** | **1** | **17** | **0.05** | **0.983** | **1** | **17** | 6.18 | **0.983** | **1** | **17** | 4.09 | 92782 | **0.983** | **1** | **17** | 0.39 | **2125** |
| monk1-o | ⎯ | ⎯ | ⎯ | ⎯ | ⎯ | ⎯ | ⎯ | ⎯ | ⎯ | ⎯ | ⎯ | ⎯ | ⎯ | **0.900** | **1** | **10** | **0.02** | **64** |
| monk2 | 0.967 | 1 | 33 | 0.55 | ⎯ | ⎯ | ⎯ | *TO* | **0.968** | **1** | **32** | 91.9 | 392759 | **0.968** | **1** | **32** | 14.2 | **60611** |
| monk2-f | 0.922 | 1 | 78 | 1.07 | ⎯ | ⎯ | ⎯ | *TO* | **0.933** | **1** | **67** | 9.78 | 149912 | **0.933** | **1** | **67** | 2.94 | **28968** |
| monk2-o | ⎯ | ⎯ | ⎯ | ⎯ | ⎯ | ⎯ | ⎯ | ⎯ | ⎯ | ⎯ | ⎯ | ⎯ | ⎯ | **0.955** | **1** | **45** | **0.18** | **1213** |
| monk3 | 0.978 | 1 | 25 | 0.04 | **0.985** | **1** | **15** | 7.87 | **0.985** | **1** | **15** | 25.2 | 185974 | **0.985** | **1** | **15** | 4.05 | **14807** |
| monk3-f | 0.975 | 1 | 25 | 0.04 | **0.983** | **1** | **17** | 4.82 | **0.983** | **1** | **17** | 2.09 | 59151 | **0.983** | **1** | **17** | 0.36 | **3026** |
| monk3-o | ⎯ | ⎯ | ⎯ | ⎯ | ⎯ | ⎯ | ⎯ | ⎯ | ⎯ | ⎯ | ⎯ | ⎯ | ⎯ | **0.987** | **1** | **13** | **0.03** | **156** |
| tic-tac-toe | ⎯ | ⎯ | ⎯ | ⎯ | ⎯ | ⎯ | ⎯ | *TO* | 0.757 | 0.792 | 7 | *TO* | 2279999 | **0.838** | **0.928** | **18** | *TO* | **390000** |
| tic-tac-toe-f | **0.850** | **0.945** | **19** | 16.4 | **0.850** | **0.945** | **19** | 207 | **0.850** | **0.945** | **19** | 57.8 | 1670379 | **0.850** | **0.945** | **19** | 16.3 | **74627** |
| tic-tac-toe-o | ⎯ | ⎯ | ⎯ | ⎯ | ⎯ | ⎯ | ⎯ | ⎯ | ⎯ | ⎯ | ⎯ | ⎯ | ⎯ | **0.773** | **0.858** | **17** | **0.68** | **3339** |
| car-eval | **0.852** | **0.927** | **15** | 154 | ⎯ | ⎯ | ⎯ | *TO* | **0.852** | **0.927** | **15** | *TO* | 5893659 | **0.852** | **0.927** | **15** | 204 | **456452** |
| car-eval-f | **0.799** | **0.869** | **14** | 58 | ⎯ | ⎯ | ⎯ | *TO* | **0.799** | **0.869** | **14** | 21.1 | 927221 | **0.799** | **0.869** | **14** | 26.6 | **108640** |
| car-eval-o | ⎯ | ⎯ | ⎯ | ⎯ | ⎯ | ⎯ | ⎯ | ⎯ | ⎯ | ⎯ | ⎯ | ⎯ | ⎯ | **0.812** | **0.882** | **14** | **0.09** | **579** |
| nursery | ⎯ | ⎯ | ⎯ | ⎯ | ⎯ | ⎯ | ⎯ | *TO* | 0.810 | 0.860 | 5 | *TO* | 299999 | **0.812** | **0.872** | **6** | *TO* | **110000** |
| nursery-f | **0.772** | **0.842** | **7** | 151 | ⎯ | ⎯ | ⎯ | *TO* | 0.765 | 0.835 | 7 | *TO* | 629999 | **0.772** | **0.842** | **7** | 24.9 | **48063** |
| nursery-o | ⎯ | ⎯ | ⎯ | ⎯ | ⎯ | ⎯ | ⎯ | ⎯ | ⎯ | ⎯ | ⎯ | ⎯ | ⎯ | **0.822** | **0.892** | **7** | **0.24** | **195** |
| mushroom | **0.955** | **0.985** | **3** | 11.1 | **0.955** | **0.985** | **3** | 10.8 | 0.925 | 0.945 | 2 | *TO* | 79999 | **0.955** | **0.985** | **3** | *TO* | **21000** |
| mushroom-f | **0.945** | **0.985** | **4** | 143 | **0.945** | **0.985** | **4** | 126 | 0.925 | 0.945 | 2 | *TO* | 99999 | **0.945** | **0.985** | **4** | *TO* | **24000** |
| mushroom-o | ⎯ | ⎯ | ⎯ | ⎯ | ⎯ | ⎯ | ⎯ | ⎯ | ⎯ | ⎯ | ⎯ | ⎯ | ⎯ | **0.975** | **0.985** | **1** | **0.15** | **6** |
| kr-vs-kp | ⎯ | ⎯ | ⎯ | ⎯ | ⎯ | ⎯ | ⎯ | *TO* | 0.815 | 0.845 | 3 | *TO* | 159999 | **0.900** | **0.940** | **4** | *TO* | **46000** |
| zoo | 0.989 | 1 | 11 | 0.04 | **0.992** | **1** | **8** | 184 | **0.992** | **1** | **8** | 87.3 | 401799 | **0.992** | **1** | **8** | 44.6 | **39199** |
| zoo-f | 0.989 | 1 | 11 | 0.3 | **0.992** | **1** | **8** | 23.4 | **0.992** | **1** | **8** | 33.5 | 300387 | **0.992** | **1** | **8** | 2.09 | **4659** |
| zoo-o | ⎯ | ⎯ | ⎯ | ⎯ | ⎯ | ⎯ | ⎯ | ⎯ | ⎯ | ⎯ | ⎯ | ⎯ | ⎯ | **0.993** | **1** | **7** | **0.87** | **1456** |
| lymph | ⎯ | ⎯ | ⎯ | ⎯ | ⎯ | ⎯ | ⎯ | *TO* | 0.790 | 0.810 | 2 | *TO* | 659999 | **0.828** | **0.898** | **7** | *TO* | **100000** |
| lymph-f | ⎯ | ⎯ | ⎯ | ⎯ | ⎯ | ⎯ | ⎯ | *TO* | 0.784 | 0.804 | 2 | *TO* | 1079999 | **0.811** | **0.891** | **8** | *TO* | **170000** |
| lymph-o | ⎯ | ⎯ | ⎯ | ⎯ | ⎯ | ⎯ | ⎯ | ⎯ | ⎯ | ⎯ | ⎯ | ⎯ | ⎯ | **0.852** | **0.952** | **10** | 12.3 | **16154** |
| balance | ⎯ | ⎯ | ⎯ | ⎯ | ⎯ | ⎯ | ⎯ | *TO* | 0.712 | 0.737 | 5 | *TO* | 1119999 | **0.776** | **0.806** | **8** | *TO* | **640000** |
| balance-f | ⎯ | ⎯ | ⎯ | ⎯ | ⎯ | ⎯ | ⎯ | *TO* | **0.673** | **0.723** | **10** | 162 | 2292545 | **0.673** | **0.723** | **10** | 190 | **676149** |
| balance-o | ⎯ | ⎯ | ⎯ | ⎯ | ⎯ | ⎯ | ⎯ | ⎯ | ⎯ | ⎯ | ⎯ | ⎯ | ⎯ | **0.713** | **0.763** | **10** | **0.004** | **178** |

a wide range of maximum depths, refer to Appendix H.4. Table 4 summarises the characteristics of the datasets we consider.

For MurTree, the implementation we used from `https://github.com/MurTree/pymurtree.git` displays a suboptimal behaviour (in terms of $\mathcal{H}_\lambda$) unlike STreeD, GOSDT and BRANCHES. This happens for monk2, monk2-f, monk3, monk3-f, car-eval-f, nursery-f, zoo and zoo-f as shown in Table 2. According to `https://bitbucket.org/EmirD/murtree/src/master/`, the authors of MurTree indicate the release of a *a newer and more general version of the algorithm* referring to STreeD. For this reason, we will focus the remaining discussion on STreeD, GOSDT and BRANCHES.

**Comparing STreeD with BRANCHES and GOSDT:** STreeD is always optimal when it terminates similarly to GOSDT and BRANCHES. On mushroom and mushroom-f it is clearly the superior method as it terminates in $10.8s$ and $126s$ respectively while both GOSDT and BRANCHES reach

timeout. In Appendix H.4 we show that the main reason behind STreeD's success on mushroom is the depth 2 solver, a technique introduced by Demirović et al. (2022). On the other hand, despite reaching timeout, BRANCHES still proposes the true optimal sparse DT unlike GOSDT, thus showcasing the superiority of BRANCHES' anytime behaviour. On the remaining datasets, STreeD's DFS strategy suffers from the large maximum depth while GOSDT and BRANCHES achieve optimality significantly faster even while running at an infinite maximum depth, this is especially the case for BRANCHES because STreeD outperforms GOSDT on monk3, monk3-f, mushroom, mushroom-f and zoo-f, while it outperforms BRANCHES only on mushroom and mushroom-f. Furthermore, STreeD's lack of anytime behaviour prevents it from proposing a solution when it reaches timeout. This happens for monk2, monk2-f, tic-tac-toe, car-eval, car-eval-f, nursery, nursery-f, kr-vs-kp, lymph, lymph-f, balance and balance-f.

The superiority of BRANCHES and GOSDT over STreeD on these experiments is mainly due to their BFS strategies,

which tend to find the optimal solution quicker than DFS at the expense of more memory consumption. We note that while neither BRANCHES nor GOSDT ran out of memory on these experiments, such an undesirable outcome can happen in some cases. To alleviate this problem, hybrid methods, using BFS until exhausting a memory budget then switching to DFS, can be employed (Pearl, 1984, Section 2.5). These extensions are outside the scope of our current work and we will consider them in a future work.

**Comparing BRANCHES with GOSDT:** BRANCHES outperforms GOSDT on all the experiments except car-eval-f and balance-f where GOSDT terminates faster. Sometimes the difference in speed is so significant that BRANCHES terminates while GOSDT reaches timeout and as a consequence fails to produce the optimal solution. This happens in car-eval and nursery-f. Moreover, whenever both algorithms reach timeout, BRANCHES proposes a better solution than GOSDT's, hence displaying a better anytime behaviour. To see this, refer to tic-tac-toe, nursery, mushroom, kr-vs-kp, lymph, lymph-f and balance. Another important point is that the experiments on ordinal encoded datasets terminate significantly faster than their binary encoded counterparts. Indeed, no experiment with ordinal encoding reaches timeout, and the slowest one is lymph-o with $12.3s$. In Appendix E we discuss the reasons behind this phenomenon. Furthermore, on many occasions, the non-binary optimal sparse DTs obtained with ordinal encoding are of better quality (in terms of $\mathcal{H}_\lambda$) than those obtained through a preliminary binary encoding. This happens for monk3-o, nursery-o, mushroom-o, zoo-o and lymph-o, but this is not the case for the remaining datasets. In fact, we shall see in Appendix E that the obtained non-binary sparse DTs can themselves be *collapsed* into sparser solutions with fewer splits. Knowing that scalability is a major issue in the literature of optimal sparse DTs, the ability to support non-binary DTs is very helpful in this regard. BRANCHES being the only method that displays this property has a clear advantage from a scalability standpoint.

Additional experiments are provided in Appendix H:

- Appendix H.1 compares BRANCHES with Python implemented algorithms OSDT and PYGOSDT.

- Appendix H.2 compares BRANCHES, CART and DL8.5 in terms of their Pareto fronts with respect to the joint optimisation of accuracy and number of splits.

- Appendix H.3 investigates the suboptimality of DL8.5 with respect to the joint optimisation of accuracy and number of splits.

- Appendix H.4 extends the comparison between BRANCHES, GOSDT and STreeD to a wide range of values for the maximum depth.

## 6. Conclusion and Future Work

In this work, we have developed BRANCHES, a novel algorithm seeking optimal sparse DTs. BRANCHES leverages an AO* approach to solve the problem within an AND/OR graph search framework and employs a custom heuristic called the Purification bound. We have shown that BRANCHES is optimal and satisfies better theoretical complexity guarantees than those derived in the literature. Furthermore, we illustrated through multiple experiments that BRANCHES outperforms the state of the art in terms of runtime, number of iterations and anytime behaviour. It is especially worth noting that BRANCHES outperforms its C++ competitors in runtime despite being currently implemented in Python, which indicates that a future C++ implementation of BRANCHES promises to widen the performance gap further with the other methods.

There are several ideas to extend this work in the future. A straightforward one is to implement BRANCHES in C++ and incorporate multi-threading as follows: During the Selection step, whenever we take a split action, we could choose a subset of the children branches and keep running Selection from each one of them in parallel. As a consequence, the Selection step would return a subset of branches, for which we run Expansion in parallel then Backpropagation in a synchronous parallel fashion. This would yield a notably better pruning of the search graph and as a result a significantly faster optimal convergence. Another promising idea is to incorporate the depth-2 solver (Demirović et al., 2022), we saw in Appendix H.4 how it is crucial to STreeD's performance. Moreover, to alleviate high memory consumption issues we could design a hybrid strategy between BFS and DFS as described in (Pearl, 1984, Section 2.5). Lastly, we could improve the Purification Bound by using a strong auxiliary classifier $f$ such as a Multi-Layer Perceptron. Given a branch $l$ for which we want to initialise the heuristic $\mathcal{V}(l)$, if we can derive the number $m$ of employed variables by $f$ in $l$ then we can tighten the Purification Bound with:

$$\mathcal{V}(l) = \max\left\{\mathcal{H}(l), -m\lambda + \mathbb{P}\left[l(X) = 1, f(X) = Y\right]\right\}$$

## Acknowledgements

We would like to thank Mathijs M. de Weerdt and Pierre Schaus for their helpful discussions, and the anonymous reviewers for their valuable feedback.

## Impact Statement

This paper presents work whose goal is to advance the field of Machine Learning. There are many potential societal consequences of our work, none which we feel must be specifically highlighted here.

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

# A. Notation



*Table 3.* Table of Notation



| | | |
|---:|:---:|:---|
| $X$ | $=$ | $\left(X^{(1)}, \ldots, X^{(q)}\right)$, a datum. |
| $X^{(i)}$ | $\in$ | $\{1, \ldots, C_i\}$, a feature. |
| $Y$ | $\in$ | $\{1, \ldots, K\}$, a class. |
| $\mathcal{D}$ | $=$ | $\{(X_m, Y_m)\}_{m=1}^{n}$, dataset of examples. |
| $l$ | $=$ | $\bigwedge_{v=1}^{\mathcal{S}(l)} \mathbb{I}\left\{X^{(i_v)} = j_v\right\}$ a branch. Also, a non-terminal state. |
| $\mathcal{S}(l)$ | $\triangleq$ | The size of branch $l$, the number of splits in $l$, the number of clauses in $l$. |
| $l(X)$ | $\triangleq$ | Valuation of $l$ for input $X$. When $l(X) = 1$, we say that $X$ is in $l$. |
| $\Omega$ | $\triangleq$ | The root. Branch that valuates to 1 for all possible inputs. |
| $\mathrm{Ch}(l, i)$ | $\triangleq$ | Children of $l$ when splitting it with respect to feature $i$. |
| $\mathrm{Ch}(l, i)$ | $=$ | $\{l_1, \ldots, l_{C_i}\}, l_j = l \wedge \mathbb{I}\{X^{(i)} = j\}$ |
| $n(l)$ | $=$ | $\sum_{m=1}^{n} l(X_m)$ Number of examples in $l$. |
| $n_k(l)$ | $=$ | $\sum_{m=1}^{n} l(X_m) \mathbb{I}\{Y_m = k\}$ Number of examples in $l$ of class $k$. |
| $k^*(l)$ | $=$ | $\mathrm{Argmax}_{1 \leq k \leq K}\{n_k(l)\}$, majority class in $l$ |
| $\mathcal{H}(l)$ | $=$ | $\mathbb{P}[l(X) = 1, Y = k^*(l)]$, probability that an example is in $l$ and is correctly classified. |
| sub-DT rooted in $l$ | $\triangleq$ | Collection of branches tha stem from a series of splits from $l$. |
| DT | $\triangleq$ | A sub-DT rooted in the root $\Omega$. |
| $T(X)$ | $\triangleq$ | Predicted class of $X$ by $T$. Majority class of the branch containing $X$. |
| $\mathcal{H}(T)$ | $=$ | $\mathbb{P}[T(X) = Y]$, accuracy of DT $T$. |
| $\mathcal{H}_\lambda(T)$ | $=$ | $\mathbb{P}[T(X) = Y] - \lambda \mathcal{S}(T)$, the objective evaluating $T$. |
| $\mathcal{S}(T)$ | $\triangleq$ | Number of splits to construct $T$ from the branch where it is rooted. |
| $\lambda$ | $\in$ | $]0, 1]$, penalty parameter. |
| $T^*$ | $=$ | $\mathrm{Argmax}_T\{\mathcal{H}_\lambda(T)\}$, optimal DT. |
| $\overline{a}$ | $\triangleq$ | Terminal action. |
| $\overline{l}$ | $\triangleq$ | Terminal state that stems from taking $\overline{a}$ in $l$. |
| $\mathcal{V}^\pi(l)$ | $\triangleq$ | Value of policy $\pi$ starting at non-terminal state $l$. |
| $\mathcal{Q}^\pi(T, a)$ | $\triangleq$ | State-action value of policy $\pi$ at non-terminal state $l$ and action $a$. |
| $T_l^\pi$ | $\triangleq$ | Sub-DT of $\pi$ rooted in $l$. See Proposition 3.1. |
| $\tau_l^\pi$ | $\triangleq$ | Minimum time for policy $\pi$ to get to $T_l^\pi$. |
| $T^\pi$ | $\equiv$ | $T_\Omega^\pi$ |
| $\pi^*$ | $\in$ | $\mathrm{Argmax}_\pi \mathcal{V}^\pi(\Omega)$, an optimal policy. |
| $T^*$ | $=$ | $T^{\pi^*}$ |
| $\mathcal{V}^*$ | $\equiv$ | $\mathcal{V}^{\pi^*}$ |
| $\mathcal{Q}^*$ | $\equiv$ | $\mathcal{Q}^{\pi^*}$ |
| $\tau_l^*$ | $\equiv$ | $\tau_l^{\pi^*}$ |
| $\mathcal{V}(l)$ | $\triangleq$ | Heuristic estimate of $\mathcal{V}^*(l)$. |
| $\mathcal{Q}(l, a)$ | $\triangleq$ | Heuristic estimate of $\mathcal{Q}^*(l, a)$ |
| $\widetilde{\pi}$ | $\triangleq$ | Selection policy $\widetilde{\pi}(l) = \mathrm{Argmax}_{a \in \mathcal{A}(l)} \mathcal{Q}(l, a)$. |

# B. Choosing an UNSOLVED branch during Selection

Regarding the choice of an UNSOLVED branch introduced in the Selection process in Section 4.2, we choose branch $l$ of lowest $\mathcal{H}_\lambda\left(\widehat{T}_l\right)$ where $\widehat{T}_l$ is the best sub-DT rooted in $l$ found so far in terms of the objective $\mathcal{H}_\lambda$. The reason is to quickly prune branches that contribute low $\mathcal{H}_\lambda\left(\widehat{T}_l\right)$ to the solution. Indeed, this approach either yields better subtrees rooted in these branches, or bad subtrees that indicate this region to be unpromising, hence incentivizing the algorithm to search elsewhere. This selection approach is likely to induce a good anytime behaviour. We note that a similar idea was proposed

in (Nilsson, 2014, p.107):

> *Possibly the expansion of that leaf node having the highest h value would most likely result in a changed estimate.*

The $h$ value in (Nilsson, 2014, p.107) refers to the heuristic value, which is akin to our $\mathcal{V}$ value albeit for a minimisation context instead of a maximisation one. As such, a direct application of (Nilsson, 2014, p.107) to our setting would yield choosing the UNSOLVED branch $l$ with lowest $\mathcal{V}(l)$. We noticed in practice that employing $\mathcal{H}_\lambda\left(\widehat{T}_l\right)$ instead yields better anytime behaviour for the reasons explained above. Currently, we have not investigated other strategies of this type, and it is a promising idea for future work.

## C. Anytime behaviour

There are settings where the search takes a substantial or even unfeasible amount of time to terminate. In these situations, we set a time limit on BRANCHES' execution and even if it does not terminate within this time, we expect it to still return a good solution. This property is called the anytime behaviour. For BFS methods, it is straightforward to implement while it is less trivial for DFS algorithms. STreeD for example, does not enjoy the anytime property.

We formulate BRANCHES' anytime behaviour as follows. For each branch $l$ that is currently in the memo, we define its greedy value $\widehat{\mathcal{V}}(l)$ recursively as:

$$\widehat{\mathcal{V}}(l) = \begin{cases} \mathcal{H}(l) \text{ if } l \text{ is a tip node (leaf) of the current search graph } G. \\ \max_{a \in \mathcal{A}(l) \setminus \{\overline{a}\}} \left\{ \mathcal{H}(l), -\lambda + \sum_{l' \in \text{Ch}(l,a)} \widehat{\mathcal{V}}(l') \right\} \end{cases} \tag{16}$$

Basically $\widehat{\mathcal{V}}(l)$ is the best value of the regularised objective found so far for all sub-DTs rooted in $l$. Based on this definition, we also introduce the greedy state-action values and the greedy policy:

$$\widehat{\mathcal{Q}}(l,a) = \begin{cases} \mathcal{H}(l) \text{ if } a = \overline{a} \\ -\lambda + \sum_{l' \in \text{Ch}(l,a)} \widehat{\mathcal{V}}(l') \text{ if } a \in \mathcal{A}(l) \setminus \{\overline{a}\} \end{cases}$$

$$\widehat{\pi}(l) = \text{Argmax}_{a \in \mathcal{A}(l)} \widehat{\mathcal{Q}}(l,a)$$

The estimates $\widehat{\mathcal{V}}(l)$ and $\widehat{\mathcal{Q}}(l,a)$ are updated in similar fashion to $\mathcal{V}(l)$ and $\mathcal{Q}(l,a)$ during the Backpropagation step, i.e. via (12) and (11). When BRANCHES reaches timeout we unroll $\widehat{\pi}$ from $\Omega$ to get $T_\Omega^{\widehat{\pi}}$ the best DT found so far in terms of the objective $\mathcal{H}_\lambda$. $T_\Omega^{\widehat{\pi}}$ is the proposed solution by BRANCHES when it runs out of time.

Unfortunately, unlike settings where BRANCHES terminates, the anytime behaviour does not provide any theoretical guarantees with respect to the quality of the proposed solution. Such guarantees necessitate the assumption of some smoothness properties of $\mathcal{H}_\lambda$ on the state space $\mathcal{S}$. Nevertheless, the use of selection strategies such as the one we propose in Appendix B can lead to empirically satisfying results for BRANCHES' anytime behaviour.

## D. Implementation Details

The search strategy we introduced in Section 4.2 is an abstract description of BRANCHES. In this section, we provide concrete elements for the implementation of the algorithm, along with micro-optimisation techniques that substantially improve its computational efficiency.

### D.1. Branch objects

For each branch $l = \bigwedge_{v=1}^{\mathcal{S}(l)} \mathbb{I}\{X^{(i_v)} = j_v\}$, we define an object with the following elements:

- `id_branch`: $l$ is identified with the unique string $"(i_1, j_1)(i_1, j_2) \ldots (i_{\mathcal{S}(l)}, j_{\mathcal{S}(l)})"$. We recall that this string is unique because we impose the condition $i_1 < i_2 < \ldots < i_{\mathcal{S}(l)}$. We store each encountered branch in a memo dictionary using its identifier.

- `attributes_categories`: Dictionary containing the number of categories per unused feature in $l$. We recall that the set of unused features is the set of split actions.

- `bit_vector`: Vector of the indices of the data contained in $l$. This vector allows quick access to the data in $l$.

- `children`: Dictionary containing the children of $l$, i.e. the set Ch $(l, i)$ for each unused feature $i$ in $l$. Initialised with an empty dictionary.

- `attribute_opt`: The greedy action $\widehat{\pi}(l)$ as per the definition in Appendix C. If $\widehat{\pi}(l) = \overline{a}$ then we set `attribute_opt` to None.

- `terminal`: Boolean describing whether $l$ is terminal or not.

- `complete`: Boolean describing whether $l$ is SOLVED or not.

- `value`: The estimated value $\mathcal{V}(l)$.

- `value_terminal`: The value of the terminal action at $l$.

$$\mathcal{Q}(l, \overline{a}) = \mathcal{Q}^*(l, \overline{a}) = \mathcal{H}(l) = \mathbb{P}[l(X) = 1, k^*(l) = Y] = \frac{n_{k^*(l)}(l)}{n}$$

- `value_greedy`: The greedy value $\widehat{\mathcal{V}}(l)$ as per the definition in Appendix C.

- `freq`: Proportion of examples in $l$:

$$\texttt{freq} = \mathbb{P}[l(X) = 1] = \frac{n(l)}{n} = \frac{1}{n}\sum_{m=1}^{n} l(X_m)$$

- `pred`: Majority class at $l$:

$$\texttt{pred} = k^*(l) = \mathrm{Argmax}_{1 \leq k \leq K} n_k(l)$$

- `queue`: Heap queue containing (-value, -value_complete, attribute, children) tuples. For each unused feature (split action) `attribute`: `value` is the estimate:

$$\texttt{value} = \mathcal{Q}(l, \texttt{attribute}) = -\lambda + \sum_{l' \in \texttt{Ch}(l, \texttt{attribute})} \mathcal{V}(l')$$

On the other hand, `value_complete` is the sum of the estimated values $\mathcal{V}(l')$ of the children $l' \in$ Ch$(l, \texttt{attribute})$ that are SOLVED. By definition, the SOLVED children $l'$ satisfy $\mathcal{V}(l') = \mathcal{V}^*(l')$, we store the sum of their values in `value_complete`, which serves to efficiently update $\mathcal{Q}(l, \texttt{attribute})$ during the Backpropagation step. `children` is a dictionary containing the UNSOLVED children, it is from this dictionary that we choose the next branch to visit during the Selection step. During Backpropagation, If an UNSOLVED branch $l'$ in `children` becomes SOLVED, it is discarded from `children`. We note that these tuples are stored in `queue`, thus the first element of `queue` is always the tuple with the highest `value`, i.e. `queue[0][2]` is the split action maximising $\mathcal{Q}(l, a)$. This is the reason we store $-$value in the tuple, Python implements min heaps instead of max heaps. We do not need to sort all actions by their values, but rather to just keep track of the action with the highest value. As a result, $l$ becomes SOLVED if and only if one of the following holds:

  - The terminal action is the current best action:

$$\mathcal{Q}(l, \overline{a}) = \mathrm{Argmax}_{a \in \mathcal{A}(l)} \mathcal{Q}(l, a)$$

    which happens if:

$$-\texttt{queue[0][0]} \leq \texttt{value\_terminal}$$

  - The dictionary of UNSOLVED children that result from taking the current best split action in $l$ `queue[0][3]` is empty.

There is an additional benefit to storing `-value_complete`. When there are multiple split actions `attribute` that maximise `value`, then we prioritise the one maximising `value_complete`. This further serves as a heuristic allowed by the flexibility of BRANCHES' selection step compared to GOSDT's. To see the benefit of this scheme, consider such situation and suppose that one of these split actions, `attribue`, maximising the estimate is such that $\text{Ch}(l, \texttt{attribute})$ are all SOLVED. Ideally we want to prioritise this action, i.e. we want `queue[0][2] = attribute`. The reason behind this is that it allows us to immediately conclude that $l$ is SOLVED and that $\texttt{attribute} = \text{Argmax}_{a \in \mathcal{A}(l) \setminus \{\overline{a}\}} \mathcal{Q}^*(l, a)$. This is exactly what happens by using a priority based on `value_complete` as well. Indeed, if $\text{Ch}(l, \texttt{attribute})$ are all SOLVED, then by definition `attribute` maximises `value_complete` among all the split actions maximising `value`.

## D.2. The Algorithm

In this section, we go over BRANCHES' search strategy, introduced in Section 4.2, and we outline it from an implementation perspective. We initialise the root $\Omega$, then we apply the search steps at each iteration as follows:

**Selection:** Initialise the current branch $l = \Omega$ and the path list to `path = [l]`. While $l$ is UNSOLVED and $l.\texttt{children}$ is not empty, i.e. $l$ has been expanded. Consider the tuple:

$$(\texttt{-value}, \texttt{-value\_complete}, \texttt{attribute}, \texttt{children}) = l.\texttt{queue}[0]$$

As we have seen in Appendix D.1, `attribute` is the optimal split action with respect to the current estimates $\mathcal{Q}(l, a)$ and `children` is the subset of UNSOLVED children in $\text{Ch}(l, \texttt{attribute})$. Therefore, we choose the next branch $l$ from the dictionary `children`. As explained in Appendix B, here choosing the UNSOLVED child of lowest `value_greedy` is our practical choice. Append $l$ to `path`.

**Expansion:** Let $l$ be the Selected branch. If $l.\texttt{complete}$, we go to the Backpropagation step. Otherwise, for each (unused) feature-category pair $(i, j) \in l.\texttt{attributes\_categories}$ let $l_{ij} = l \wedge \mathbb{I}\{X^{(i)} = j\}$ be the child branch of $l$ that corresponds to feature $i$ taking the value $j$. Our objective is to calculate $\mathcal{V}(l_{ij})$. We first check whether $l_{ij}.\texttt{id\_branch}$ is in the memo, if it is, then we can directly access $\mathcal{V}(l_{ij})$. Otherwise, we initialise $\mathcal{V}(l_{ij})$ according to Proposition 4.2. To do this efficiently, consider a fixed feature $i$ and let us go over its categories $j \in \{1, \ldots, C_i\}$ one by one. For $l_{i1}$, we first extract the data in $l$ using $l.\texttt{bit\_vector}$:

$$\mathcal{D}_l = \{X_m \in \mathcal{D} : l(X_m) = 1\} = \mathcal{D}[l.\texttt{bit\_vector}]$$

Since $l_{i1}(X) = 1 \implies l(X) = 1$, we can extract the data in $l_{i1}$ directly from the smaller set $\mathcal{D}_l$ instead of $\mathcal{D}$:

$$\mathcal{D}_{l_{i1}} = \{X_m \in \mathcal{D} : l_{i1}(X_m) = 1\} = \{X_m \in \mathcal{D}_l : l_{i1}(X_m) = 1\}$$

The indices of the data in $\mathcal{D}_{l_{i1}}$ form the vector $l_{i1}.\texttt{bit\_vector}$. Now we can initialise $\mathcal{V}(l_{i1})$ with Proposition 4.2 using $\mathcal{D}_{l_{i1}}$. For $l_{i2}$, if $l_{i2}.\texttt{id\_branch}$ is not in the memo, then to initialise $\mathcal{V}(l_{i2})$, instead of extracting $\mathcal{D}_{l_{i2}}$ from $\mathcal{D}_l$ via:

$$\mathcal{D}_{l_{i2}} = \{X_m \in \mathcal{D}_l : l_{i2}(X_m) = 1\}$$

We rather use the fact that $l_{i1}$ and $l_{i2}$ are mutually exclusive, in the sense that:

$$\forall X \in \mathcal{X} : l_{i2}(X) = 1 \implies l_{i1}(X) = 0$$

Which means that we can extract $\mathcal{D}_{l_{i2}}$ from the smaller set $\mathcal{D}_l \setminus \mathcal{D}_{l_{i1}}$ instead of $\mathcal{D}_l$ and then initialise $\mathcal{V}(l_{i2})$. We repeat this process for all categories $j \in \{1, \ldots, C_i\}$ and then we do the same thing for the remaining unused features in $l.\texttt{attributes\_categories}$. These micro-optimisations we perform allow for a substantial computational gain.

**Backpropagation:** For $j = length(\texttt{path}) - 1, \ldots, 1$ let `parent = path[j-1]` and `child = path[j]`, then we pop the heap queue `parent.queue`:

$$(\texttt{-value}, \texttt{-value\_complete}, \texttt{attribute}, \texttt{children}) = \texttt{parent.queue.pop()}$$

During the Selection step, `attribute` was the action taken at the branch `parent` to transition to the branch `child`. Now during Backpropagation, we need to update the estimates $\mathcal{Q}(\texttt{parent}, \texttt{attribute})$ and $\mathcal{V}(\texttt{parent})$, hence why

we pop the corresponding tuple from `parent.queue`, and once we update its values, we will push the tuple back in the heap queue. If `child.complete` then we add its value to `value_complete`:

$$\texttt{value\_complete} \leftarrow \texttt{value\_complete + child.value}$$

and we pop `child` from the dictionary of UNSOLVED children `children.pop(child)`. Now we push the tuple back in `parent.queue`, this rearranges the tuples is such a way that the most promising one (with maximum `value`) is at `parent.queue[0]`. The next step is to check:

$$\texttt{(-value, -value\_complete, attribute, children)} = \texttt{parent.queue[0]}$$

if `attribute` is the same attribute we have just treated with the last tuple, then we stop the update of `parent.queue`. Otherwise, we pop the queue again and we update this tuple in similar fashion to what we did with the previous tuple. The reason we check this is that an update could already be made here because the UNSOLVED children in dictionary `children` might have been updated during other iterations of BRANCHES. We continue this process until we get an `attribute` that has already been treated in this process, in which case there is no need for further updates. Now `parent.queue[0]` is the tuple corresponding to the best split action:

$$\texttt{(-value, -value\_complete, attribute, children)} = \texttt{parent.queue[0]}$$

Therefore, the value of `parent` is equal to the maximum between the value of taking this best split action and the value of taking the terminal action:

$$\mathcal{V}\left(\texttt{parent}\right) = \max\left\{\mathcal{Q}\left(\texttt{parent}, \overline{a}\right), \mathcal{Q}\left(\texttt{parent}, \texttt{attribute}\right)\right\}$$

Which, in our implementation translates to:

$$\texttt{parent.value} \leftarrow \max\left\{\texttt{parent.value\_terminal}, \texttt{value}\right\}$$

Moreover, if $\mathcal{V}\left(\texttt{parent}\right) = \mathcal{Q}\left(\texttt{parent}, \overline{a}\right)$, then $\overline{a} = \text{Argmax}_{a \in \mathcal{A}(\texttt{parent})} \mathcal{Q}\left(\texttt{parent}, a\right)$, and since we know that $\mathcal{Q}^*\left(\texttt{parent}, \overline{a}\right) = \mathcal{Q}\left(\texttt{parent}, \overline{a}\right)$ (according to Equation (10)), then we deduce that `parent` is SOLVED and $\mathcal{V}^*\left(\texttt{parent}\right) = \mathcal{Q}^*\left(\texttt{parent}, \overline{a}\right)$. Therefore we update:

$$\texttt{parent.complete} \leftarrow \texttt{True}$$

This is not the only condition that makes `parent` SOLVED. Indeed, `parent` can also be SOLVED if Ch (`parent, attribute`) are all SOLVED, which happens when the dictionary `children` is empty.

---

**Algorithm 1** BRANCHES

---

1: **Input:** Dataset $\mathcal{D} = \{(X_m, Y_m)\}_{m=1}^n$, penalty parameter $\lambda \geq 0$.
2: memo $\leftarrow \{\}$        ▷ Initialise an empty memo
3: INITIALISE$(\Omega, \mathcal{D})$
4: **while** not $\Omega$.complete **do**        ▷ While the root $\Omega$ is not SOLVED
5:      $(l, \text{path}) \leftarrow$ SELECT$()$
6:      **if** $l$.complete **then**        ▷ If $l$ is SOLVED
7:          BACKPROPAGATE$(\text{path})$
8:      **else**
9:          EXPAND$(l, \mathcal{D})$
10:          BACKPROPAGATE$(\text{path})$
11:      **end if**
12: **end while**
13: **return** INFER$()$
14: **procedure** SELECT$()$
15:      $l \leftarrow \Omega$
16:      $\text{path} \leftarrow [l]$
17:      **while** $l$.expanded and (not $l$.complete) **do**
18:          $(\mathcal{Q}(l, i), \text{return\_complete}, i, \text{children\_incomplete}) \leftarrow l.\text{queue}[0]$
19:          $l \leftarrow \text{children\_incomplete}[0]$
20:          $\text{path.append}(l)$
21:      **end while**
22:      **return** $(l, \text{path})$
23: **end procedure**
24: **procedure** EXPAND$(l, \mathcal{D})$
25:      $l.\text{expanded} \leftarrow True$
26:      **for** $i \in \mathcal{A}(l) \setminus \{\overline{a}\}$ **do**
27:          SPLIT$(l, i, \mathcal{D})$
28:          $\mathcal{V}(l) \leftarrow \max\left\{\mathcal{Q}(l, \overline{a}), l.\text{queue}[0][0]\right\}$        ▷ This update comes from Equation (12)
29:      **end for**
30:      **if** $\mathcal{V}(l) = \mathcal{Q}(l, \overline{a})$ **then**        ▷ In this case $\mathcal{V}^*(l) = \mathcal{Q}^*(l, \overline{a}) = \mathcal{H}(l)$
31:          $l.\text{complete} \leftarrow True$        ▷ $\mathcal{V}^*(l)$ is known
32:          $l.\text{terminal} \leftarrow True$        ▷ Label $l$ terminal if the optimal action at $l$ is $\pi^*(l) = \overline{a}$
33:      **end if**
34: **end procedure**

---

35: **procedure** BACKPROPAGATE(path)
36:     $N \leftarrow$ length (path)
37:     **for** $t = N - 2$ to $0$ **do**
38:         $l \leftarrow$ path $[t]$
39:         $(\mathcal{Q}(l, i), \text{return\_complete}, i, \text{children\_incomplete}) \leftarrow l.\text{queue.pop}()$
40:         $\mathcal{Q}(l, i) \leftarrow \text{return\_complete}$                                      ▷ Initialise $\mathcal{Q}(l, i)$
41:         **for** $l' \in$ children\_incomplete **do**
42:             $\mathcal{Q}(l, i) \leftarrow \mathcal{Q}(l, i) + \mathcal{V}(l')$
43:             **if** $l'.\text{complete}$ **then**                                      ▷ Check if $l'$ is SOLVED now
44:                 children\_incomplete.discard $(l')$                          ▷ Delete $l'$ from children\_incomplete
45:             **end if**
46:         **end for**
47:         $l.\text{queue.push}((\mathcal{Q}(l, i), \text{return\_complete}, i, \text{children\_incomplete}))$
48:         $(\mathcal{Q}(l, i^*), \text{return\_complete}, i^*, \text{children\_incomplete}) \leftarrow l.\text{queue}[0][0]$
49:         $\mathcal{V}(l) \leftarrow \mathcal{Q}(l, i^*)$
50:         **if** $(\mathcal{V}(l) = \mathcal{Q}(l, \overline{a}))$ or (children\_incomplete is empty) **then**
51:             $l.\text{complete} \leftarrow True$
52:             $l.\text{terminal} \leftarrow True$                          ▷ Label $l$ terminal if the optimal action at $l$ is $\pi^*(l) = \overline{a}$
53:         **end if**
54:     **end for**
55: **end procedure**
56: **procedure** INFER()
57:     $T \leftarrow []$
58:     $Q \leftarrow$ queue ()
59:     $Q.\text{put}(\Omega)$
60:     **while** $Q$ is not empty **do**
61:         $l \leftarrow Q.\text{pop}()$
62:         **if** $l.\text{terminal}$ **then**
63:             $T.\text{append}(l)$
64:         **else**
65:             $(\mathcal{Q}(l, i), \text{return\_complete}, i, \text{children\_incomplete}) \leftarrow l.\text{queue}[0]$
66:             **for** $l' \in l.\text{children}[i]$ **do**
67:                 $Q.\text{put}(l')$
68:             **end for**
69:         **end if**
70:     **end while**
71:     **return** $T$
72: **end procedure**

```
73: procedure INITIALISE(l, D)
74:     l.expanded ← False                                          ▷ Label l as not expanded yet
75:     l.children ← dict ()                                        ▷ Initialise the dictionary of children
76:     l.queue ← queue ([])                                        ▷ Initialise the priority queue of l
77:     Q (l, ā) ← H (l)                                            ▷ H (l) is calculated with D
78:     if A (l) = {ā} then
79:         l.terminal ← True                                       ▷ Label l as terminal if it cannot be split
80:         l.complete ← True                                       ▷ l is SOLVED, V* (l) is known
81:         V (l) ← Q (l, ā)                                        ▷ In this case V* (l) = Q* (l, ā) = H (l)
82:     else
83:         l.terminal ← False
84:         Initialise V (l) according to Equation (12) and Equation (11)
85:         if V (l) = Q (l, ā) then
86:             l.complete ← True                                   ▷ V* (l) is known, V* (l) = Q* (l, ā) = H (l)
87:             l.terminal ← True                                   ▷ Label l terminal if the optimal action at l is π* (l) = ā
88:         else
89:             l.complete ← False                                  ▷ V* (l) is still unknown
90:         end if
91:     end if
92:     memo.add(l)                                                 ▷ Add the initialised branch to the memo
93: end procedure
94: procedure SPLIT(l, i, D)
95:     l.children[i] ← []                                          ▷ Initialise the list of children that stem taking split action i in l
96:     Q (l, i) ← −λ                                               ▷ Initialise the Upper Bound Q (l, i)
97:     return_complete ← −λ                                        ▷ Initialise the return due to SOLVED children
98:     children_incomplete ← []                                    ▷ Initialise the list of UNSOLVED children
99:     for j ∈ {1, . . . , C_i} do
100:        l_ij ← l ∧ I{X^(i) = j}
101:        if l_ij ∉ memo then                                     ▷ Only initialise the branches that are not in the memo
102:            INITIALISE(l_ij, D)
103:        end if
104:        l.children[i].append (l_ij)
105:        Q (l, i) ← Q (l, i) + V (l_ij)                          ▷ Update the Upper Bound Q (l, i)
106:        if l_ij.complete then
107:            return_complete ← return_complete + V (l_ij)
108:        else
109:            children_incomplete.append (l_ij)
110:        end if
111:    end for
112:    l.queue.push ((Q (l, i) , return_complete, i, children_incomplete))
113: end procedure
```

## E. Drawbacks of Binary Encoding

In Table 2, we have shown that optimal sparse DTs are always found significantly faster in the context of ordinal encoding compared to binary encoding. In this section, we investigate the reason behind this phenomenon.

To answer this question, let us consider the following simple binary classification problem. Suppose there is only one feature $X^{(1)}$ with 4 categories, i.e. the feature space is $\mathcal{X} = \{1, 2, 3, 4\}$ and that class $Y$ satisfies $Y = 1$ if and only if $X^{(1)} = 1$ or $X^{(1)} = 3$. The optimal sparse DT in this case consists of only one split splitting the root $\Omega$ with respect to feature $X^{(1)}$, as shown in Figure 3. In this setting, BRANCHES only needs one iteration to terminate. Indeed, on its first iteration, it expands $\Omega$, estimates $\mathcal{Q}(\Omega, \bar{a})$ and $\mathcal{Q}(\Omega, a)$ where $a$ is the split action with respect to $X^{(1)}$. In this case, BRANCHES can already

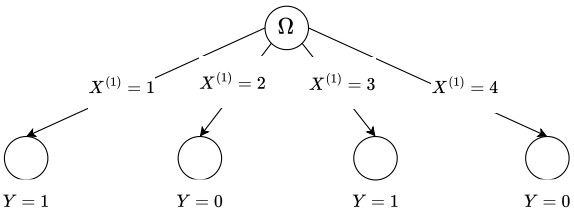

*Figure 3.* Optimal DT depicting the class variable that satisfies $Y = 1$ if and only if $X^{(1)} = 1$ or $X^{(1)} = 3$ on the space $\mathcal{X} = \{1, 2, 3, 4\}$.

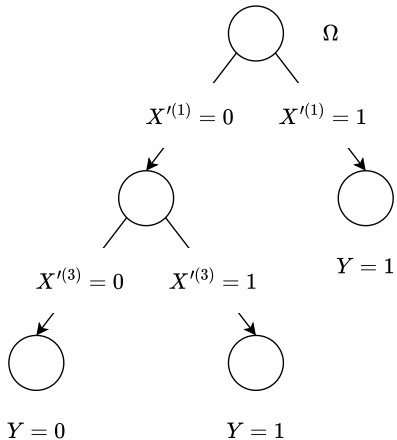

*Figure 4.* The new optimal sparse DT on the new feature space $\mathcal{X}'$.

deduce that:

$$\mathcal{Q}^* (\Omega, a) = \mathcal{Q} (\Omega, a) = -\lambda + \underbrace{\mathbb{P} [\Omega (X) = 1]}_{=1} > \mathbb{P} [\Omega (X) = 1, k^* (\Omega) = Y] = \mathcal{Q}^* (\Omega, \overline{a})$$

and therefore that $\Omega$ is SOLVED and $a = \text{Argmax}_{a' \in \mathcal{A}(\Omega)} \mathcal{Q}^* (\Omega, a')$.

Consider now the binary encoding of feature space $\mathcal{X}$, this yields a new feature space $\mathcal{X}' = \{0, 1\} \times \{0, 1\} \times \{0, 1\} \times \{0, 1\}$ where the new features $X'^{(1)}, X'^{(2)}, X'^{(3)}, X'^{(4)}$ describe the existence of a category or its absence:

$$\forall i \in \{1, 2, 3, 4\} : X'^{(i)} = \mathbb{I}\{X^{(1)} = i\}$$

Figure 4 depicts the new optimal sparse DT on $\mathcal{X}'$. In this setting, BRANCHES can no longer achieve optimality from the first iteration, because the first iteration only explores branches of size 1 and the optimal solution includes also branches of sizes 2. Moreover, Binary encoding introduces unnecessary branches that make the search space larger than necessary, thereby wasting some of the search time. To see this, consider the branch:

$$l' = \mathbb{I}\{X'^{(1)} = 1\} \wedge \mathbb{I}\{X'^{(2)} = 1\}$$

This branch exists in the new search space (search graph) constructed on $\mathcal{X}'$ and it could be explored at some point by the search algorithm. However, this would be a waste of time because $l'$ does not describe a possible subset of $\mathcal{X}$. Indeed, translating $l'$ to its corresponding branch on $\mathcal{X}$ yields:

$$l = \mathbb{I}\{X^{(1)} = 1\} \wedge \mathbb{I}\{X^{(1)} = 2\}$$

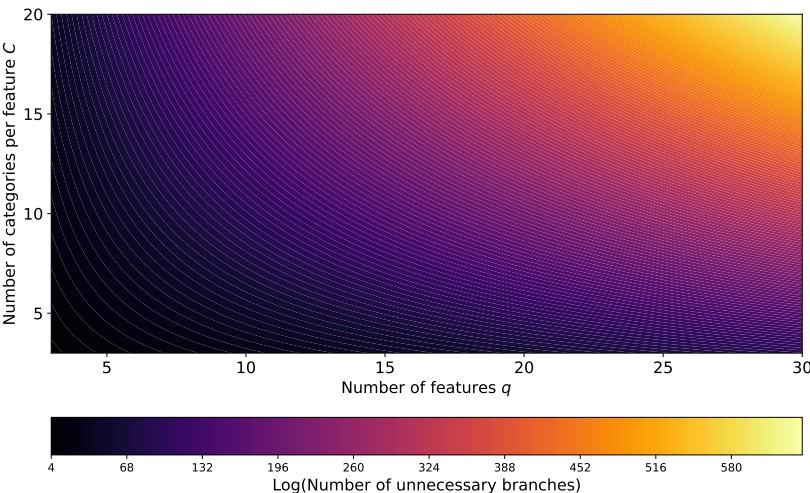

*Figure 5.* The number of unnecessary branches introduced by Binary Encoding.

which always valuates to 0 for any datum $X \in \mathcal{X}$. As a consequence, $l$ can never be part of the optimal solution, in fact, it can never be part of any Decision Tree on $\mathcal{X}$, $l$ is not even a proper branch by virtue of the definition in Section 3.1 as it violates condition (1). While it is true that the algorithm will prune this branch because its support set $\{X \in \mathcal{D} : l(X) = 1\}$ is null, it still has to waste time calculating this support set. To properly evaluate the computational inefficiency induced by binary encoding, we analyse the number of these introduced unnecessary branches. Theorem E.1 provides this analysis for the case where all features have an equal number categories.

**Theorem E.1.** *Consider a classification problem where all features share the same number of categories $C$, i.e. $\mathcal{X} = \{1, \ldots, C\}^q$ is the feature space. Performing a binary encoding on $\mathcal{X}$ yields the new feature space $\mathcal{X}' = \{0, 1\}^{qC}$. We define an unnecessary branch $l$ on $\mathcal{X}'$ as a branch that valuates to 0 for any input vector $X \in \mathcal{X}$:*

$$\forall X \in \mathcal{X} : l(X) = 0$$

*The number of unnecessary branches introduced by binary encoding is:*

$$\mathcal{U}(q, C) = \mathcal{A}(q, C) - \mathcal{B}(q, C) = 3^{qC} - [2C + 1]^q$$

*Proof.* The proof of this Theorem proceeds by counting the total number of branches possible on $\mathcal{X}'$ and subtracting the total number of branches that are not unnecessary.

Let us start with the total number of branches on $\mathcal{X}'$. Any branch on $\mathcal{X}'$ has the form:

$$l = \bigwedge_{v=1}^{w} \mathbb{I}\{X'^{(i_v)} = z_v\}$$

Where $X'^{(i_v)}$ are the features on the space $\mathcal{X}'$, $w \in \{0, \ldots, qC\}, i_v \in \{1, \ldots, qC\}, z_v \in \{0, 1\}$. We note that $w = 0$ corresponds to $l = \Omega$ by definition.

- There are $qC$ possibilities for choosing $w$.

- For each possible value $w$, there are $\binom{qC}{w}$ possible combinations $\{i_1, \ldots, i_w\}$.

- For each combination $\{i_1, \ldots, i_w\}$, there are $2^w$ possible assignments $(z_1, \ldots, z_w)$

Therefore the total number of branches on $\mathcal{X}'$ is:

$$\mathcal{A}(q, C) = \sum_{w=0}^{qC} \binom{qC}{w} 2^w = 3^{qC} \tag{17}$$

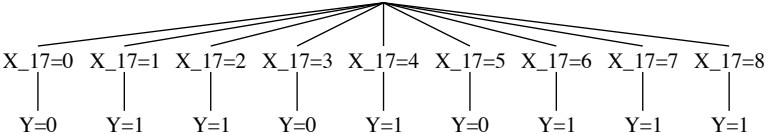

Figure 6. Optimal sparse DT for mushroom-o.

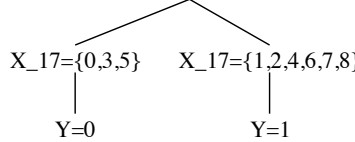

Figure 7. Collapsed optimal sparse DT for mushroom-o.

Let us now count the number of non-unnecessary branches. To do this, we consider a slightly different notation of the features on $\mathcal{X}'$.

$$\forall i \in \{1, \ldots, q\}, \forall j \in \{1, \ldots, C\} : X'^{(i,j)} = \mathbb{I}\{X^{(i)} = j\}$$

A branch $l = \bigwedge_{v=1}^{w} \mathbb{I}\{X'^{(i_v, j_v)} = z_v\}$ is not unnecessary if and only if $w \in \{0, \ldots, q\}, i_v \in \{1, \ldots, q\}, j_v \in \{1, \ldots, C\}, z_v \in \{0, 1\}$.

- For each possible value $w \in \{0, \ldots, q\}$, there are $\binom{q}{w}$ possible combinations $\{i_1, \ldots, i_w\}$.

- For each combination $\{i_1, \ldots, i_w\}$, there are $C^w$ possible assignments $(j_1, \ldots, j_w)$.

- For each assignment $(j_1, \ldots, j_w)$, there are $2^w$ possible assignments $(z_1, \ldots, z_w)$.

The total number of branches that are not unnecessary is therefore:

$$\mathcal{B}(q, C) = \sum_{w=0}^{q} \binom{q}{w} 2^w C^w = [2C + 1]^q \tag{18}$$

From Equation (17) and Equation (18) we deduce that the total number of unnecessary branches is:

$$\mathcal{U}(q, C) = \mathcal{A}(q, C) - \mathcal{B}(q, C) = 3^{qC} - [2C + 1]^q$$

$\square$

There is a subtlety here. We define $l$ on $\mathcal{X}'$, which means that it involves clauses defined with the features of $\mathcal{X}'$, and yet the definition in Theorem E.1 pertains to valuating $l$ on inputs from the feature space $\mathcal{X}$. There is no mistake or lack of rigour in this definition, we are allowed to do this because Binary Encoding is an injective map from $\mathcal{X}$ to $\mathcal{X}'$, thus implicitly, valuating $l$ on an input $X \in \mathcal{X}$ is defined as valuating $l$ on the image of $X$ in $\mathcal{X}'$ with this map.

Figure 5 is a contour plot of the number of unnecessary branches as a function of $q$ and $C$ in Logarithmic scale. It shows how immense this number becomes as $q$ and $C$ increase. We should note that, not all of these unnecessary branches will be explored by BRANCHES, in fact many of them (depending on the problem) will be pruned pre-emptively. Nevertheless, many will inevitably be visited, which hinders the search efficiency as we clearly demonstrated in Table 2. This inefficiency is most apparent on the mushroom dataset, where GOSDT and BRANCHES reach timeout on the binary encoding. However, in contrast, when applied to the ordinal encoding of mushroom, BRANCHES achieves an extremely fast optimal convergence in only $0.16s$ and 6 iterations. Moreover, in the mushroom dataset case, the optimal sparse DT only involves one split as depicted in Figure 6. This solution can further be collapsed into a single split DT with only two leaves as shown in Figure 7, which is a highly interpretable solution. On the other hand, binary encoding does not allow for such flexibility and its solution, depicted in Figure 8, is clearly less interpretable than the one in Figure 7.

When using binary encoding, it is usually a good idea to drop one category per feature. The reason pertains to reducing the number of the resulting binary features, which in turn makes it easier for the algorithm to quickly find the optimal

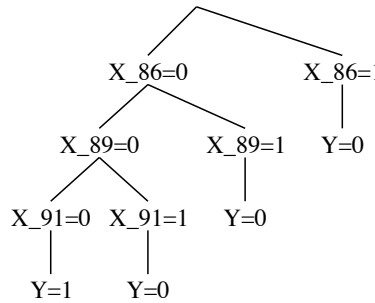

*Figure 8.* Optimal sparse DT for mushroom.

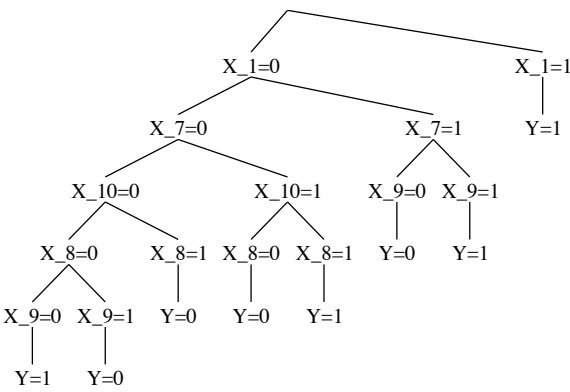

*Figure 9.* Optimal sparse DT for monk1-l, it has 7 splits.

sparse DT. Notice in Table 2 the difference in execution times between dropping and not dropping a category. In some cases like tic-tac-toe/tic-tac-toe-f, nursery/nursery-f and balance/balance-f, just dropping the first category pushes BRANCHES to terminate in time while it reaches timeout otherwise. However, choosing the adequate category to drop is not trivial and can lead to widely varying solutions and difficulties. We illustrate this point with monk1-l and monk1-f. Figure 9 and Figure 10 show very different optimal sparse DTs, with the option of dropping the last category leading to a solution with only 7 splits while dropping the first category induces a solution with 17 splits. Obviously in this case we prefer dropping the last category, but there is no trivial way of knowing this beforehand. On the other hand, employing a direct ordinal encoding yields the optimal solution in Figure 11 with 10 splits. By noticing sibling branches that share the same sub-DTs rooted in them, this solution can be collapsed into the DT in Figure 12 reducing its number of splits to 7. In this case, ordinal encoding allowed us to achieve a solution of similar quality to the one induced by monk1-l without the need to guess an adequate category to drop.

To conclude, we demonstrated in this section the reason why finding an optimal sparse DT is significantly faster via ordinal encoding than binary encoding. This discrepancy stems from the large amount of unnecessary branches that binary encoding introduces, as explained in Theorem E.1. Furthermore, we showed that these faster to get solutions can be even more desirable from an interpretability standpoint, we showcased this for the mushroom dataset. Using the monk1 dataset example, we explained the notion of collapsing the non-binary DT solution to yield better interpretable alternatives. This notion of collapse should be further investigated in the future. It could be of great use from a scalability standpoint. Indeed, for large datasets, using a preliminary binary encoding can greatly amplify the difficulty of the already challenging optimisation task. The idea is to rather utilise ordinal encoding to quickly find a non-binary DT solution, and then transform this solution, through some form of collapse, into a more interpretable version, maybe even a binary DT depending on how we conduct the collapsing process.

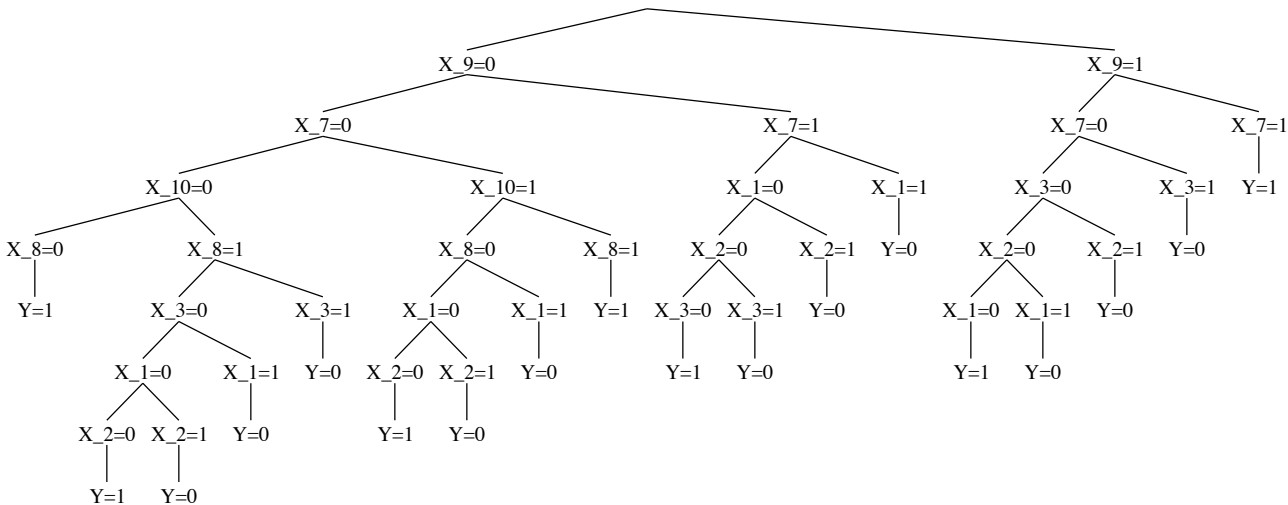

*Figure 10.* Optimal sparse DT for monk1-f, it has 17 splits.

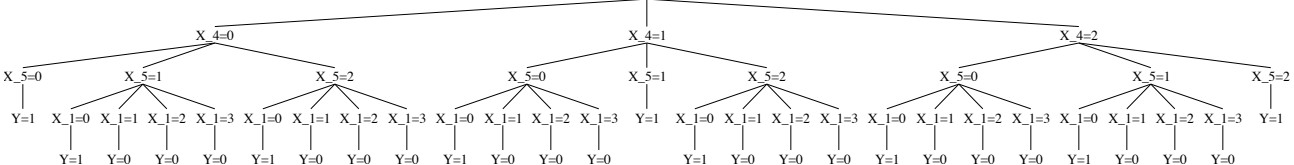

*Figure 11.* Optimal sparse DT for monk1-o, it has 10 splits.

## F. BRANCHES vs GOSDT

Among the state of the art algorithms, GOSDT is the closest to BRANCHES. Indeed, MurTree and STreeD are DFS methods unlike OSDT, GOSDT and BRANCHES. Moreover OSDT operates on the space of DTs instead of the space of branches unlike GOSDT and BRANCHES. For this reason, we dedicate this section to discussing the differences between BRANCHES and GOSDT.

**Support for ordinal encoding:** The first straightforward difference is BRANCHES' support for non-binary DTs that stem from ordinal encoding. In contrast, GOSDT (as well as all the algorithms we compare with) is only applicable to binary features, which necessitates a preliminary binary encoding. As we see in Section 5 and Appendix H.4, this detail is crucial, it confers BRANCHES' more scalability potential as these non-binary DT solutions are always found significantly faster (in the span of few seconds). In Appendix E we explained the drawbacks of binary encoding from a theoretical standpoint.

**The Purification Bound:** It is true that (Lin et al., 2020, bounds (9) and (10)) and the purification bound stem from the same reasoning, which we call purification when explaining the intuition behind Proposition 4.2. Furthermore (Lin et al., 2020, bounds (9) and (10)) support equivalent points, a situation that arises when the dataset includes duplicate instances with different classes, while our bound does not currently support this. However, this is straightforward to incorporate in the purification bound and it will be in a future version of BRANCHES. Moreover, we emphasize that while (Lin et al., 2020, bounds (9) and (10)) and the purification bound stem from a similar reasoning, they are different. The purification bound pertains to penalising the number of splits while (Lin et al., 2020, bounds (9) and (10)) rather correspond to penalising the number of leaves. This could lead to differences when considering non-binary features. Moreover, (Lin et al., 2020, bounds (9) and (10)) are formulated for a binary classification setting with binary features. On the other hand, the purification bound is formulated for the general case of multiclass classification with categorical features that are not necessarily binary. We further prove in Proposition 4.2 that the purification bound upper bounds the true optimal values $\mathcal{Q}^*(l, a)$ and $\mathcal{V}^*(l)$ when it is initialised and also when it is updated during Backpropagation.

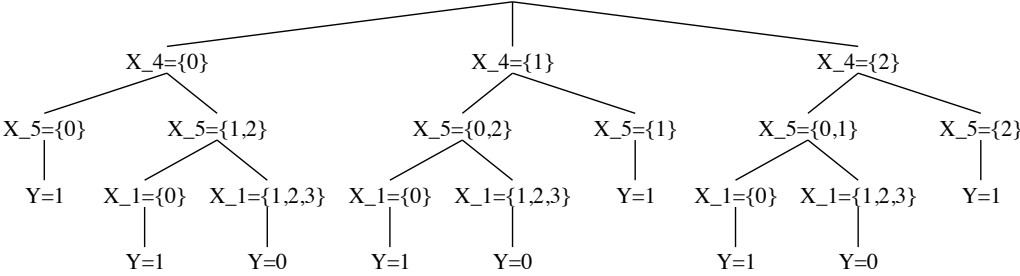

*Figure 12.* Collapsed optimal sparse DT for monk1-o, it has 7 splits similarly to the solution induced for monk1-l.

**The Selection process:** By Selection process, we refer to the strategy of choosing the branch to explore, by expanding it into children branches and then updating its value estimate (or bound). For BRANCHES, this is performed within our AO\* framework where we start from $\Omega$ and follow the action with highest value estimate $\text{Argmax}_{a \in \mathcal{A}(l)} \mathcal{Q}(l, a)$ until reaching a branch that is either SOLVED or that has not been expanded yet. This process is achieved through the use of multiple small priority queues at the level of each branch to keep track of $\text{Argmax}_{a \in \mathcal{A}(l)} \mathcal{Q}(l, a)$. In contrast, GOSDT employs one global priority queue where all the encountered branches are stored. Using this queue, GOSDT selects the most promising branch, i.e. the branch with the best (lowest in GOSDT's setting) bound. However, this branch would not necessarily be the most promising one for BRANCHES because it does not necessarily lie in BRANCHES' selection path. Indeed, the branch with highest $\mathcal{V}(l)$ does not necessarily lie in the path following $\text{Argmax}_{a \in \mathcal{A}(l)} \mathcal{Q}(l, a)$ from $\Omega$, in the AO\* terms of (Nilsson, 2014, Chapter 3), the branch chosen by GOSDT does not necessarily even lie in BRANCHES' current best partial solution graph. Due to the use of this one global priority queue, GOSDT resembles more an OR search than an AND/OR search algorithm. This difference in the selection strategies explain the large difference in the number of iterations of BRANCHES and GOSDT, which is in favour of BRANCHES. In addition to this fundamental difference, our Selection process allows us to incorporate heuristics for adequately choosing an UNSOLVED child, as explained in Appendix B. This flexibility has the potential of improving the anytime behaviour as explained in Appendix C, and indeed our experiments in Section 5 and Appendix H.4 show BRANCHES' anytime behaviour to always outperform GOSDT's. We emphasize here that these properties are provided due to the AND/OR nature of BRANCHES.

**The Backpropagation process:** BRANCHES updates value estimates along the selection path only. It is true that multiple paths lead to the same selected branch $l$, and as such the branches along these paths could also be updated in principle. However, the number of these paths is equal to the number of permutations of the clauses of $l$, which quickly becomes computationally costly. Moreover, many of these branches will be pre-emptively pruned and thus updating them would be a waste of time. Those that are not pruned will inevitably be visited by BRANCHES, and only then do we update them. In contrast, GOSDT updates the values along all the ancestors of the selected branch. This idea of backpropagating along the selection path only is discussed in (Nilsson, 2014, p. 106):

> *Therefore, not all ancestors need have cost revisions, but only those ancestors having best partial solution graphs containing descendants with revised costs (hence step 12).*

In addition to this difference, we explained in Appendix D that BRANCHES further sorts split actions via their `value_complete`. As a consequence, if many split actions share the value $\text{Argmax}_{a \in \mathcal{A}(l) \setminus \{\bar{a}\}} \mathcal{Q}(l, a)$, then the action $a$ for which $\text{Ch}(l, a)$ are all SOLVED (when it exists) will always be prioritised, thus deducing immediately that $l$ is SOLVED and avoiding wasting time exploring it again in the future.

**Caching procedure:** BRANCHES uses branch caching while GOSDT uses dataset caching or support set caching. We could implement this caching procedure in the future for BRANCHES, however, we have some doubts about its soundness when it comes to the sparsity problem, or even for optimisation with a hard constraint on depth. Indeed, two branches $l_1$ and $l_2$ could share the same support set (subset of the dataset that they contain) but differ in their number of splits $\mathcal{S}(l_1) < \mathcal{S}(l_2)$. In this case, the support set is insufficient to fully grasp the optimisation problem from each branch. From $l_1$ we have more splits to use before reaching the maximum depth than from $l_2$. Using dataset caching in this situation might lead to a suboptimal solution. This warrants further investigation in the future, in the absence of a proof of optimality (to our best knowledge) specific to dataset caching, we chose the conservative option of branch caching for BRANCHES.

**Computational complexity analysis:** In Section 4.2 we analysed the computational complexity of BRANCHES by bounding the number of branch evaluations it performs before terminating. Such analysis was not conducted for GOSDT. (Lin et al., 2020, Theorem H.1) provides a big-O bound on the total number of binary DTs that can be constructed from a set of $M$ binary features, but it does not specifically analyse GOSDT's computational complexity. To our knowledge, only (Hu et al., 2019, Theorem E.2) provides such analysis for OSDT, hence why we compared with it in Table 1.

**Empirical results:** The experiments we conduct in Section 5 and Appendix H.4 show that BRANCHES dominates GOSDT on the majority of experiments and for different metrics. From a lower execution time and significantly fewer iterations to a better anytime behaviour. This further supports the more efficient AND/OR search strategy of BRANCHES.

# G. Proofs and Additional Theoretical Results

**Proposition G.1.** *Let $0 < \lambda < 1$ and consider $T^* = \text{Argmax}_T \mathcal{H}_\lambda(T)$, then for any DT $T$ we have the following:*

$$\mathcal{S}(T) \leq \mathcal{S}(T^*) \implies \mathcal{H}(T) \leq \mathcal{H}(T^*)$$

*Proof.* Let $T$ be a DT and suppose that $\mathcal{S}(T) \leq \mathcal{S}(T^*)$. Since $T^* = \text{Argmax}_T \mathcal{H}_\lambda(T)$ we have:

$$\mathcal{H}_\lambda(T) \leq \mathcal{H}_\lambda(T^*)$$
$$\implies -\lambda \mathcal{S}(T) + \mathcal{H}(T) \leq -\lambda \mathcal{S}(T^*) + \mathcal{H}(T^*)$$
$$\implies 0 \leq \lambda(\mathcal{S}(T^*) - \mathcal{S}(T)) \leq \mathcal{H}(T^*) - \mathcal{H}(T)$$

$\square$

**Lemma G.2.** *Let $\pi$ be a policy and $l \in \mathcal{S} \setminus \mathcal{T}$ a non-terminal state. If $\pi(l) = \bar{a}$ then $T_{l,0}^\pi = \{l\}$ and $\forall t \geq 1 : T_{l,0}^\pi = \{\bar{l}\}$, otherwise we can define the trajectory $\left(T_{l,t}^\pi\right)_{t \geq 0}$ recursively as follows:*

$$\begin{cases} T_{l,0}^\pi &= \{l\} \\ T_{l,t}^\pi &= \bigcup_{l_u \in T_{l,1}^\pi} T_{l_u,t-1}^\pi \ \forall t \geq 1 \end{cases}$$

*Proof.* The case $\pi(l) = \bar{a}$ is trivial by the definition of the trajectory, thus we focus on the case $\pi(l) \neq \bar{a}$. By induction on $t \geq 1$, for $t = 1$ we have by definition $\forall l_u \in T_{l,1}^\pi : T_{l_u,0}^\pi = \{l_u\}$ and thus $T_{l,1}^\pi = \bigcup_{l_u \in T_{l,1}^\pi} T_{l_u,0}^\pi$.

Now suppose that the inductive hypothesis is true for some $t \geq 1$ and let us prove it for $t + 1$. We have:

$$T_{l,t+1}^\pi = \bigcup_{l' \in T_{l,t}^\pi \setminus \mathcal{T}} F(l', \pi(l')) \bigcup_{\bar{l}' \in T_{l,t}^\pi \cap \mathcal{T}} \{\bar{l}'\}$$

On the other hand, the inductive hypothesis states that:

$$T_{l,t}^\pi = \bigcup_{l_u \in T_{l,1}^\pi} T_{l_u,t-1}^\pi$$

Moreover, we know that $\left(T_{l_u,t-1}^\pi\right)_{l_u \in T_{l,1}^\pi}$ are mutually exclusive because they are sub-DTs rooted in mutually exclusive roots $l_u \in T_{l,1}^\pi$, hence $\left(T_{l_u,t-1}^\pi\right)_{l_u \in T_{l,1}^\pi}$ forms a partition of $T_{l,t}^\pi$ and we can rewrite $T_{l,t+1}^\pi$ as:

$$T_{l,t+1}^\pi = \bigcup_{l_u \in T_{l,1}^\pi} \left\{ \underbrace{\bigcup_{l' \in T_{l_u,t-1}^\pi} F(l', \pi(l')) \bigcup_{\bar{l}' \in T_{l_u,t-1}^\pi \cap \mathcal{T}} \{\bar{l}'\}}_{=T_{l_u,t}^\pi} \right\}$$

$$= \bigcup_{l_u \in T_{l,1}^\pi} T_{l_u,t}^\pi$$

Thus concluding the induction proof. $\square$

**Proposition 3.1.** *Let $\pi$ be a policy and $l \in \mathcal{S} \setminus \mathcal{T}$, then there exists a minimum $\tau_l^\pi \geq 1$ such that for any $t \geq \tau_l^\pi$, $T_{l,t}^\pi = \{\bar{l}_1, \ldots, \bar{l}_{|T_{\tau_l^\pi}^\pi|}\}$ is composed of terminal states only.*

*Proof.* We follow a proof by induction on $\mathcal{S}(l) \in \{0, \ldots, q\}$.

If $\mathcal{S}(l) = q$ :
Then $\mathcal{A}(l) = \{\bar{a}\}$ and $\pi(l) = \bar{a}$. Therefore $T_{l,0}^\pi = \{l\}$ and $T_{l,1}^\pi = F(l, \pi(l)) = F(l, \bar{a}) = \{\bar{l}\}$, and thus for $\tau_l^\pi = 1$ we have:
$$\forall t \geq \tau_l^\pi : T_{l,t}^\pi = \{\bar{l}\}$$

Now assume the inductive hypothesis to hold for some $1 \leq \mathcal{S}(l) = p \leq q$ and let us prove it for $\mathcal{S}(l) = p - 1$:
If $\pi(l) = \bar{a}$ then $T_{l,1}^\pi = \{\bar{l}\}$ and the result holds for $\tau_l^\pi = 1$. Otherwise when $\pi(l) \neq \bar{a}$, Lemma G.2 states that:
$$\forall t \geq 1 : T_{l,t}^\pi = \bigcup_{l_u \in T_{l,1}^\pi} T_{l_u, t-1}^\pi$$

On the other hand we know that $\forall l_u \in T_{l,1}^\pi : \mathcal{S}(l_u) = \mathcal{S}(l) + 1 = p$, thus the inductive hypothesis implies that:
$$\forall l_u \in T_{l,1}^\pi \; \exists \tau_{l_u}^\pi \geq 1 \; \forall t \geq \tau_{l_u}^\pi : T_{l_u,t}^\pi = \left\{ \bar{l}_1, \ldots, \bar{l}_{\left| T_{\tau_{l_u}^\pi}^\pi \right|} \right\} = T_{l_u, \tau_{l_u}^\pi}^\pi$$

Take $\tau_l^\pi = \max_{l_u \in T_{l,1}^\pi} \{\tau_{l_u}^\pi\} + 1$, then we get:
$$\forall t \geq \tau_l^\pi : T_{l,t}^\pi = \bigcup_{l_u \in T_{l,1}^\pi} T_{l_u, t-1}^\pi = \bigcup_{l_u \in T_{l,1}^\pi} T_{l_u, \tau_{l_u}^\pi}^\pi$$

Since $\forall l' \in T_{l_u, \tau_{l_u}^\pi}^\pi : l' \in \mathcal{T}$ then:
$$\forall t \geq \tau_l^\pi \; \forall l' \in T_{l,t}^\pi : l' \in \mathcal{T}$$

Thus concluding the induction proof. $\square$

**Proposition 3.2.** *Let $\pi$ be a policy and $l \in \mathcal{S} \setminus \mathcal{T}$ a non-terminal state, then $\mathcal{V}^\pi(l)$ satisfies the following:*
$$\mathcal{V}^\pi(l) = \mathcal{H}_\lambda(T_l^\pi) = -\lambda \mathcal{S}(T_l^\pi) + \mathcal{H}(T_l^\pi)$$
*Moreover, the optimal DT $T^*$ is the DT of an optimal policy $\pi^*$, in other words $T^* = T^{\pi^*}$.*

*Proof.* We proceed by induction on $\mathcal{S}(l) \in \{0, \ldots, q\}$.

If $\mathcal{S}(l) = q$:
Then we have $\pi(l) = \bar{a}$ and thus $\tau_l^\pi = 1$ and $T_l^\pi = \{l\}$. Therefore:
$$\mathcal{V}^\pi(l) = r(l, \pi(l)) = r(l, \bar{a}) = \mathcal{H}(l) = \mathcal{H}_\lambda(T_l^\pi)$$

The equality $\mathcal{H}(l) = \mathcal{H}_\lambda(T_l^\pi)$ stems from the fact that $T_l^\pi = \{l\}$ is the sub-DT rooted in $l$ that requires no splits of $l$.

Now let us assume the inductive hypothesis to hold true for some $1 \leq \mathcal{S}(l) = p \leq q$ and let us prove the result for $\mathcal{S}(l) = p - 1$:
If $\pi(l) = \bar{a}$, the result holds trivially with the same reasoning as the one we used for the case $\mathcal{S}(l) = q$.
Now consider the case $\pi(l) \neq \bar{a}$, we have the following:
$$\mathcal{V}^\pi(l) = \sum_{t=0}^{\tau_l^\pi - 1} \sum_{l_u \in T_{l,t}^\pi \setminus \mathcal{T}} r(l_u, \pi(l_u))$$
$$= r(l, \pi(l)) + \sum_{t=1}^{\tau_l^\pi - 1} \sum_{l_u \in T_{l,t}^\pi \setminus \mathcal{T}} r(l_u, \pi(l_u))$$
$$= -\lambda + \sum_{t=1}^{\tau_l^\pi - 1} \sum_{l_u \in T_{l,t}^\pi \setminus \mathcal{T}} r(l_u, \pi(l_u))$$

The last equality stems from the fact that $r\left(l, \pi\left(l\right)\right) = -\lambda$ because $\pi\left(l\right) \neq \bar{a}$ is a split action. Lemma G.2 states that $\left(T_{l_u,t-1}^{\pi}\right)_{l_u \in F(l,\pi(l))}$ is a partition of $T_{l,t}^{\pi}$ and therefore we can write:

$$\mathcal{V}^{\pi}\left(l\right) = -\lambda + \sum_{t=1}^{\tau_l^{\pi}-1} \sum_{l' \in F(l,\pi(l))} \sum_{l_u \in T_{l',t-1}^{\pi} \setminus \mathcal{T}} r\left(l_u, \pi\left(l_u\right)\right)$$

$$= -\lambda + \sum_{l' \in F(l,\pi(l))} \sum_{t=1}^{\tau_l^{\pi}-1} \sum_{l_u \in T_{l',t-1}^{\pi} \setminus \mathcal{T}} r\left(l_u, \pi\left(l_u\right)\right)$$

We know that $\forall l' \in T_{l,1}^{\pi} : \tau_{l'}^{\pi} \leq \tau_l^{\pi} - 1$ and thus:

$$\mathcal{V}^{\pi}\left(l\right) = -\lambda + \sum_{l' \in F(l,\pi(l))} \sum_{t=0}^{\tau_{l'}^{\pi}-1} \sum_{l_u \in T_{l',t}^{\pi} \setminus \mathcal{T}} r\left(l_u, \pi\left(l_u\right)\right)$$

$$= -\lambda + \sum_{l' \in F(l,\pi(l))} \mathcal{V}^{\pi}\left(l'\right)$$

On the other hand we know that $\forall l' \in F\left(l, \pi\left(l\right)\right) : \mathcal{S}\left(l'\right) = \mathcal{S}\left(l\right) + 1 = p$, therefore the inductive hypothesis implies:

$$\forall l' \in F\left(l, \pi\left(l\right)\right) : \mathcal{V}^{\pi}\left(l'\right) = \mathcal{H}_{\lambda}\left(T_{l'}^{\pi}\right)$$

Going back to $\mathcal{V}^{\pi}\left(l\right)$ induces:

$$\mathcal{V}^{\pi}\left(l\right) = -\lambda + \sum_{l' \in F(l,\pi(l))} \mathcal{H}_{\lambda}\left(T_{l'}^{\pi}\right)$$

$$= -\lambda + \sum_{l' \in F(l,\pi(l))} \left\{-\lambda \mathcal{S}\left(T_{l'}^{\pi}\right)\right\} + \sum_{l' \in F(l,\pi(l))} \mathcal{H}\left(T_{l'}^{\pi}\right)$$

$$= -\lambda \left\{1 + \sum_{l' \in F(l,\pi(l))} \mathcal{S}\left(T_{l'}^{\pi}\right)\right\} + \sum_{l' \in F(l,\pi(l))} \mathcal{H}\left(T_{l'}^{\pi}\right)$$

The sub-DT $T_l^{\pi}$ is constituted of the sub-DTs $T_{l'}^{\pi}$ that are each rooted in $l' \in F\left(l, \pi\left(l\right)\right)$, thus the number of splits of $T_l^{\pi}$ is equal to 1 (the first splits at $l$) plus the sum of the numbers of splits of $T_{l'}^{\pi}$ for $l' \in F\left(l, \pi\left(l\right)\right)$, i.e.

$$\mathcal{S}\left(T_l^{\pi}\right) = 1 + \sum_{l' \in F(l,\pi(l))} \mathcal{S}\left(T_{l'}^{\pi}\right)$$

Moreover $(T_{l'}^{\pi})_{l' \in F(l,\pi(l))}$ is a partition of $T_l^{\pi}$ and thus:

$$\mathcal{H}\left(T_l^{\pi}\right) = \sum_{l' \in F(l,\pi(l))} \mathcal{H}\left(T_{l'}^{\pi}\right)$$

Going back to $\mathcal{V}^{\pi}\left(l\right)$ we get:

$$\mathcal{V}^{\pi}\left(l\right) = -\lambda \mathcal{S}\left(T_l^{\pi}\right) + \mathcal{H}\left(T_l^{\pi}\right) = \mathcal{H}_{\lambda}\left(T_l^{\pi}\right)$$

This concludes the induction proof. $\qquad\square$

**Lemma G.3.** *Let $\pi^*$ be an optimal policy, i.e. $\pi^* \in \mathrm{Argmax}_{\pi} \mathcal{V}^{\pi}\left(\Omega\right)$ and consider the set of non-terminal states in its trajectory from $\Omega$:*

$$\mathcal{S}^* = \bigcup_{t=0}^{\tau_{\Omega}^*-1} T_{\Omega,t}^* \setminus \mathcal{T}$$

*Then for any policy $\pi$ we have the following:*

$$\forall l \in \mathcal{S}^* : \mathcal{V}^{\pi}\left(l\right) = \mathcal{V}^*\left(l\right)$$

*Proof.* We follow a proof by contradiction procedure. Suppose that there exists a policy $\pi$ such that:

$$\exists l \in \mathcal{S}^* : \mathcal{V}^\pi (l) > \mathcal{V}^* (l)$$

Since $l \in \mathcal{S}^*$, there exists $1 \leq t_l \leq \tau_l^* - 1$ such that $l \in T_{\Omega,t_l}^*$. We write:

$$\mathcal{V}^* (\Omega) = \sum_{t=0}^{t_l-1} \sum_{l_u \in T_{\Omega,t}^* \setminus \mathcal{T}} r (l_u, \pi^* (l_u)) + \sum_{l' \in T_{\Omega,t_l}^* \setminus \mathcal{T} \setminus \{l\}} \mathcal{V}^* (l') + \mathcal{V}^* (l) \tag{19}$$

Now we define a new policy $\pi'$ as follows:

$$\begin{cases} \pi' = \pi \text{ on } l \text{ and all of its descendents.} \\ \pi' = \pi^* \text{ elsewhere.} \end{cases}$$

The first term in (19) is therefore equal to:

$$\sum_{t=0}^{t_l-1} \sum_{l_u \in T_{\Omega,t}^* \setminus \mathcal{T}} r (l_u, \pi^* (l_u)) = \sum_{t=0}^{t_l-1} \sum_{l_u \in T_{\Omega,t}^* \setminus \mathcal{T}} r (l_u, \pi' (l_u))$$

Let us now analyse the second term $\sum_{l' \in T_{\Omega,t_l}^* \setminus \mathcal{T} \setminus \{l\}} \mathcal{V}^* (l')$. Since $l \in T_{\Omega,t_l}^*$, then for all $l' \in T_{\Omega,t_l}^* \setminus \mathcal{T} \setminus \{l\}$, $l$ and $l'$ share no descendents. Therefore, for any descendent $l"$ of $l'$ we have $\pi' (l') = \pi^* (l')$ and therefore:

$$\forall l' \in T_{\Omega,t_l}^* \setminus \mathcal{T} \setminus \{l\} : \mathcal{V}^* (l') = \mathcal{V}^{\pi'} (l')$$
$$\implies \sum_{l' \in T_{\Omega,t_l}^* \setminus \mathcal{T} \setminus \{l\}} \mathcal{V}^* (l') = \sum_{l' \in T_{\Omega,t_l}^* \setminus \mathcal{T} \setminus \{l\}} \mathcal{V}^{\pi'} (l')$$

For the third term in (19), since $\pi' = \pi$ on $l$ and all of its descendents, then we have $\mathcal{V}^{\pi'} (l) = \mathcal{V}^\pi (l) > \mathcal{V}^* (l)$. We can now rewrite (19) as follows:

$$\mathcal{V}^* (\Omega) < \underbrace{\sum_{t=0}^{t_l-1} \sum_{l_u \in T_{\Omega,t}^* \setminus \mathcal{T}} r (l_u, \pi' (l_u)) + \sum_{l' \in T_{\Omega,t_l}^* \setminus \mathcal{T} \setminus \{l\}} \mathcal{V}^{\pi'} (l') + \mathcal{V}^{\pi'} (l')}_{=\mathcal{V}^{\pi'} (\Omega)}$$
$$< \mathcal{V}^{\pi'} (\Omega)$$

This contradicts the fact that $\pi^*$ is optimal, which concludes our proof. $\qquad\square$

**Proposition 4.1** (Bellman Optimality Equations). *Let $\pi^*$ be an optimal policy, i.e. $\pi^* \in \mathrm{Argmax}_\pi \mathcal{V}^\pi (\Omega)$ and consider the set of non-terminal states in its trajectory from $\Omega$:*

$$\mathcal{S}^* = \bigcup_{t=0}^{\tau_\Omega^*-1} T_{\Omega,t}^* \setminus \mathcal{T}$$

*Now consider a policy $\pi$ and suppose that for all $l \in \mathcal{S}^*$:*

$$\mathcal{V}^\pi (l) = \max_{a \in \mathcal{A}(l)} \mathcal{Q}^\pi (l, a) \tag{8}$$

*Then $\pi$ is optimal and we also have:*

$$\pi (l) = \mathrm{Argmax}_{a \in \mathcal{A}(l)} \mathcal{Q}^\pi (l, a) \tag{9}$$

*Proof.* We know that $\forall l \in \mathcal{S}^* \ \exists 0 \leq t_l \leq \tau_\Omega^* : l \in T_{\Omega,t_l}^* \setminus \mathcal{T}$. We will now show that $\forall l \in \mathcal{S}^* : \mathcal{V}^\pi (l) = \mathcal{V}^* (l)$ which would include $\mathcal{V}^\pi (\Omega) = \mathcal{V}^* (\Omega)$. The proof proceeds by backward induction on $t_l$.

For $t_l = \tau_\Omega^* - 1$, by definition we have:

$$\mathcal{V}^\pi(l) = \max_{a \in \mathcal{A}(l)} \mathcal{Q}^\pi(l,a) \geq \mathcal{Q}^\pi(l,\overline{a}) = \mathcal{H}(l) = \mathcal{Q}^*(l,\overline{a}) \tag{20}$$

On the other hand, we know that $\pi^*(l) = \overline{a}$. Indeed, this is true because otherwise $\pi^*(l)$ would be a split action leading to non-terminal states, which means that $T_{\Omega,t_l+1}^* = T_{\Omega,\tau_\Omega^*}^{\pi^*}$ includes non-terminal states, and this would contradict the definition of $\tau_\Omega^*$. Therefore:

$$\mathcal{V}^*(l) = \mathcal{Q}^*(l,\pi(l)) = \mathcal{Q}^*(l,\overline{a}) \tag{21}$$

From Equation (20) and Equation (21) we get:

$$\mathcal{V}^\pi(l) \geq \mathcal{V}^*(l) \tag{22}$$

On the other hand, since $\pi^*$ is optimal and $l \in \mathcal{S}^*$ we have $\mathcal{V}^\pi(l) \leq \mathcal{V}^{\pi^*}(l)$ according to Lemma G.3, and we deduce that:

$$\mathcal{V}^\pi(l) = \mathcal{V}^*(l)$$

Now suppose that $\mathcal{V}^\pi(l) = \mathcal{V}^*(l)$ for all $l \in \mathcal{S}^*$ such that $t_l = t$ for some $1 \leq t \leq \tau_\Omega^* - 1$ and let us show that the result still holds for $t - 1$. Let $l \in \mathcal{S}^*$ such that $t_l = t - 1$, we have $\mathcal{V}^\pi(l) = \max_{a \in \mathcal{A}(l)} \mathcal{Q}^\pi(l,a)$. On the other hand:

$$\mathcal{Q}^\pi(l,\pi^*(l)) = r(l,\pi^*(l)) + \sum_{l_u \in F(l,\pi^*(l)) \setminus \mathcal{T}} \mathcal{V}^\pi(l_u) \tag{23}$$

Moreover $\forall l_u \in F(l,\pi^*(l)) \setminus \mathcal{T} : l_u \in \mathcal{S}^*, t_{l_u} = t$. The induction hypothesis then implies that:

$$\forall l_u \in F(l,\pi^*(l)) \setminus \mathcal{T} : \mathcal{V}^\pi(l_u) = \mathcal{V}^*(l_u)$$

Going back to Equation (23), we get:

$$\mathcal{Q}^\pi(l,\pi^*(l)) = r(l,\pi^*(l)) + \sum_{l_u \in F(l,\pi^*(l)) \setminus \mathcal{T}} \mathcal{V}^*(l_u) = \mathcal{Q}^*(l,\pi^*(l)) = \mathcal{V}^*(l)$$

Now we have:

$$\mathcal{V}^\pi(l) = \max_{a \in \mathcal{A}(l)} \mathcal{Q}^\pi(l,a) \geq \mathcal{Q}^\pi(l,\pi^*(l)) = \mathcal{V}^*(l)$$

On the other hand $\mathcal{V}^\pi(l) \leq \mathcal{V}^*(l)$ because $\pi^*$ is optimal. Therefore we deduce that:

$$\mathcal{V}^\pi(l) = \mathcal{V}^*(l)$$

which concludes the inductive proof. $\qquad\square$

**Proposition 4.2** (Purification Bound). *Let $l \in \mathcal{S} \setminus \mathcal{T}$, we define the Purification Bound as follows:*

*If $\mathcal{A}(l) \setminus \{\overline{a}\} \neq \emptyset$:*

$$\mathcal{V}(l) = \max\{\mathcal{H}(l), -\lambda + \mathbb{P}[l(X) = 1]\}$$
$$= \max\left\{\frac{n_{k^*(l)}(l)}{n}, -\lambda + \frac{n(l)}{n}\right\} \tag{13}$$

*Otherwise:*

$$\mathcal{V}(l) = \mathcal{V}^*(l) = \mathcal{H}(l) = \frac{n_{k^*(l)}(l)}{n} \tag{14}$$

*The bounds $\mathcal{V}(l)$ are initialised with (13) or (14), then they are recursively backpropagated to the ancestors of $l$ in the AND/OR graph through (11) and (12). The resulting heuristic estimates $\mathcal{Q}(l,a)$ and $\mathcal{V}(l)$ are upper bounds on the true optimal values $\mathcal{Q}^*(l,a)$ and $\mathcal{V}^*(l)$ respectively.*

*Proof.* Let us first show that the initialisations in Equation (13) and Equation (14) are upper bounds on the true optimal values $\mathcal{V}^*(l)$:

The case $\mathcal{A}(l) = \{\overline{a}\}$ is trivial because the only action that can be taken at $l$ is $\overline{a}$, which means that all policies $\pi$ map $l$ to $\pi(l) = \overline{a}$, therefore:

$$\forall \pi : \mathcal{V}^\pi(l) = r(l, \overline{a}) = \mathcal{H}(l)$$

Now consider the case $\mathcal{A}(l) \setminus \{\overline{a}\} \neq \emptyset$, we have the following:

$$\mathcal{V}(l) = \max\{\mathcal{H}(l), -\lambda + \mathbb{P}[l(X) = 1]\}$$
$$= \max\{\mathcal{Q}^*(l, \overline{a}), -\lambda + \mathbb{P}[l(X) = 1]\}$$

It suffices to show now that:

$$\forall a \in \mathcal{A}(l) \setminus \{\overline{a}\} : \mathcal{Q}^*(l, a) \leq -\lambda + \mathbb{P}[l(X) = 1]$$

Let $a \in \mathcal{A}(l) \setminus \{\overline{a}\}$ be a split action, we have the following:

$$\mathcal{Q}^*(l, a) = r(l, a) + \sum_{l_u \in F(l,a) \setminus \mathcal{T}} \mathcal{V}^*(l_u)$$
$$= -\lambda + \sum_{l_u \in F(l,a)} \mathcal{V}^*(l_u) \tag{24}$$

On the other hand, Proposition 3.2 implies that:

$$\forall l_u \in F(l, a) : \mathcal{V}^*(l_u) = \mathcal{H}_\lambda(T^*_{l_u}) = -\lambda \mathcal{S}(T^*_{l_u}) + \mathcal{H}(T^*_{l_u})$$
$$\leq \mathbb{P}[l_u(X) = 1, T^*_{l_u}(X) = Y]$$
$$\leq \mathbb{P}[l_u(X) = 1]$$

The second line stems from the definition of $\mathcal{H}(T^*_{l_u})$ and the fact that $\mathcal{S}(T^*_{l_u}) \geq 0$. Going back to Equation (24) yields:

$$\mathcal{Q}^*(l, a) \leq -\lambda + \sum_{l_u \in F(l,a)} \mathbb{P}[l_u(X) = 1]$$

Since $F(l, a)$ is a partition of $l$ we know that $\mathbb{P}[l(X) = 1] = \sum_{l_u \in F(l,a)} \mathbb{P}[l_u(X) = 1]$ and thus:

$$\mathcal{Q}^*(l, a) \leq -\lambda + \mathbb{P}[l(X) = 1]$$

We deduce that:

$$\mathcal{V}(l) = \max\{\mathcal{Q}^*(l, \overline{a}), -\lambda + \mathbb{P}[l(X) = 1]\}$$
$$\geq \max_{a \in \mathcal{A}(l)} \mathcal{Q}^*(l, a) \geq \mathcal{Q}^*(l, \pi^*(l)) = \mathcal{V}^*(l)$$

Now let $l \in \mathcal{S} \setminus \mathcal{T}$ and suppose that for all its children, the upper bounds $\mathcal{V}(l_u)$ are available. For all split actions $a \in \mathcal{A}(l) \setminus \{\overline{a}\}$, the definition (11) implies that:

$$\mathcal{Q}(l, a) = -\lambda + \sum_{l_u \in F(l,a)} \mathcal{V}(l_u)$$

Since $\forall l_u \in F(l, a) : \mathcal{V}(l_u) \geq \mathcal{V}^*(l_u)$, we get:

$$\mathcal{Q}(l, a) \geq -\lambda + \sum_{l_u \in F(l,a)} \mathcal{V}^*(l_u) = \mathcal{Q}^*(l, a)$$

On the other hand, the definition (12) implies that:

$$\mathcal{V}(l) = \max_{a \in \mathcal{A}(l)} \mathcal{Q}(l, a)$$
$$\geq \max_{a \in \mathcal{A}(l)} \mathcal{Q}^*(l, a)$$
$$\geq \mathcal{Q}^*(l, \pi^*(l))$$
$$\geq \mathcal{V}^*(l)$$

which concludes our proof. $\qquad\square$

**Theorem 4.3** (Optimality). *Upon termination, the selection policy $\widetilde{\pi}$ becomes optimal. In other words:*

$$\mathcal{V}^{\widetilde{\pi}}\left(\Omega\right) = \mathcal{V}^*\left(\Omega\right) = \max_{\pi} \mathcal{V}^{\pi}\left(\Omega\right)$$

*Proof.* To prove this we show that for any $l \in \mathcal{S} \setminus \mathcal{T}$, $l$ is SOLVED if and only if:

$$\mathcal{V}^{\widetilde{\pi}}\left(l\right) = \mathcal{V}\left(l\right) \geq \mathcal{V}^*\left(l\right)$$

where $\mathcal{V}^*$ is the value function of an optimal policy $\pi^*$. We proceed by induction on $1 \leq \tau_l^{\widetilde{\pi}} \leq q$:

If $\tau_l^{\widetilde{\pi}} = 1$, then we have:

$$\widetilde{\pi}\left(l\right) = \bar{a} = \text{Argmax}_{a \in \mathcal{A}(l)} \mathcal{Q}\left(l, a\right) \tag{25}$$

On the other hand:

$$\mathcal{V}^{\widetilde{\pi}}\left(l\right) = \mathcal{Q}^{\widetilde{\pi}}\left(l, \widetilde{\pi}\left(l\right)\right) = \mathcal{Q}^{\widetilde{\pi}}\left(l, \bar{a}\right) = \mathcal{Q}\left(l, \bar{a}\right) \tag{26}$$

From Equation (25) and Equation (26) we deduce that:

$$\mathcal{V}^{\widetilde{\pi}}\left(l\right) = \max_{a \in \mathcal{A}(l)} \mathcal{Q}\left(l, a\right) = \mathcal{V}\left(l\right) \geq \mathcal{V}^*\left(l\right)$$

The last inequality is due to Proposition 4.2. Note that we do not necessarily have $\mathcal{V}^{\widetilde{\pi}}\left(l\right) \leq \mathcal{V}^*\left(l\right)$ even though $\pi^*$ is an optimal policy. Indeed this would only be necessarily satisfied if $l \in \mathcal{S}^*$ as per Lemma G.3.

Now suppose that the result is true for $\tau_l^{\widetilde{\pi}} = t$ for some $1 \leq t \leq q - 1$ and let us prove it for any $l \in \mathcal{S} \setminus \mathcal{T}$ such that $\tau_l^{\widetilde{\pi}} = t + 1$. We have:

$$\begin{aligned} \mathcal{V}^{\widetilde{\pi}}\left(l\right) &= \mathcal{Q}^{\widetilde{\pi}}\left(l, \widetilde{\pi}\left(l\right)\right) \\ &= r\left(l, \widetilde{\pi}\left(l\right)\right) + \sum_{l_u \in F(l, \widetilde{\pi}(l))} \mathcal{V}^{\widetilde{\pi}}\left(l_u\right) \end{aligned}$$

Since $\tau_l^{\widetilde{\pi}} = t + 1 \geq 2$ then we necessarily have $\widetilde{\pi}\left(l\right) \in \mathcal{A}\left(l\right) \setminus \{\bar{a}\}$, i.e. $\widetilde{\pi}\left(l\right)$ is a split action. Indeed, this is true because otherwise we would have $\tau_l^{\widetilde{\pi}} = 1$. Therefore we get:

$$\mathcal{V}^{\widetilde{\pi}}\left(l\right) = -\lambda + \sum_{l_u \in F(l, \widetilde{\pi}(l))} \mathcal{V}^{\widetilde{\pi}}\left(l_u\right)$$

On the other hand $\forall l_u \in F\left(l, \widetilde{\pi}\left(l\right)\right) : \tau_{l_u}^{\widetilde{\pi}} = t$ and thus the inductive hypothesis implies that $\mathcal{V}^{\widetilde{\pi}}\left(l_u\right) = \mathcal{V}\left(l_u\right) \geq \mathcal{V}^*\left(l_u\right)$, which means that:

$$\begin{aligned} \mathcal{V}^{\widetilde{\pi}}\left(l\right) = -\lambda + \sum_{l_u \in F(l, \widetilde{\pi}(l))} \mathcal{V}\left(l_u\right) = \mathcal{Q}\left(l, \widetilde{\pi}\left(l\right)\right) &= \max_{a \in \mathcal{A}(l)} \mathcal{Q}\left(l, a\right) = \mathcal{V}\left(l\right) \\ &\geq \mathcal{Q}\left(l, \pi^*\left(l\right)\right) \\ &\geq \mathcal{Q}^*\left(l, \pi^*\left(l\right)\right) \\ &\geq \mathcal{V}^*\left(l\right) \end{aligned}$$

which concludes the inductive proof.

Now, since BRANCHES terminates when $\Omega$ becomes SOLVED, then we have:

$$\mathcal{V}^{\widetilde{\pi}}\left(\Omega\right) \geq \mathcal{V}^*\left(\Omega\right)$$

On the other hand, since $\Omega \in \mathcal{S}^*$, then by Lemma G.3, we necessarily have:

$$\mathcal{V}^{\widetilde{\pi}}\left(\Omega\right) \leq \mathcal{V}^*\left(\Omega\right)$$

Hence deducing that:

$$\mathcal{V}^{\widetilde{\pi}}\left(\Omega\right) = \mathcal{V}^*\left(\Omega\right)$$

$\square$

**Lemma G.4.** *A branch $l \in S \setminus \mathcal{T}$ can be chosen for Expansion only if there exists a DT $T$ such that:*

$$\begin{cases} l \in T \setminus L \\ -\lambda S(T) + \sum_{l' \in L} \mathcal{H}(l') + \sum_{l' \in T \setminus L} \left\{ -\lambda + \mathbb{P}\left[l'(X) = 1\right] \right\} \geq -\lambda S(T^*) + \mathcal{H}(T^*) \end{cases}$$

*Where $L = \{l' \in T : \mathcal{H}(l') \geq -\lambda + \mathbb{P}\left[l'(X) = 1\right]\}$.*

*Proof.* A branch $l \in S \setminus \mathcal{T}$ can only be chosen for Expansion if it is a tip node (leaf) of the search graph $G$ and is selected by the selection policy $\widetilde{\pi}$, i.e.

$$\exists t_l \geq 0 : l \in T_{\Omega, t_l}^{\widetilde{\pi}}$$

We know that there exists $\tau \geq 0$ such that for all $l \in T_{\Omega, \tau}^{\widetilde{\pi}}$, $l$ is a tip node of $G$. This is true because otherwise $G$ would be bottomless, which is false because it has a maximum depth of $q$ (the total number of features). On the other hand, the recursive definition of $\mathcal{V}$ yields:

$$\mathcal{V}(\Omega) = -\lambda S\left(T_{\Omega, \tau}^{\widetilde{\pi}}\right) + \sum_{l' \in T_{l, \tau}^{\widetilde{\pi}}} \mathcal{V}(l')$$

Since all branches $l' \in T_{\Omega, \tau}^{\widetilde{\pi}}$ are tip nodes of $G$, then $\mathcal{V}(l') = \mathcal{H}(l')$ if $\mathcal{H}(l') \geq -\lambda + \mathbb{P}\left[l'(X) = 1\right]$ and $\mathcal{V}(l') = -\lambda + \mathbb{P}\left[l'(X) = 1\right]$ otherwise. Define $L = \{l' \in T_{\Omega, \tau}^{\widetilde{\pi}} : \mathcal{H}(l') \geq -\lambda + \mathbb{P}\left[l'(X) = 1\right]\}$, we have:

$$\mathcal{V}(\Omega) = -\lambda S\left(T_{\Omega, \tau}^{\widetilde{\pi}}\right) + \sum_{l' \in L} \mathcal{H}(l') + \sum_{l' \in T_{\Omega, \tau}^{\widetilde{\pi}} \setminus L} \left\{ -\lambda + \mathbb{P}\left[l'(X) = 1\right] \right\}$$

On the other hand, according to Proposition 4.2 and Proposition 3.2 we have the following:

$$\mathcal{V}(\Omega) \geq \mathcal{V}^*(\Omega) = -\lambda S(T^*) + \mathcal{H}(T^*)$$

Thus:

$$-\lambda S\left(T_{\Omega, \tau}^{\widetilde{\pi}}\right) + \sum_{l' \in L} \mathcal{H}(l') + \sum_{l' \in T_{\Omega, \tau}^{\widetilde{\pi}} \setminus L} \left\{ -\lambda + \mathbb{P}\left[l'(X) = 1\right] \right\} \geq \lambda S(T^*) + \mathcal{H}(T^*)$$

Moreover, $l$ can only be expanded if $l \in T_{\Omega, \tau}^{\widetilde{\pi}}$ because all the branches in $L$ are SOLVED. This concludes our proof. $\qquad \square$

**Theorem G.5** (Problem-dependent complexity of BRANCHES). *Let $\Gamma(q, C, \lambda)$ denote the total number of branch evaluations performed by BRANCHES for an instance of the classification problem with $q \geq 2$ features, $0 < \lambda \leq 1$ the penalty parameter, and $C \geq 2$ the number of categories per feature. Then, $\Gamma(q, C, \lambda)$ satisfies the following bound:*

$$\Gamma(q, C, \lambda) \leq \sum_{h=0}^{\kappa} (q - h) C^{h+1} \binom{q}{h}; \quad \kappa = \min\left\{ \left\lfloor S(T^*) - 1 + \frac{1 - \mathcal{H}(T^*)}{\lambda} \right\rfloor, q \right\}$$

*Proof.* Let $l$ be a branch. According to Lemma G.4, for $l$ to be considered for Expansion, there has to exist a DT $T$ such that:

$$\begin{cases} l \in T \setminus L \\ -\lambda S(T) + \sum_{l' \in L} \mathcal{H}(l') + \sum_{l' \in T \setminus L} \left\{ -\lambda + \mathbb{P}\left[l'(X) = 1\right] \right\} \geq -\lambda S(T^*) + \mathcal{H}(T^*) \end{cases}$$

where $L = \{l' \in T : \mathcal{H}(l') \geq -\lambda + \mathbb{P}\left[l'(X) = 1\right]\}$. Suppose $l$ is such a branch, then we have:

$$-\lambda S(T) + \sum_{l' \in L} \underbrace{\mathcal{H}(l')}_{\leq \mathbb{P}[l'(X)=1]} + \sum_{l' \in T \setminus L} \left\{ -\lambda + \mathbb{P}\left[l'(X) = 1\right] \right\} \geq -\lambda S(T^*) + \mathcal{H}(T^*)$$

$$\implies -\lambda \left\{ S(T) + |T \setminus L| \right\} + \sum_{l' \in T} \mathbb{P}\left[l'(X) = 1\right] \geq -\lambda S(T^*) + \mathcal{H}(T^*)$$

Since $l \in T \setminus L$, then $|T \setminus L| \geq 1$ and we get:

$$-\lambda \left\{ \mathcal{S}\left(T\right) + 1 \right\} + 1 \geq -\lambda \mathcal{S}\left(T^*\right) + \mathcal{H}\left(T^*\right)$$

$$\implies \mathcal{S}\left(T\right) \leq \mathcal{S}\left(T^*\right) - 1 + \frac{1 - \mathcal{H}\left(T^*\right)}{\lambda}$$

$$\implies \mathcal{S}\left(l\right) \leq \mathcal{S}\left(T^*\right) - 1 + \frac{1 - \mathcal{H}\left(T^*\right)}{\lambda}$$

Let $\mathcal{C} = \left\{ l \text{ branch} : \mathcal{S}\left(l\right) \leq \mathcal{S}\left(T^*\right) - 1 + \frac{1 - \mathcal{H}\left(T^*\right)}{\lambda} \right\}$. Then the number of branches that are expanded is upper bounded by $|\mathcal{C}|$.

We recall that we rather seek to upper bound the number of branches that are evaluated, i.e. for which we calculate $\mathcal{H}\left(l\right)$. These evaluations happen during the Expansion step of BRANCHES. When a branch $l$ is expanded, we evaluate all of its children. There are $q - \mathcal{S}\left(l\right)$ features left to use for splitting $l$, and for each split, $C$ children branches are created. Thus, there are $(q - \mathcal{S}\left(l\right)) C$ children of $l$, hence $(q - \mathcal{S}\left(l\right)) C$ evaluations happen during the expansion of $l$. Let us now upper bound $\Gamma\left(q, C, \lambda\right)$.

For each branch $l \in \mathcal{C}$:

- We choose $\mathcal{S}\left(l\right) \in \left\{ 0, \ldots, \min \left\{ \left\lfloor \mathcal{S}\left(T^*\right) - 1 + \frac{1 - \mathcal{H}\left(T^*\right)}{\lambda} \right\rfloor, q \right\} \right\}$. The minimum comes from the fact that $l \in \mathcal{C}$ and $\mathcal{S}\left(l\right) \leq q$.

- For each $h = \mathcal{S}\left(l\right)$, we construct $l$ by choosing $h$ features among the total $q$ features, there are $\binom{q}{h}$ such choices.

- For each choice among the $\binom{q}{h}$ choices, for each feature among the $h$ features, there are $C$ choices of values, therefore there are $C^h \binom{q}{h}$ branches with depth $h$.

- For each branch of depth $h$, when it is expanded, $(q - h) C$ evaluations occur.

With these considerations, we deduce that:

$$\Gamma\left(q, C, \lambda\right) \leq \sum_{h=0}^{\kappa} (q - h) C^{h+1} \binom{q}{h}; \quad \kappa = \min \left\{ \left\lfloor \mathcal{S}\left(T^*\right) - 1 + \frac{1 - \mathcal{H}\left(T^*\right)}{\lambda} \right\rfloor, q \right\}$$

$\square$

**Theorem 4.4** (Complexity). *Let* $\Gamma\left(q, C, \lambda\right)$ *denote the total number of branch evaluations performed by* BRANCHES *for a classification problem with a number of features* $q \geq 2$, *a penalty parameter* $0 < \lambda < 1$, *and a number of categories per feature* $C \geq 2$. *Then,* $\Gamma\left(q, C, \lambda\right)$ *satisfies:*

$$\Gamma\left(q, C, \lambda\right) \leq \sum_{h=0}^{\kappa} (q - h) C^{h+1} \binom{q}{h}$$

*where* $\kappa = \min \left\{ \left\lfloor \frac{1}{K\lambda} \right\rfloor - 1, q \right\}$.

*Proof.* This is a corollary of Theorem G.5. To make the bound problem-independent, let us upper bound $\kappa$ and make it independent of $T^*$. We know that:

$$\mathcal{H}_\lambda\left(T^*\right) = -\lambda \mathcal{S}\left(T^*\right) + \mathcal{H}\left(T^*\right) \geq \mathcal{H}_\lambda\left(\Omega\right) = \mathcal{H}\left(\Omega\right) = \mathbb{P}\left[Y = k^*\left(\Omega\right)\right] \geq \frac{1}{K}$$

$$\implies \mathcal{S}\left(T^*\right) - 1 + \frac{1 - \mathcal{H}\left(T^*\right)}{\lambda} \leq \frac{K - 1}{K\lambda} - 1$$

Which concludes the proof. $\square$

*Table 4.* Number of examples $n$, number of features $q$, number of classes $K$ and penalty parameter $\lambda$ for the different datasets used in our experiments. $d$ is the depth of optimal solution, which we employ in Appendix H.3.

| Dataset | $n$ | $q$ | $K$ | $\lambda$ | $d$ |
|---|---|---|---|---|---|
| monk1 | 124 | 17 | 2 | 0.01 | 4 |
| monk1-l | 124 | 11 | 2 | 0.01 | 5 |
| monk1-f | 124 | 11 | 2 | 0.001 | 7 |
| monk1-o | 124 | 6 | 2 | 0.01 | 3 |
| monk2 | 169 | 17 | 2 | 0.001 | 6 |
| monk2-f | 169 | 11 | 2 | 0.001 | 9 |
| monk2-o | 169 | 6 | 2 | 0.001 | 3 |
| monk3 | 122 | 17 | 2 | 0.001 | 8 |
| monk3-f | 122 | 11 | 2 | 0.001 | 7 |
| monk3-o | 122 | 6 | 2 | 0.001 | 4 |
| tic-tac-toe | 958 | 27 | 2 | 0.005 | 6 |
| tic-tac-toe-f | 958 | 18 | 2 | 0.005 | 6 |
| tic-tac-toe-o | 958 | 9 | 2 | 0.005 | 5 |
| car-eval | 1728 | 21 | 4 | 0.005 | 8 |
| car-eval-f | 1728 | 15 | 4 | 0.005 | 8 |
| car-eval-o | 1728 | 6 | 4 | 0.005 | 4 |
| nursery | 12960 | 27 | 5 | 0.01 | 6 |
| nursery-f | 12960 | 19 | 5 | 0.01 | 4 |
| nursery-o | 12960 | 8 | 4 | 0.01 | 3 |
| mushroom | 8124 | 117 | 2 | 0.01 | 3 |
| mushroom-f | 8124 | 95 | 2 | 0.01 | 3 |
| mushroom-o | 8124 | 22 | 2 | 0.01 | 1 |
| kr-vs-kp | 3196 | 73 | 2 | 0.01 | 4 |
| zoo | 101 | 36 | 7 | 0.001 | 6 |
| zoo-f | 101 | 20 | 7 | 0.001 | 7 |
| zoo-o | 101 | 16 | 7 | 0.001 | 5 |
| lymph | 148 | 59 | 4 | 0.01 | 4 |
| lymph-f | 148 | 41 | 4 | 0.01 | 5 |
| lymph-o | 148 | 18 | 4 | 0.01 | 4 |
| balance | 576 | 20 | 3 | 0.01 | 8 |
| balance-f | 576 | 16 | 3 | 0.01 | 10 |
| balance-o | 576 | 4 | 3 | 0.01 | 3 |

## H. Additional Experiments

All the experiments were run on a personal Machine (2,6 GHz 6-Core Intel Core i7), they are easily reproducible. Below we provide references to the implementations we used:

- BRANCHES https://github.com/Chaoukia/branches.

- DL8.5 https://github.com/aia-uclouvain/pydl8.5.git.

- OSDT https://github.com/xiyanghu/OSDT.git.

- GOSDT https://github.com/ubc-systopia/gosdt-guesses.git.

- MurTree https://github.com/MurTree/pymurtree.git.

- STreeD https://github.com/AlgTUDelft/pystreed.git.

*Table 5.* Comparing BRANCHES with the state of the art python implementations for a large maximum depth 20.

| Dataset | OSDT | | | | | PYGOSDT | | | | | BRANCHES | | | | |
|---|---|---|---|---|---|---|---|---|---|---|---|---|---|---|---|
| | objective | accuracy | splits | time (s) | iterations | objective | accuracy | splits | time (s) | iterations | objective | accuracy | splits | time (s) | iterations |
| monk1 | **0.940** | **1** | **6** | 2.38 | 94901 | **0.940** | **1** | **6** | 6.03 | 174523 | **0.940** | **1** | **6** | 0.05 | 146 |
| monk1-l | **0.930** | **1** | **7** | 71 | 2028577 | **0.930** | **1** | **7** | 181 | 3731292 | **0.930** | **1** | **7** | 0.02 | 117 |
| monk1-f | 0.971 | 1 | 29 | *TO* | 22308 | 0.970 | 1 | 30 | *TO* | 2018 | **0.983** | **1** | **17** | 0.39 | 2125 |
| monk1-o | — | — | — | — | — | — | — | — | — | — | 0.900 | 1 | 10 | 0.02 | 64 |
| monk2 | 0.948 | 1 | 52 | *TO* | 41 | 0.948 | 1 | 52 | *TO* | 44 | **0.968** | **1** | **32** | 14.2 | 60611 |
| monk2-f | 0.904 | 0.982 | 76 | *TO* | 44083 | 0.903 | 0.982 | 77 | *TO* | 32475 | **0.933** | **1** | **67** | 2.94 | 28968 |
| monk2-o | — | — | — | — | — | — | — | — | — | — | 0.955 | 1 | 45 | 0.18 | 1213 |
| monk3 | 0.976 | 0.991 | 15 | *TO* | 17728 | 0.976 | 0.991 | 15 | *TO* | 5765 | **0.985** | **1** | **15** | 4.05 | 14807 |
| monk3-f | 0.975 | 0.991 | 15 | *TO* | 11875 | 0.973 | 0.991 | 17 | *TO* | 897 | **0.983** | **1** | **17** | 0.36 | 3026 |
| monk3-o | — | — | — | — | — | — | — | — | — | — | 0.987 | 1 | 13 | 0.03 | 156 |
| tic-tac-toe | 0.794 | 0.869 | 15 | *TO* | 76 | 0.794 | 0.869 | 15 | *TO* | 69 | **0.838** | **0.928** | **18** | *TO* | 390000 |
| tic-tac-toe-f | 0.764 | 0.824 | 11 | *TO* | 40 | 0.808 | 0.824 | 11 | *TO* | 37 | **0.850** | **0.945** | **19** | 16.3 | 74627 |
| tic-tac-toe-o | — | — | — | — | — | — | — | — | — | — | 0.773 | 0.858 | 17 | 0.68 | 3339 |
| mushroom | **0.955** | **0.985** | **3** | 76.2 | 1186819 | **0.955** | **0.985** | **3** | 211 | 2681260 | **0.955** | **0.985** | **3** | *TO* | 21000 |
| mushroom-f | **0.945** | **0.985** | **4** | *TO* | 4704419 | **0.945** | **0.985** | **4** | *TO* | 2487909 | **0.945** | **0.985** | **4** | *TO* | 24000 |
| mushroom-o | — | — | — | — | — | — | — | — | — | — | 0.975 | 0.985 | 1 | 0.15 | 6 |
| kr-vs-kp | **0.900** | **0.940** | **4** | *TO* | 67161 | **0.900** | **0.940** | **4** | *TO* | 25379 | **0.900** | **0.940** | **4** | *TO* | 46000 |

MurTree, STreeD and GOSDT use support set caching and the similarity bound. On the other hand, the current implementation of BRANCHES only supports branch caching and it does not include a similarity bound. Additionally, we set the maximum nodes for MurTree to 80. We noticed that without this constraint, the kernel dies immediately. The optimal sparse DT for all the experiments we consider have less than 80 nodes, thus this constraint should not cause MurTree to be suboptimal.

Table 4 summarises the properties of the datasets we employed in our experiments. The $\lambda$ values reported in Table 4 are those employed in the experiments of Section 5, they were chosen from a pool of values $\lambda \in \{0.1, 0.05, 0.025, 0.01, 0.005, 0.001\}$ to yield meaningful solutions. Appendix H.2 provides the induced Pareto fronts by these values for a 10-fold crossvalidation.

### H.1. Comparing BRANCHES with Python implementations

We have shown in Section 5 that BRANCHES outperforms its C++ competitors in terms of runtime even though it is implemented in Python. To further illustrate the importance of this achievement, we compare BRANCHES with Python implementations and show the vast difference in performance in favour of BRANCHES.

Table 5 compares BRANCHES with OSDT and PYGOSDT, it contains less datasets than Table 2 because the implementations of OSDT and PYGOSDT are restrained to binary classification problems. Table 5 shows that BRANCHES outperforms OSDT and GOSDT in terms of runtime, number of iterations and quality of the proposed solution on all the experiments except mushroom. On mushroom, OSDT and PYGOSDT terminate in $76.2s$ and $211s$ respectively while BRANCHES reaches timeout. Nevertheless, BRANCHES' anytime behaviour still allowed the retrieval of the true optimal sparse DT in this case.

Table 5 showcases the shortcomings of Python implementations for the sparsity problem as both OSDT and PYGOSDT reach timeout and are suboptimal on the vast majority of datasets. Due to the optimisation difficulty of the sparsity problem, a lot of care has to be dedicated to low-level optimisation of the proposed implementations, a property that C++ offers better than Python, hence the vast adoption of C++ in the state of the art algorithms solving the sparsity problem. BRANCHES being implemented in Python and yet outperforming its C++ competitors is a testament to its efficient AO\*-based search strategy. Moreover, a future C++ implementation of BRANCHES is very promising especially when coupled with multi-threading.

### H.2. Pareto fronts

The purpose of this section is to compare DT algorithms of different natures. BRANCHES solves the sparsity problem, CART is a greedy search algorithms for DTs and DL8.5 seeks the optimal DT subject to a hard constraint on maximum depth. Figure 13, Figure 14, Figure 15 and Figure 16 report the median and the quartiles of the accuracy and number of splits of the proposed solutions induced by a 10 fold crossvalidation, branches-o in the legends represents BRANCHES applied to ordinal

encoding. For DL8.5, we ran it with with maximum depths raging between 1 and 10. As for BRANCHES and CART, they were run with $\lambda \in \{0.1, 0.05, 0.025, 0.01, 0.005, 0.001\}$. Note that $\lambda$ in the context of CART refers to the cost-complexity parameter (ccp_alpha in scikit-learn) that is used for the pruning phase. It plays a similar role to the penalty parameter $\lambda$, hence why we tested both with the same set of values. A similar set of experiments was conducted by Hu et al. (2020) albeit with the maximum depth for CART instead of the cost-complexity parameter. We noticed that the cost-complexity parameter yields better frontiers for CART than the maximum depth, hence our choice. Figure 15 and Figure 16 report the same results as Figure 13 and Figure 14 respectively but with the inclusion of DL8.5. We made this distinction for clarity purposes because DL8.5's solutions become so complex (high number of splits) that the differences between the frontiers of CART and BRANCHES become less apparent.

Figure 13 shows that BRANCHES displays better training frontiers than those of CART, and on many occasions branches-o yields even better training frontiers as we can see in monk1-f, monk2-f, careval-f, nursery, nursery-f, mushroom, mushroom-f, zoo, zoo-f, lymph-f and balance-f. This indicates that applying BRANCHES on an ordinal encoding of the data is not only good from a scalability standpoint (as we have seen with the extremely fast convergence in Table 2) but can also yield better solutions in terms of accuracy and the number of splits. On the other hand, the differences in the test frontiers, depicted in Figure 14 are less clear than for the training frontiers. Nevertheless, we can still observe an advantage for BRANCHES' test frontiers over CART's. Moreover, Figure 14 illustrates a tendency of branches-o to overfit as we observe in monk2, monk2-f, tictactoe, tictactoe-f, lymph, lymph-f, balance and balance-f. This phenomenon can be explained by the tendency of non-binary DTs on ordinal encodings to induce branches containing smaller subsets of data than those induced by their binary DTs counterparts, thus increasing the risk of overfitting. Notice that for large datasets (nursery with 12960 data and mushroom with 8124 data) branches-o does not run into this overfitting issue.

Figure 15 and Figure 16 illustrate how DL8.5 actively seeks more and more complex DTs as the maximum depth parameter increases, thus disregarding sparsity concerns and inducing poor frontiers compared to BRANCHES and CART. To alleviate this tendency, Aglin et al. (2020) further include a constraint on the minimum support size of DL8.5, i.e. the minimum number of data that a branch has to contain in order to be considered for expansion. In the next section, we investigate this constraint and show that even with the best configurations DL8.5 is still suboptimal.

### H.3. Suboptimality of DL8.5

In all the experiments, we set the maximum depth of DL8.5 to the depth of the true optimal sparse DT that we derive using BRANCHES, these depth values are reported in Table 4. We analyse DL8.5's performance over 20 values of minimum support size (minimum allowed number of instances per branch) taken between 1 and 50. Figure 17 shows that, even with the knowledge of the best possible maximum depth parameter and the best configuration of minimum support size, DL8.5 almost never approaches the baseline derived by BRANCHES. Figure 18 further illustrates that BRANCHES' solution is always outside the accuracy-splits frontier displayed by DL8.5, which means that BRANCHES' solution always dominates DL8.5's solution from a sparsity perspective.

### H.4. Comparison across a wide range of maximum depths

We compare the performance of BRANCHES, GOSDT and STreeD on a set of maximum depth values ranging from 4 to 20. For all the algorithms, we compare the objective $\mathcal{H}_\lambda$, accuracy and number of splits of the proposed solution in addition to the execution time. Moreover, we compare GOSDT and BRANCHES in terms of the number of iterations. We also compare the depth of the proposed solutions between BRANCHES and STreeD only because, to our knowledge, the implementation of GOSDT does not provide this metric. The legends of the figures contain:

- **branches:** BRANCHES applied to binary encoding.

- **branches-o:** BRANCHES applied to ordinal encoding.

- **gosdt:** GOSDT applied to binary encoding.

- **streed:** STreeD applied to binary encoding.

- **streed1:** STreeD with the depth 2 solver, introduced by Demirović et al. (2022), disabled. The reason we introduce this baseline is to assess the contribution of the depth 2 solver to STreeD's performance. We shall see that this option improves STreeD's performance significantly.

The results are reported in Figures 19 to 40. We start by discussing the performance of GOSDT. Interestingly, the objective $\mathcal{H}_\lambda$ reported for GOSDT does not match the one reported for BRANCHES and STreeD. This might indicate that the implementation of max depth for GOSDT forces the solution to have a depth strictly lower than max depth, unlike BRANCHES and STreeD where the solution is allowed to reach max depth. Nevertheless, this does not undermine the following discussion. Surprisingly, GOSDT performs often worse when limiting its maximum depth than otherwise. There are even cases where it runs out of memory even though this phenomenon has not been observed in Table 2. This happens for mushroom, mushroom-f, lymph, lymph-f, tic-tac-toe, krvskp, nursery and nursery-f, hence why GOSDT's results are missing in those figures. We note that this phenomenon has been observed by McTavish et al. (2022), the authors that incorporate the maximum depth parameter into GOSDT. They state:

*Interestingly, using a large depth constraint is often less efficient than removing the depth constraint entirely for GOSDT, because when we use a branch-and-bound approach with recursion, the ability to re-use solutions of recurring sub-problems diminishes in the presence of a depth constraint.*

On the other hand, despite being a BFS method as well and as such being more memory consuming than DFS methods, BRANCHES never ran into this issue in both sets of experiments. This is likely explained by BRANCHES' ability to terminate significantly earlier than GOSDT in terms of iterations, and thus terminating before running into memory issues. This large discrepancy in the number of iterations was observed in Table 2 and is further observed in all the experiments in this section. Furthermore, except on few cases such as balance, BRANCHES dominates GOSDT in terms of execution speed.

On these experiments the comparison with STreeD is more insightful. A common pattern is that STreeD dominates BRANCHES in terms of speed for small depths up until a certain point where BRANCHES becomes the dominating method. This is best observed in tic-tac-toe-f, car-eval-f, nursery-f, zoo and balance-f. We note also that in some cases such as balance, lymph-f, lymph and tic-tac-toe, STreeD proposes better solutions altogether than BRANCHES because the latter reaches timeout, albeit for higher depths we cannot even compare the solutions of BRANCHES and STreeD because STreeD does not even propose solutions then. A very insightful experiment here is mushroom. We have seen in Table 2 that STreeD performs exceptionally well on mushroom and mushroom-f even for a large maximum depth. The experiments on this section further support this observation. Thus we naturally wonder: **Why does STreeD perform exceptionally well on the mushroom datasets? Is it the DFS strategy or something else?** To investigate this, we looked into a powerful tool in STreeD's arsenal, the depth 2 solver that was introduced by Demirović et al. (2022). This solver allows for finer estimates to be computed early on, it has been proven to yield significant runtime improvements, and moreover neither GOSDT nor BRANCHES utilise it for now. Fortunately, STreeD's implementation allows the disabling of the depth 2 solver. Indeed, the depth 2 solver turns out to be paramount to STreeD's performance, without it, STreeD always runs out of time on the mushroom datasets. Moreover, now BRANCHES largely dominates STreeD across the full range of max depth values, except on very few cases such as depths 13, 14 and 15 in balance-f. With this, we conclude that STreeD's better performance on mushroom and smaller depths is not due to its DFS strategy, but rather to the depth 2 solver. This observation motivates us to consider adapting the depth 2 solver to the purification bound and incorporating it in a future version of BRANCHES. This is a promising idea that could push BRANCHES to dominate STreeD for small depths as well.

Next we discuss the anytime behaviour. We have seen in Table 2 that STreeD lacks the anytime behaviour unlike GOSDT and BRANCHES, which hinders its applicability. To cite a few examples, STreeD (with the depth 2 solver) starts reaching timeout from depth 7 for lymph and depth 8 for lymph-f and tic-tac-toe. On the other hand, notice that the solutions proposed by BRANCHES even after reaching timeout always dominate those proposed by GOSDT in terms of the objective $\mathcal{H}_\lambda$. This is especially the case for car-eval, balance and balance-f and we recall that we could not report GOSDT's performance on many other datasets because it runs our of memory.

Lastly, we discuss the application of BRANCHES to ordinal encoding. On all experiments, the optimal sparse DT is found significantly faster in this setting, even for the largest dataset nursery with 12960 examples. This further supports the scalability potential of this property. Moreover, while it is true that from the objective's perspective, the induced solutions with ordinal encodings are not always the best compared to those induced by binary encoding, however, they happen to be better often such as in monk2-f, monk3, monk3-f, car-eval-f, nursery, nursery-f zoo, zoo-f, lymph, lymph-f, balance-f, mushroom and mushroom-f.

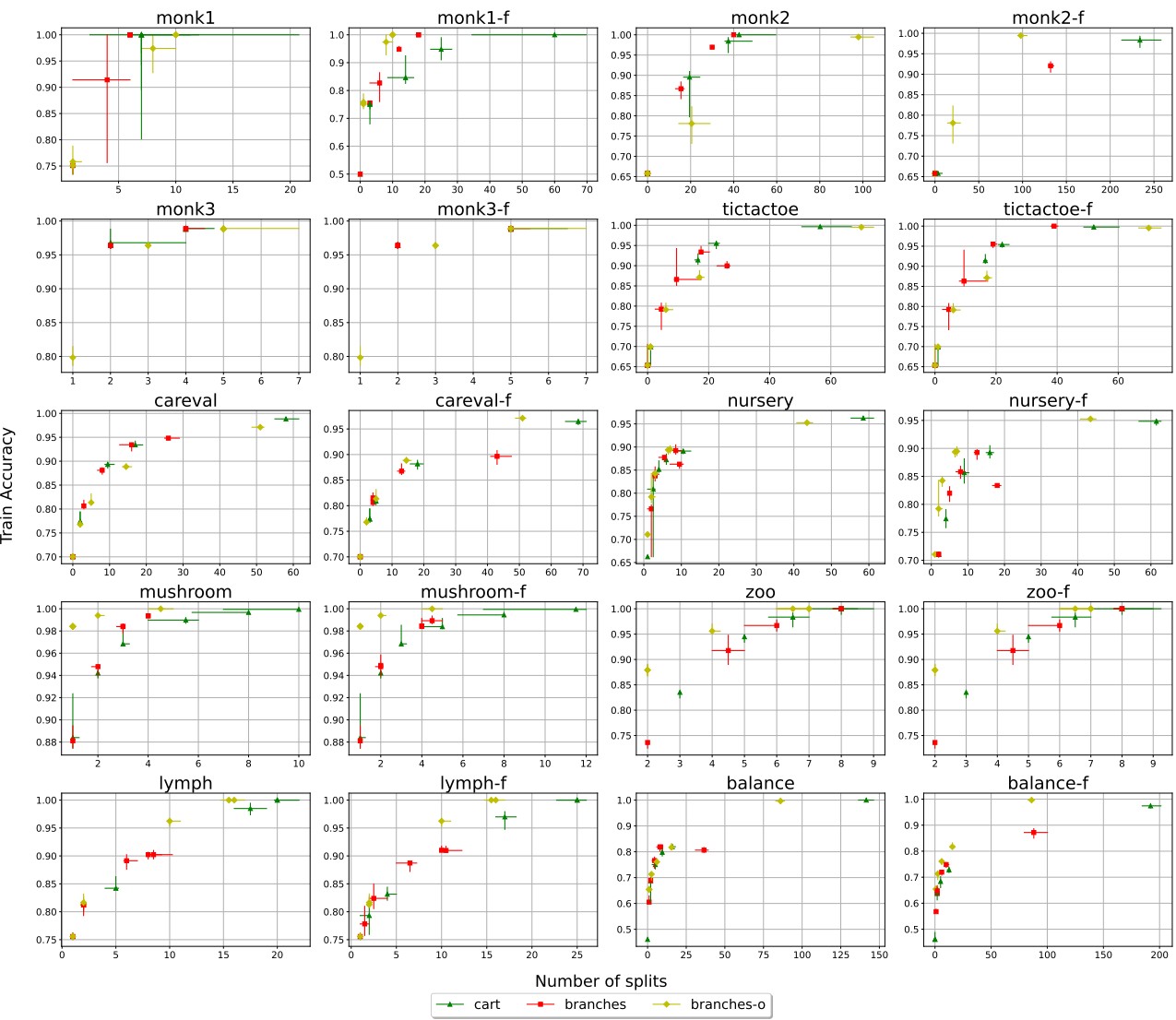

*Figure 13.* Pareto fronts of training accuracy against the number of splits of the proposed solutions. branches-o indicated that BRANCHES is applied to an ordinal encoding of the data.

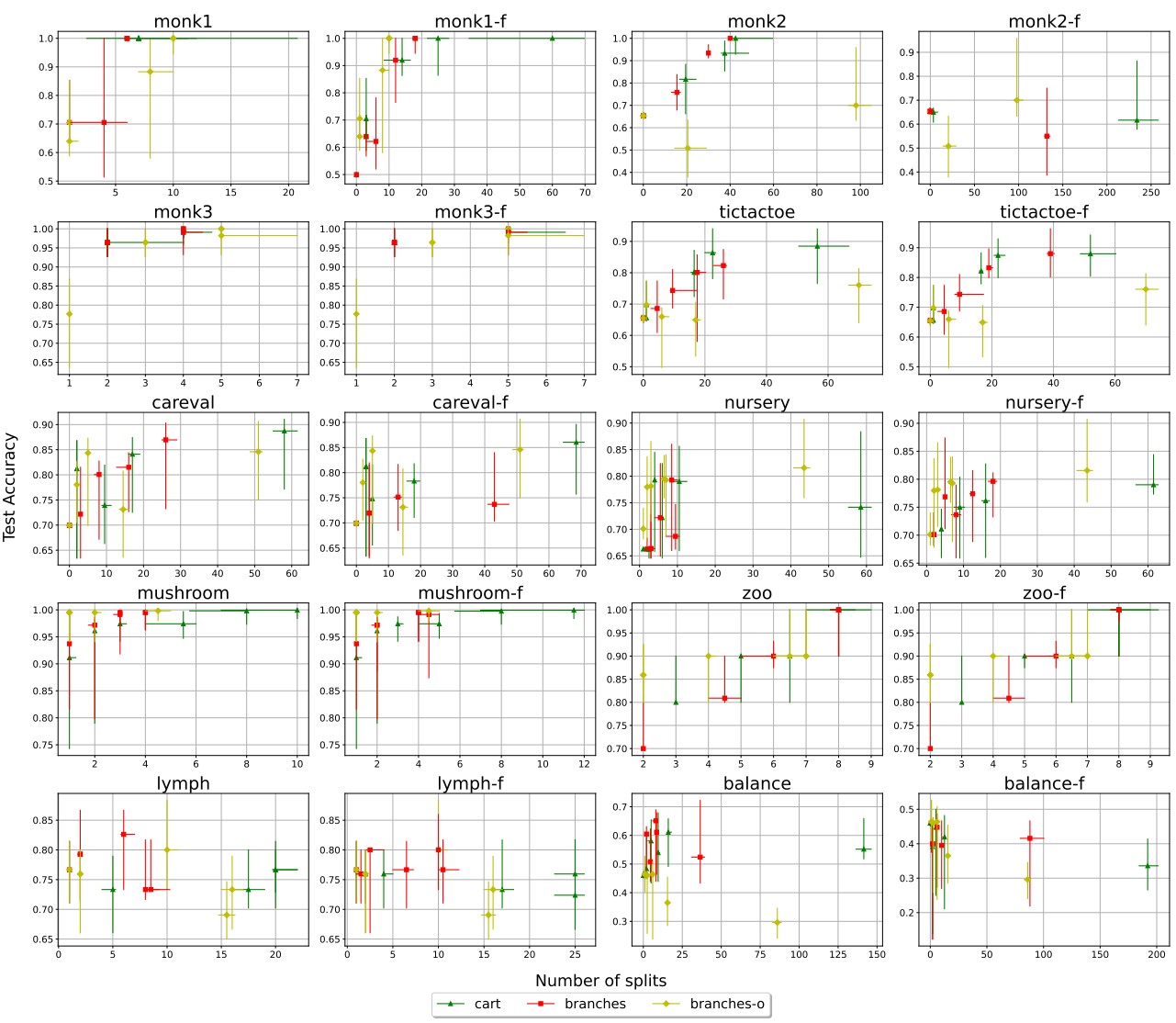

*Figure 14.* Pareto fronts of test accuracy against the number of splits of the proposed solutions. branches-o indicated that BRANCHES is applied to an ordinal encoding of the data.

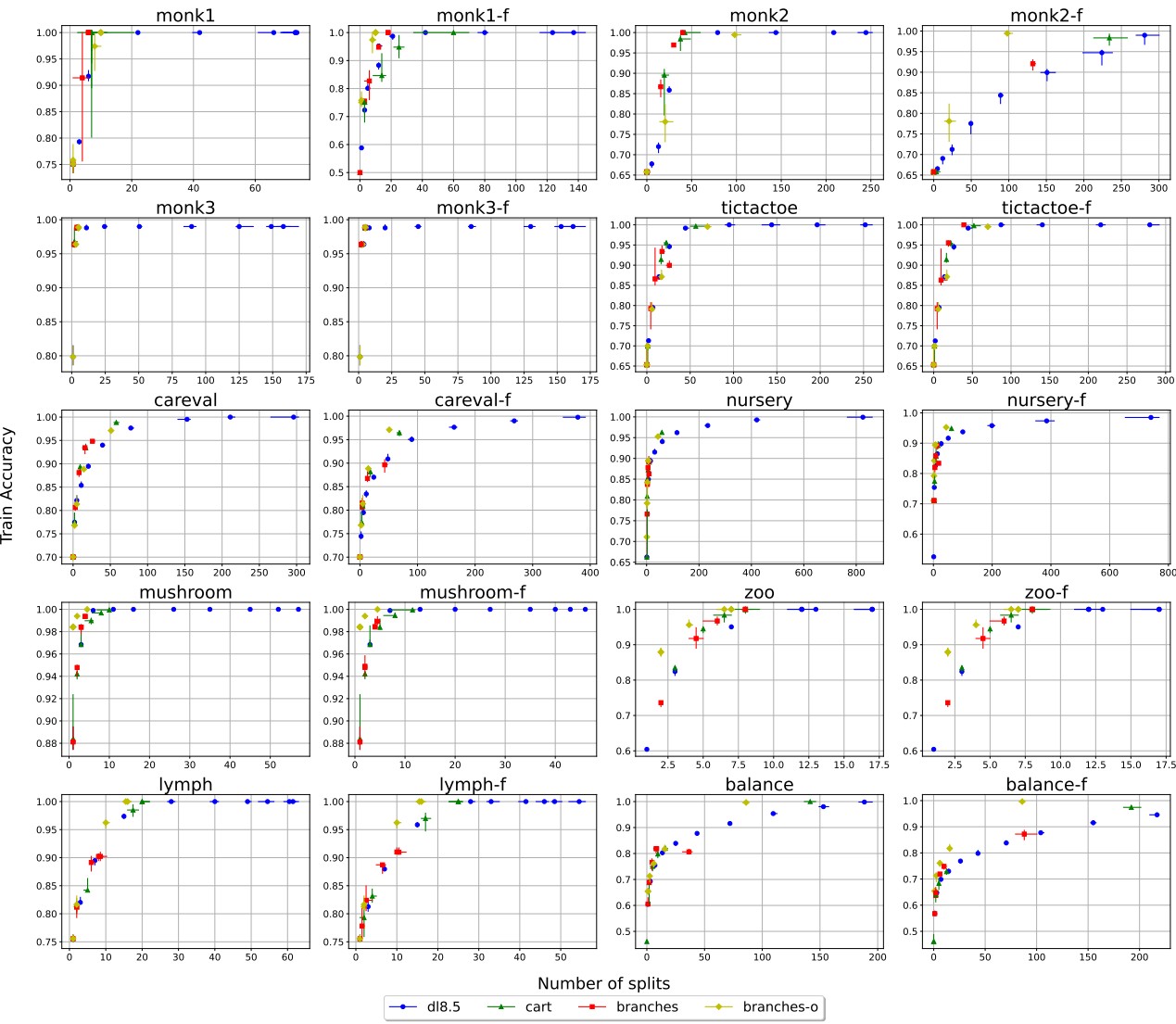

*Figure 15.* Pareto fronts of training accuracy against the number of splits of the proposed solutions. This figure is similar to Figure 13 but further includes DL8.5.

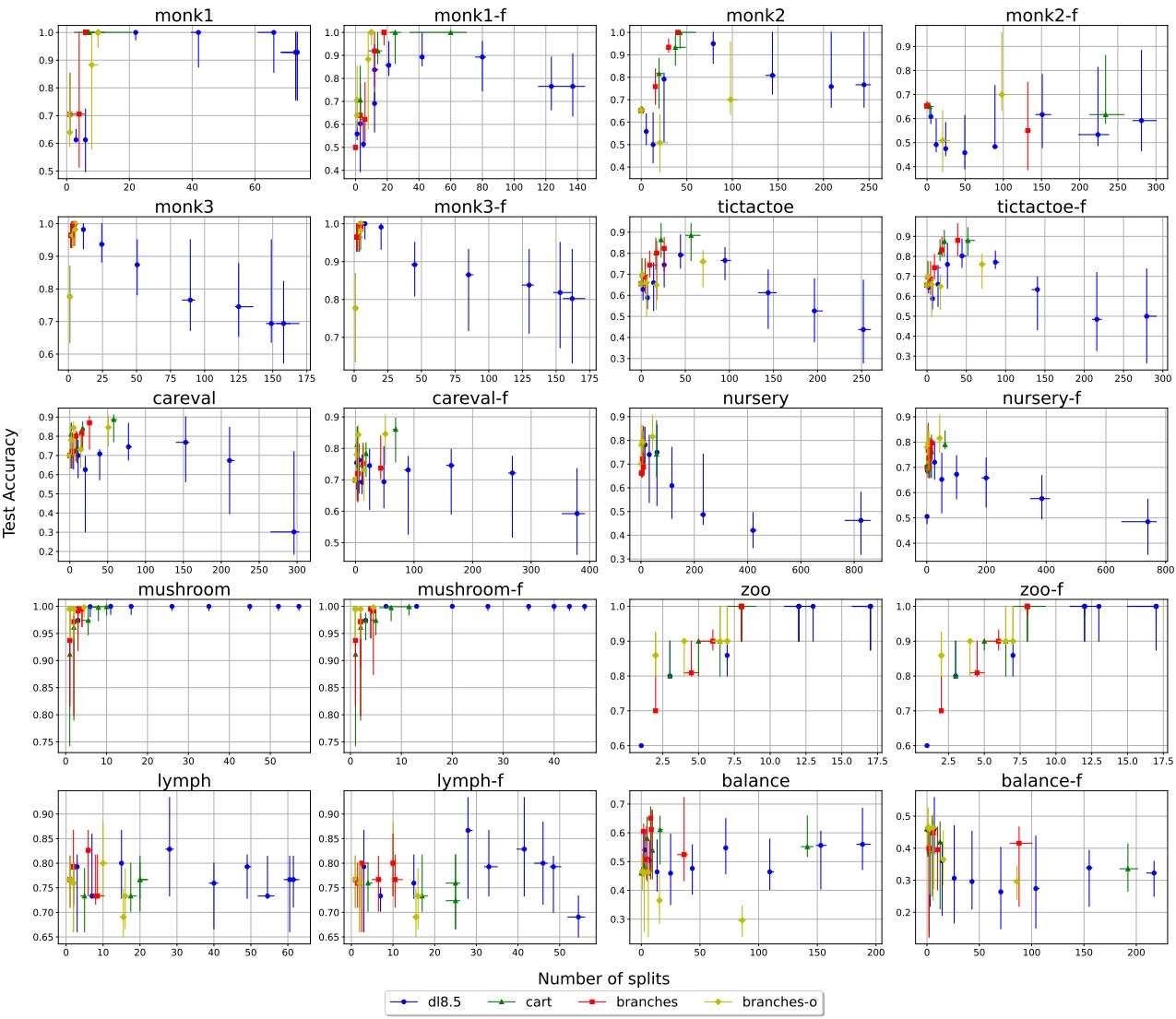

*Figure 16.* Pareto fronts of test accuracy against the number of splits of the proposed solutions. This figure is similar to Figure 14 but further includes DL8.5.

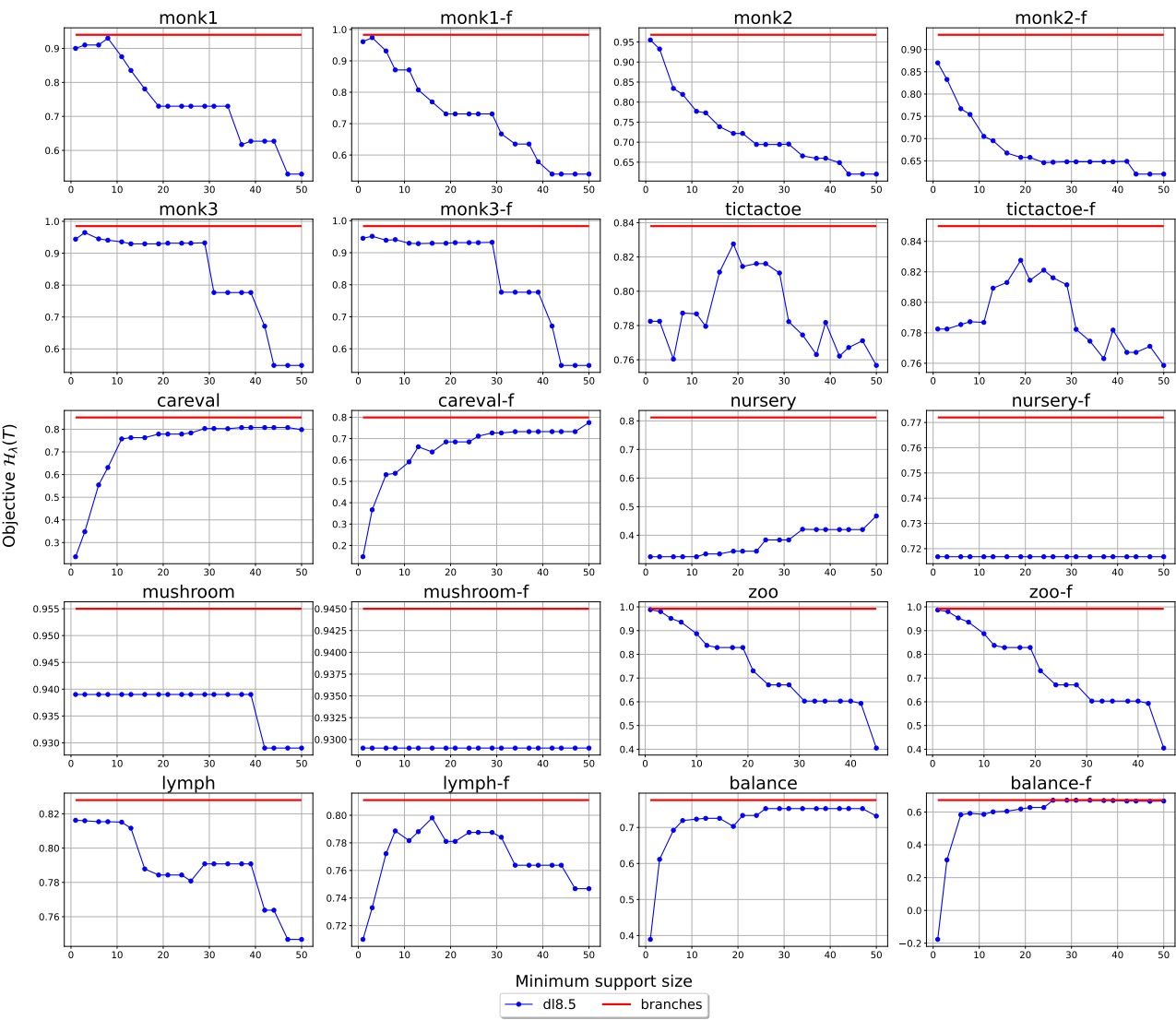

*Figure 17.* Comparing $\mathcal{H}_\lambda(T)$ of the proposed solutions by DL8.5 for different values of the minimum support size with the BRANCHES baseline. The maximum depth of DL8.5 is set to the depth of the true optimal solution, which we derive using BRANCHES.

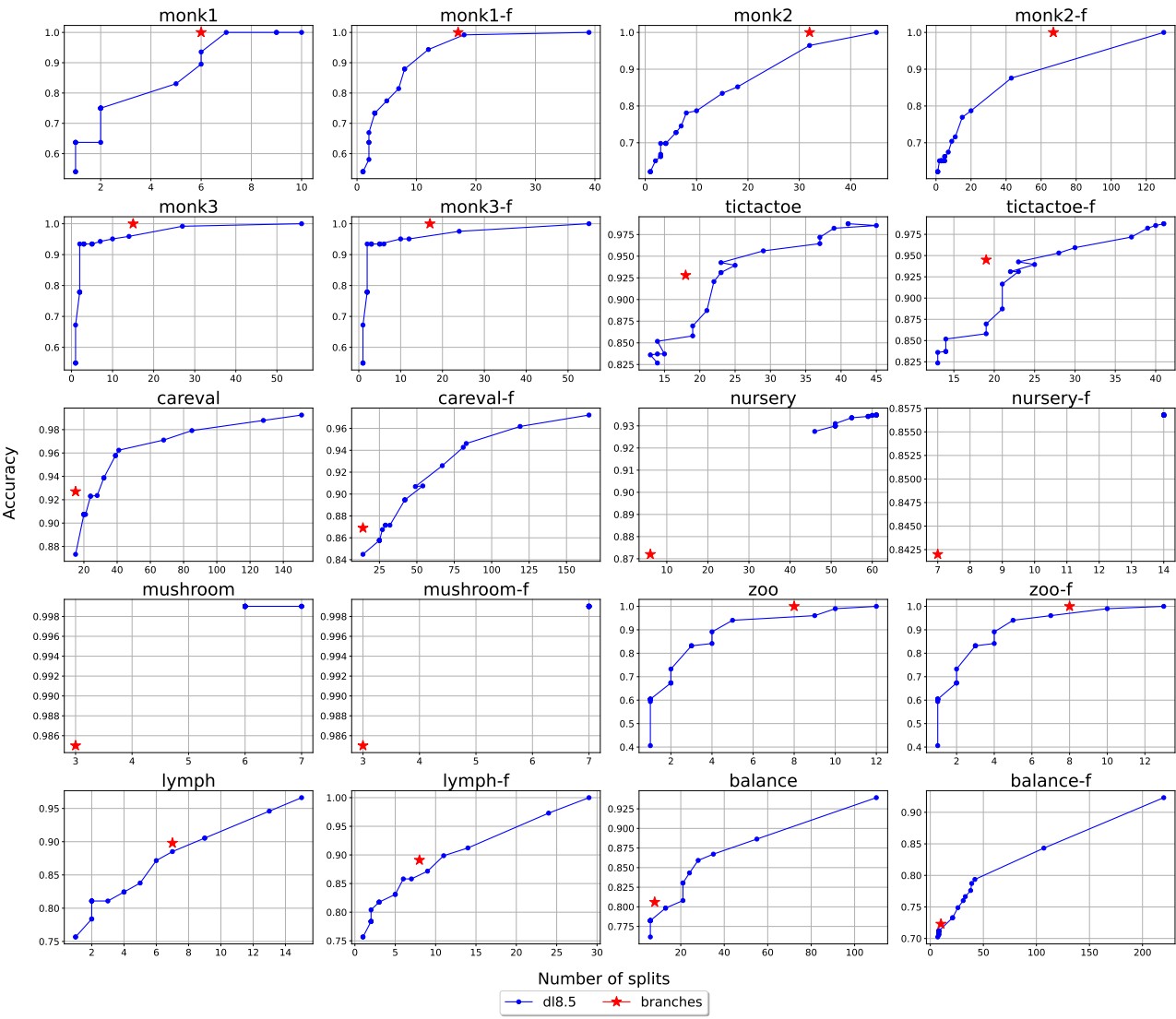

*Figure 18.* Comparing the Accuracy and number of splits of the induced solution by DL8.5 for different values of the minimum support size with the BRANCHES baseline.

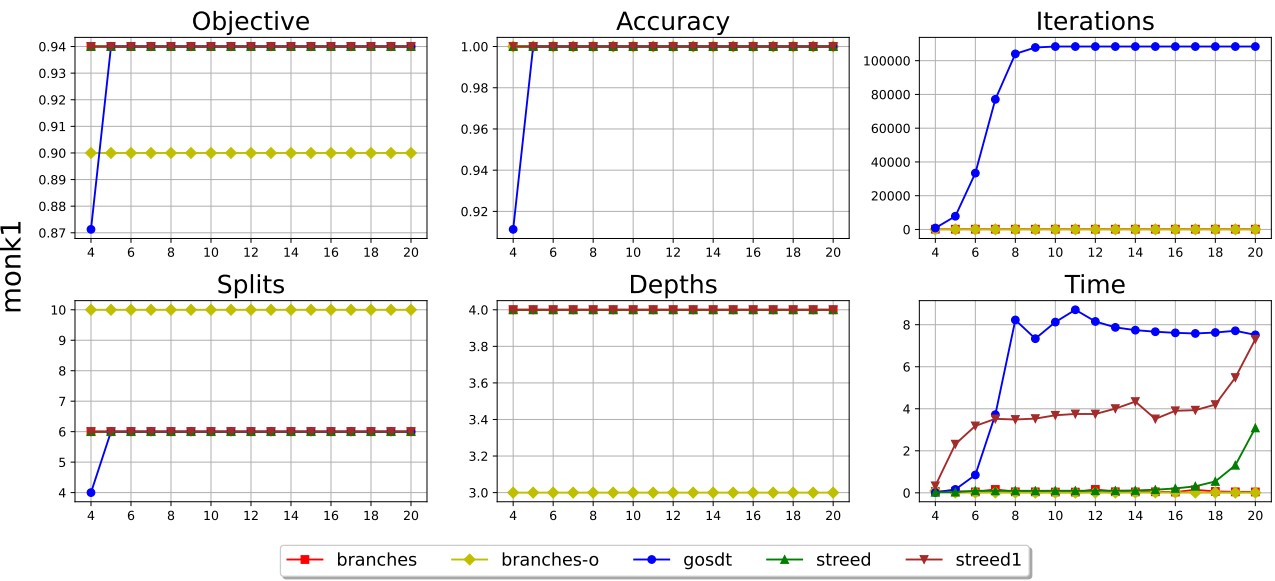

*Figure 19.* Depth analysis for monk1.

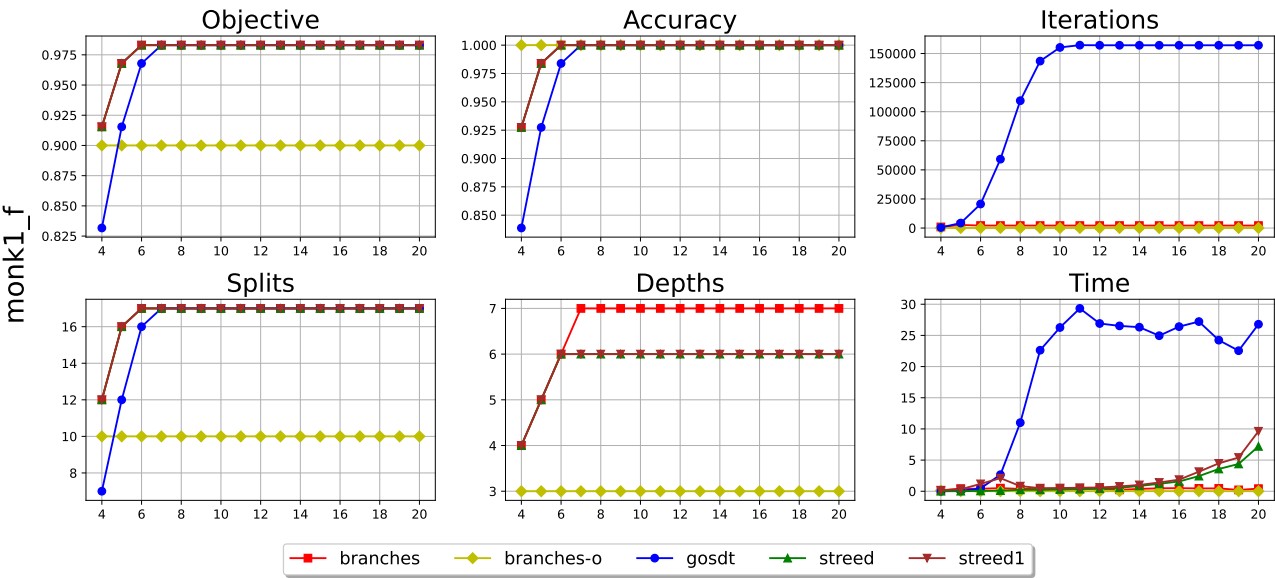

*Figure 20.* Depth analysis for monk1-f.

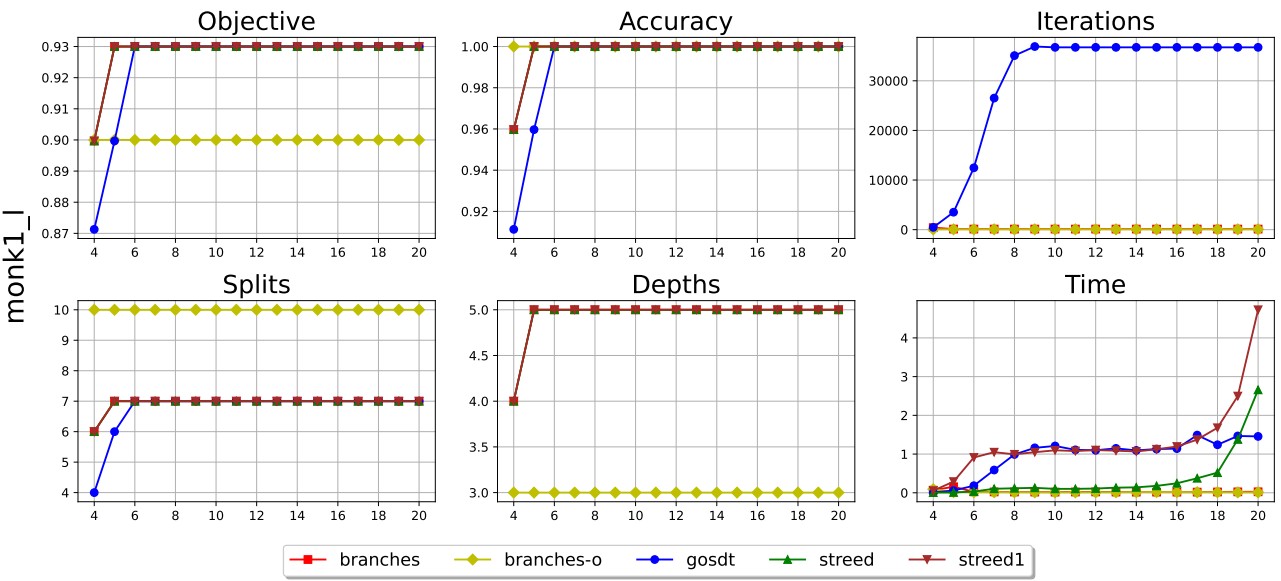

*Figure 21.* Depth analysis for monk1-l.

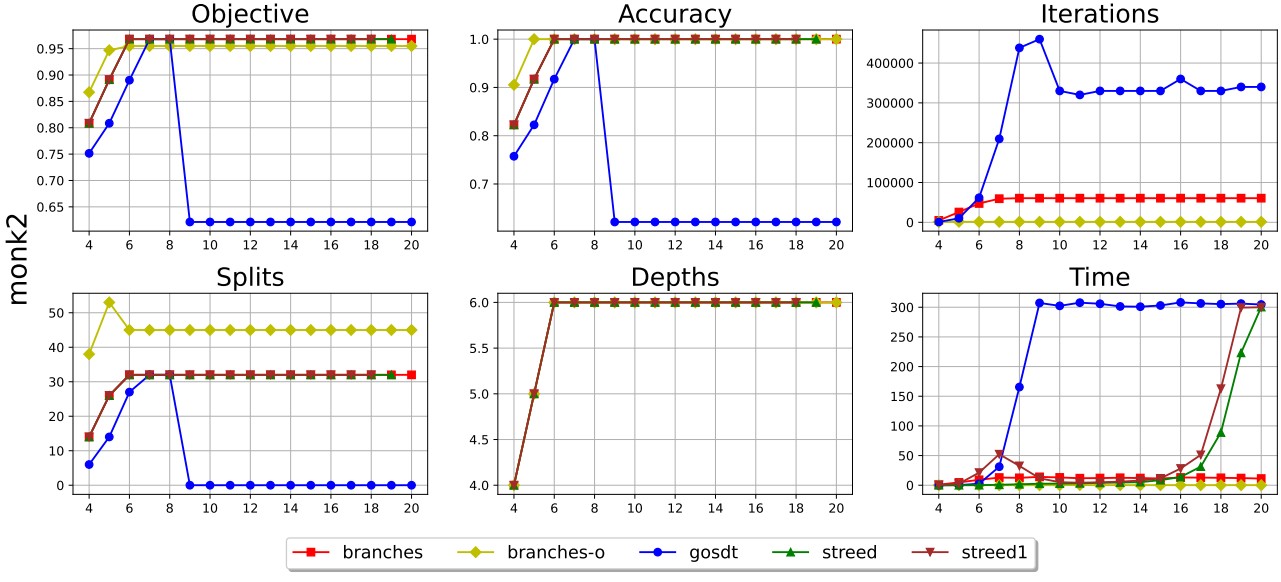

*Figure 22.* Depth analysis for monk2.

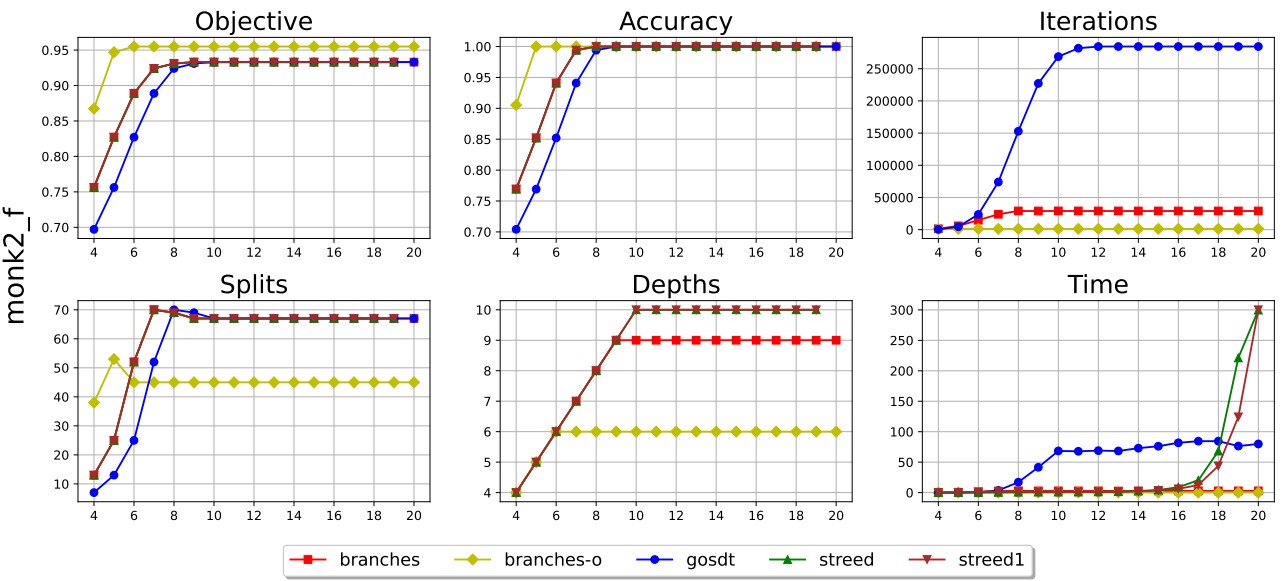

*Figure 23.* Depth analysis for monk2-f.

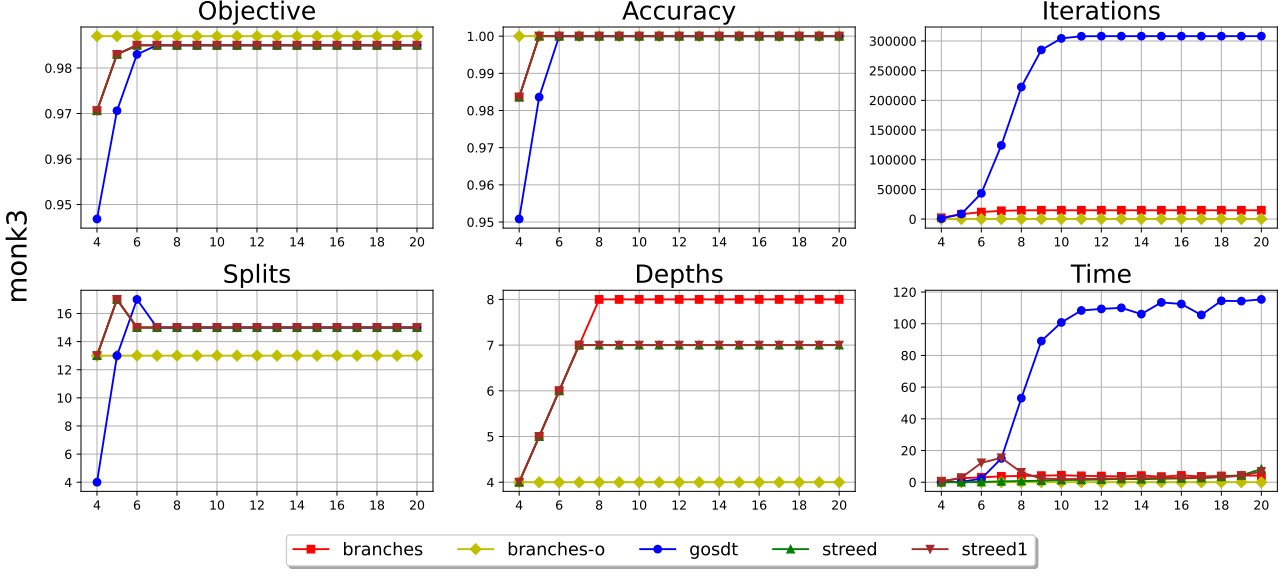

*Figure 24.* Depth analysis for monk3.

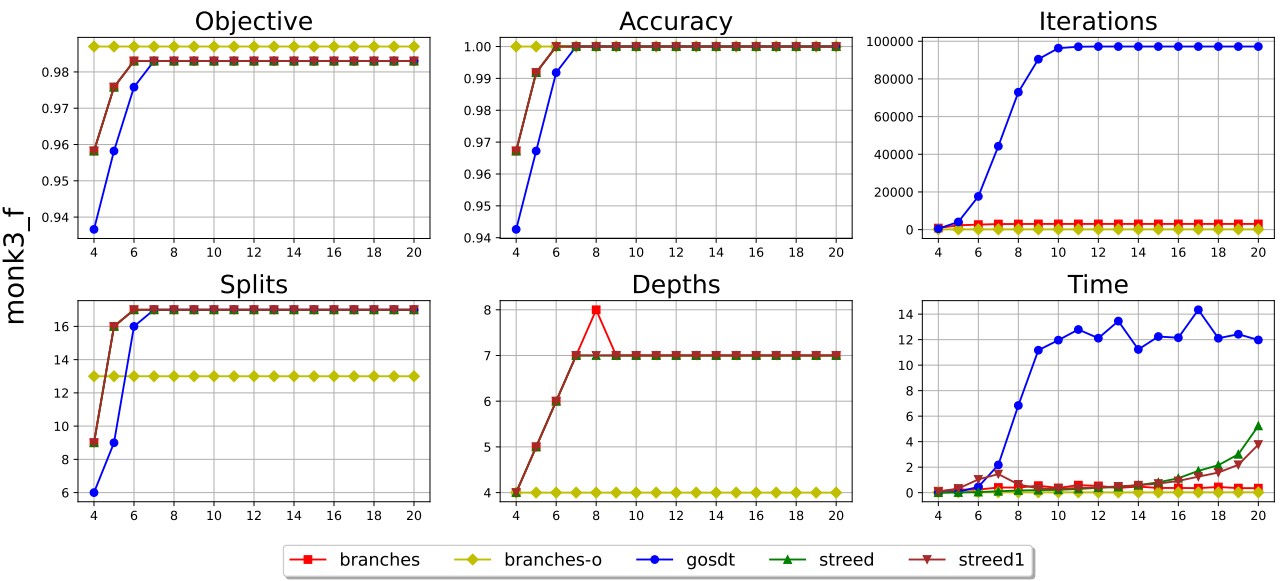

*Figure 25.* Depth analysis for monk3-f.

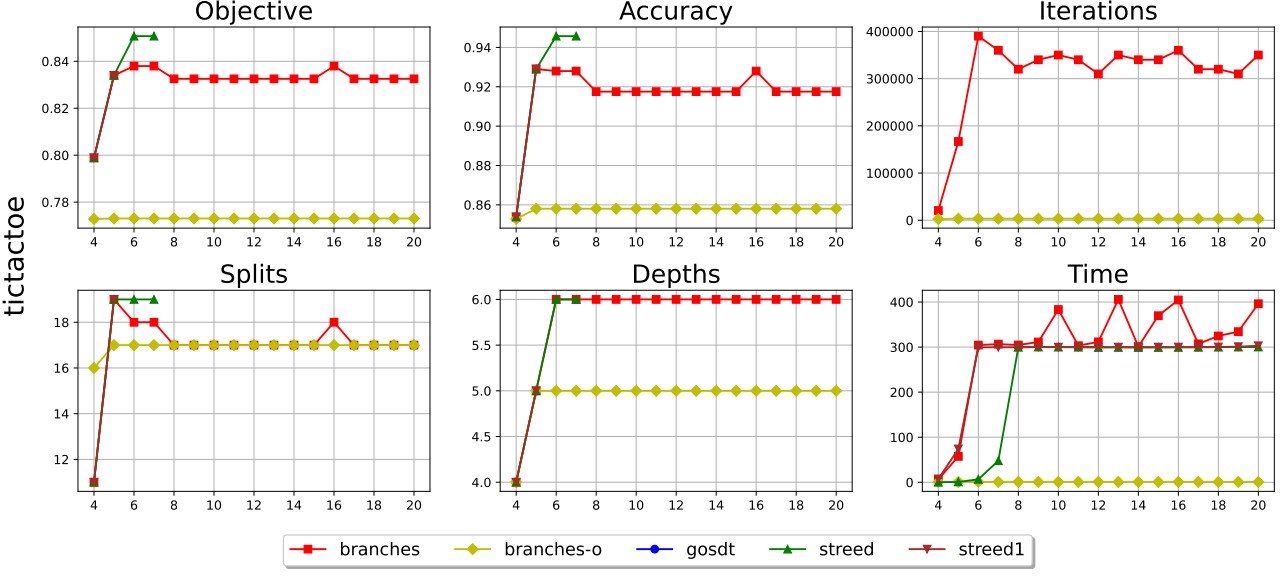

*Figure 26.* Depth analysis for tic-tac-toe.

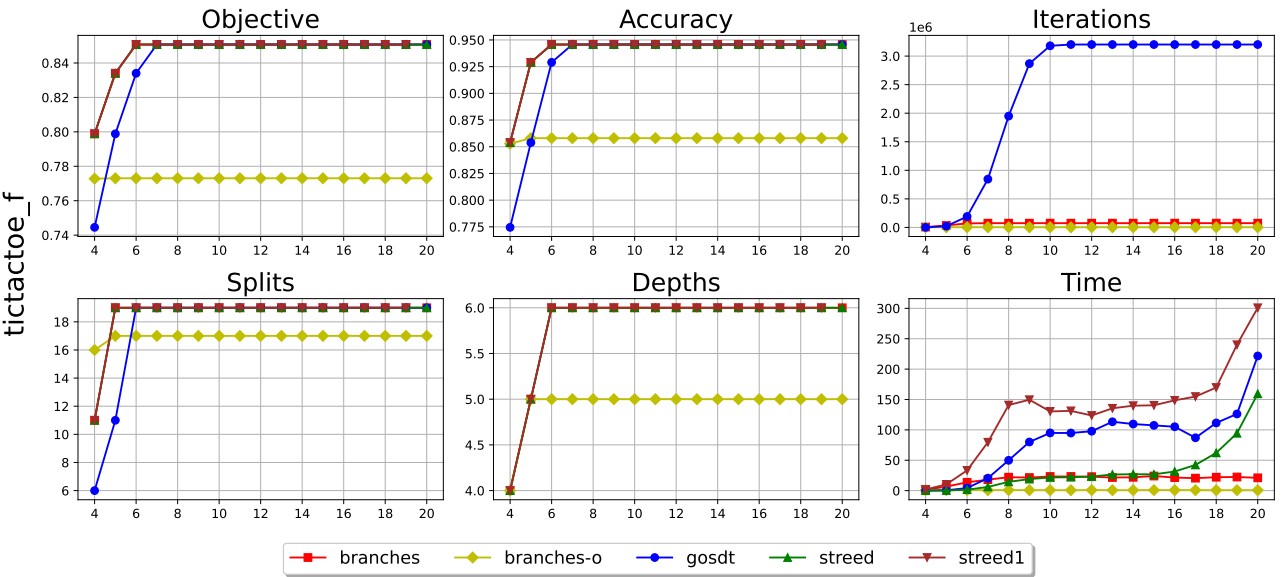

*Figure 27.* Depth analysis for tic-tac-toe-f.

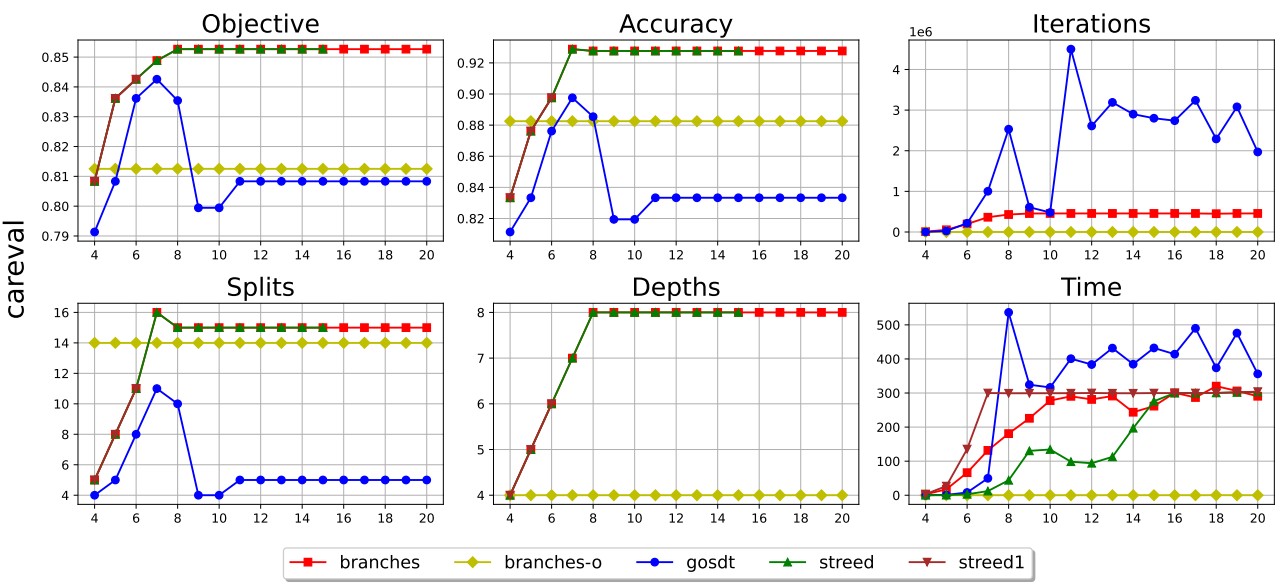

*Figure 28.* Depth analysis for car-eval.

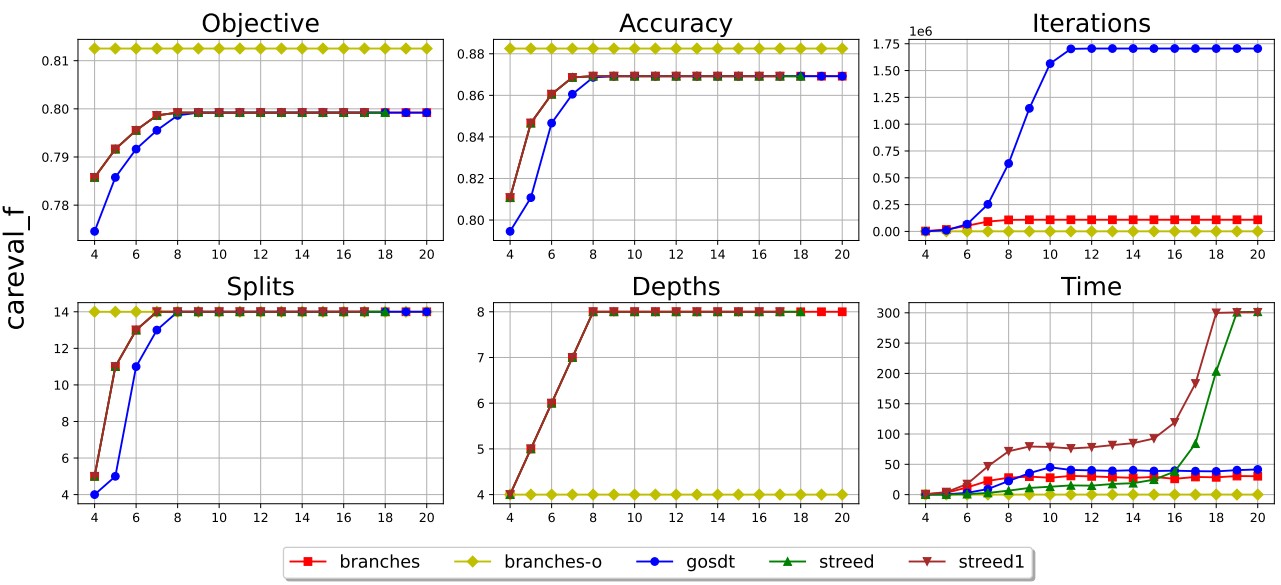

*Figure 29.* Depth analysis for car-eval-f.

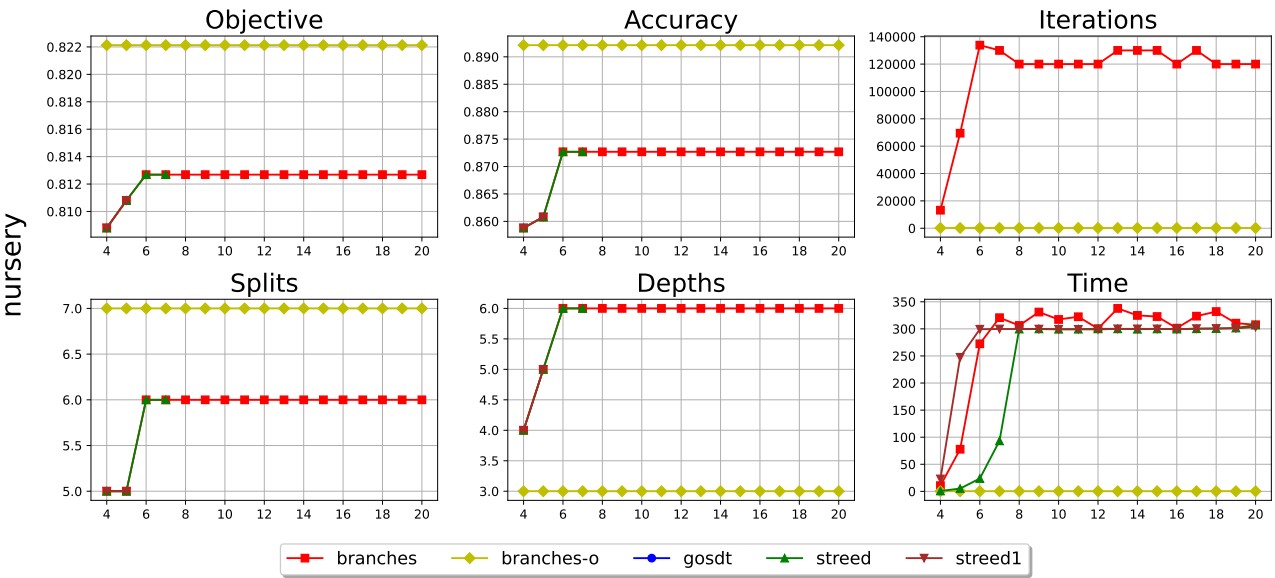

*Figure 30.* Depth analysis for nursery.

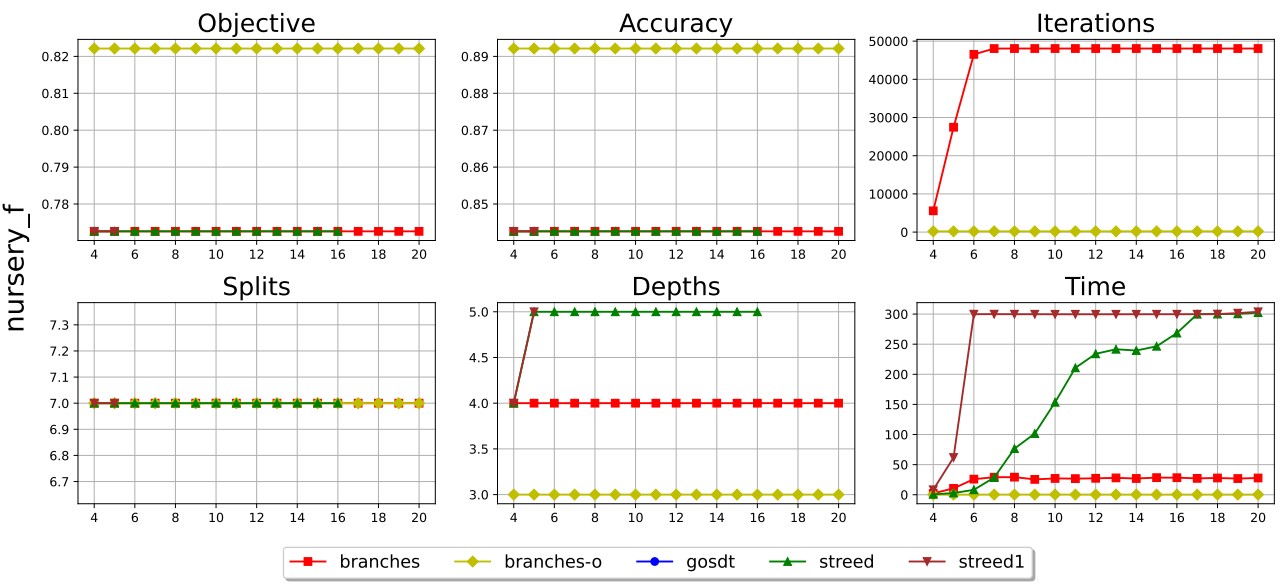

*Figure 31.* Depth analysis for nursery-f.

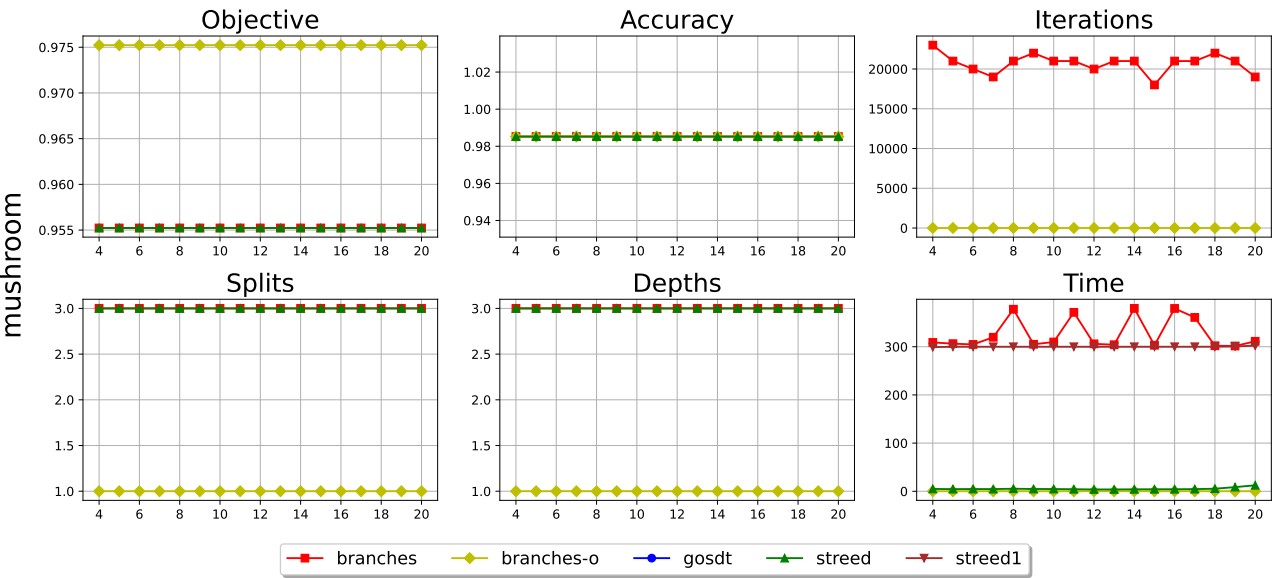

*Figure 32.* Depth analysis for mushroom.

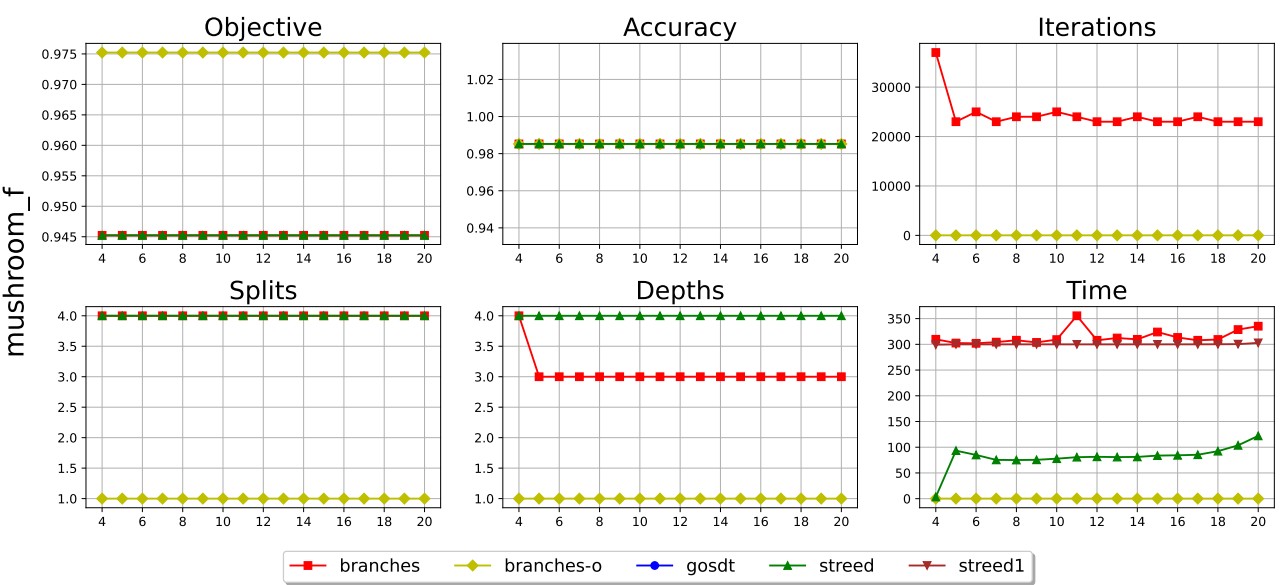

*Figure 33.* Depth analysis for mushroom-f.

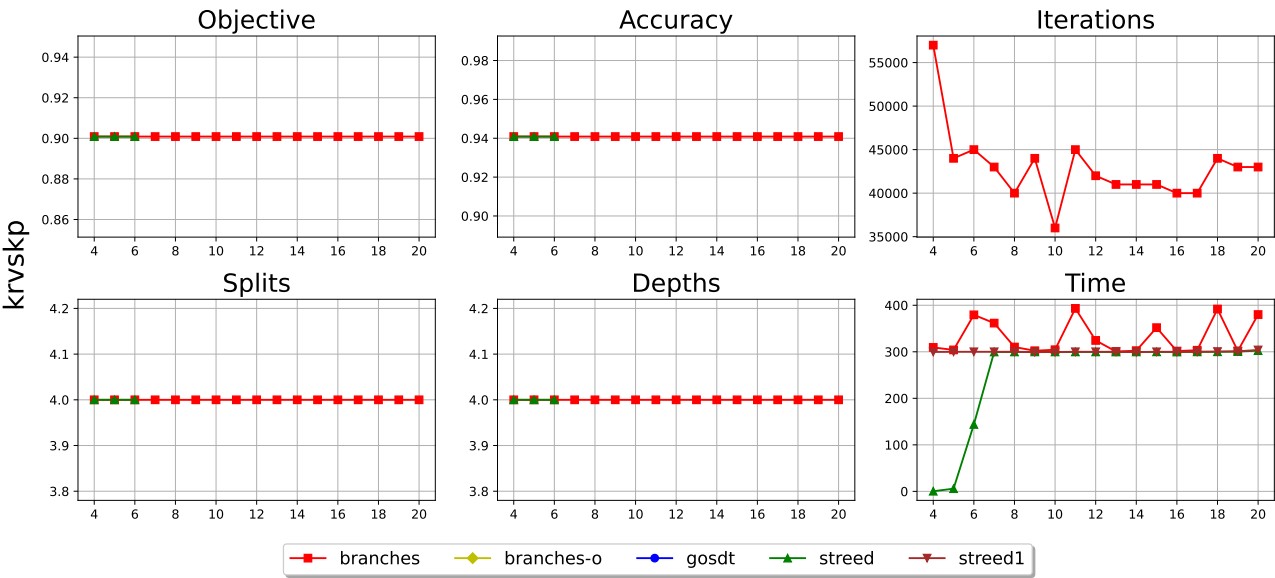

*Figure 34.* Depth analysis for kr-vs-kp.

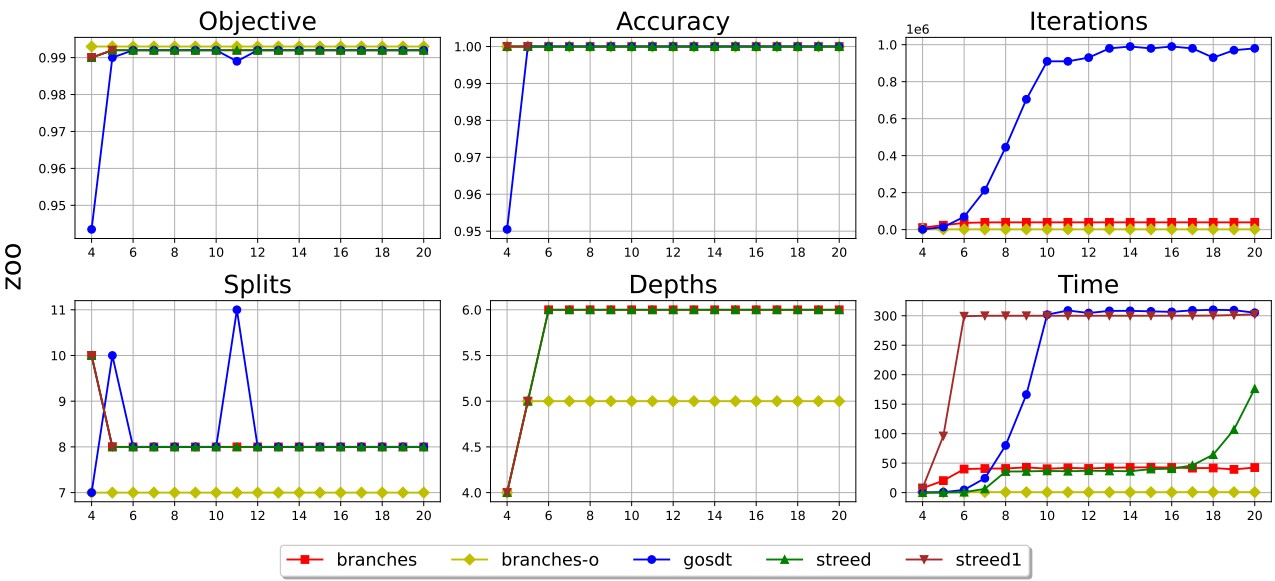

*Figure 35.* Depth analysis for zoo.

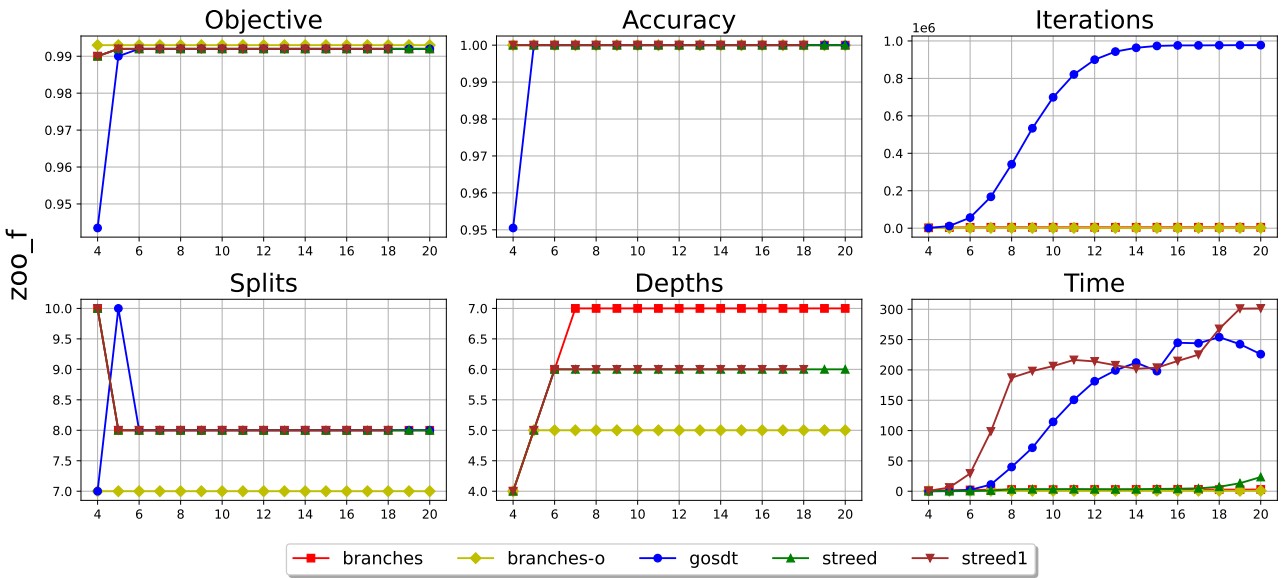

*Figure 36.* Depth analysis for zoo-f.

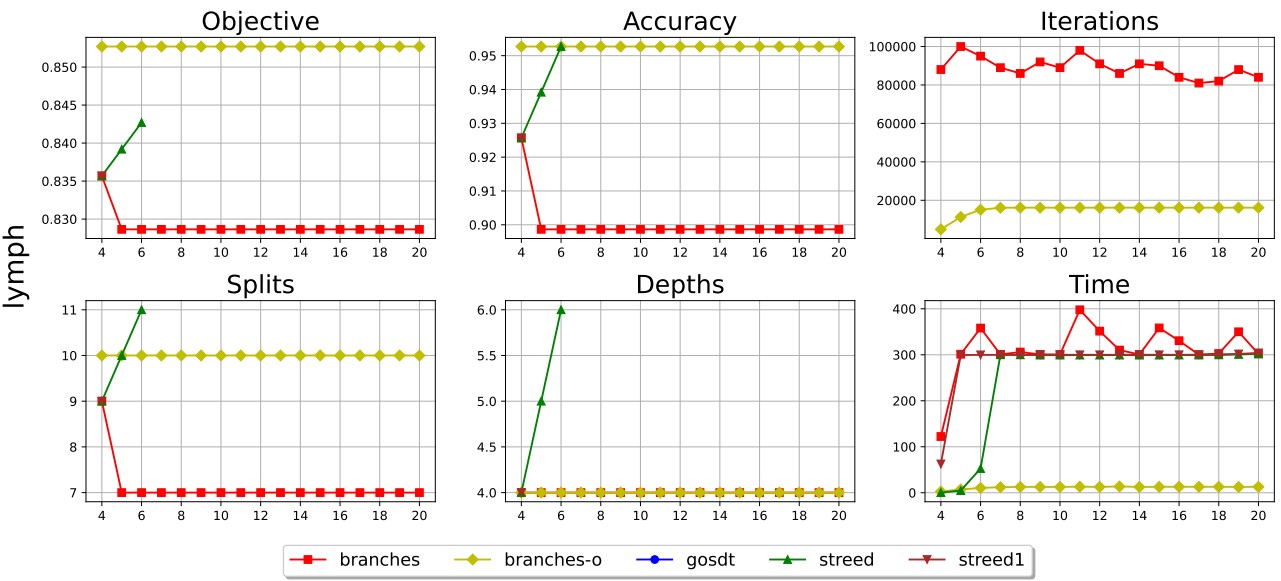

*Figure 37.* Depth analysis for lymph.

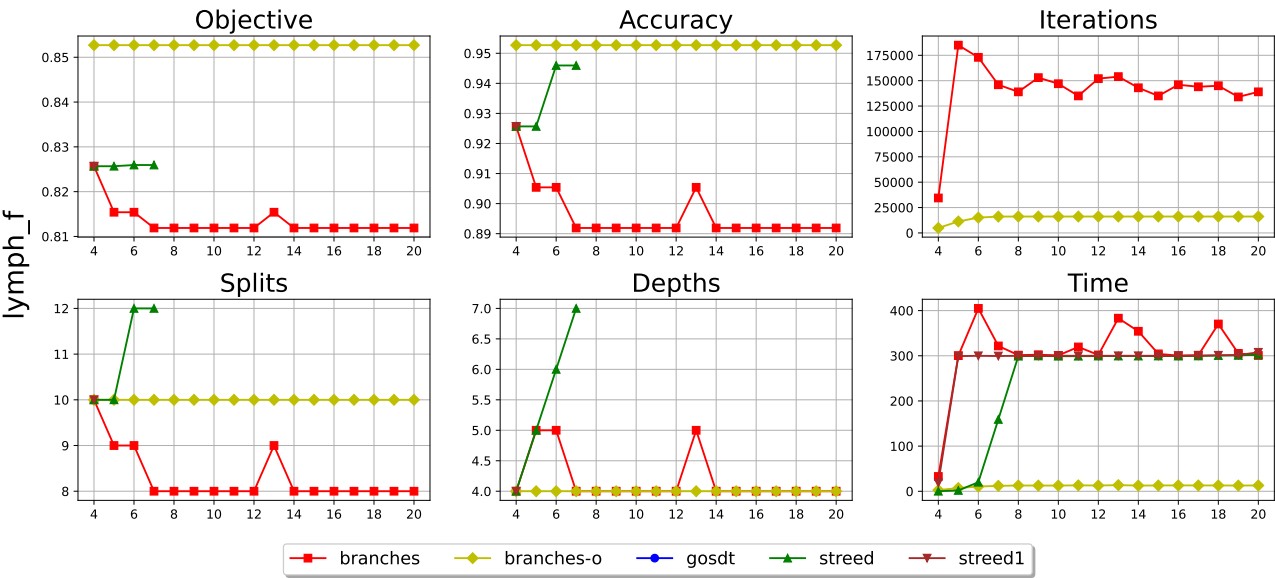

*Figure 38.* Depth analysis for lymph-f.

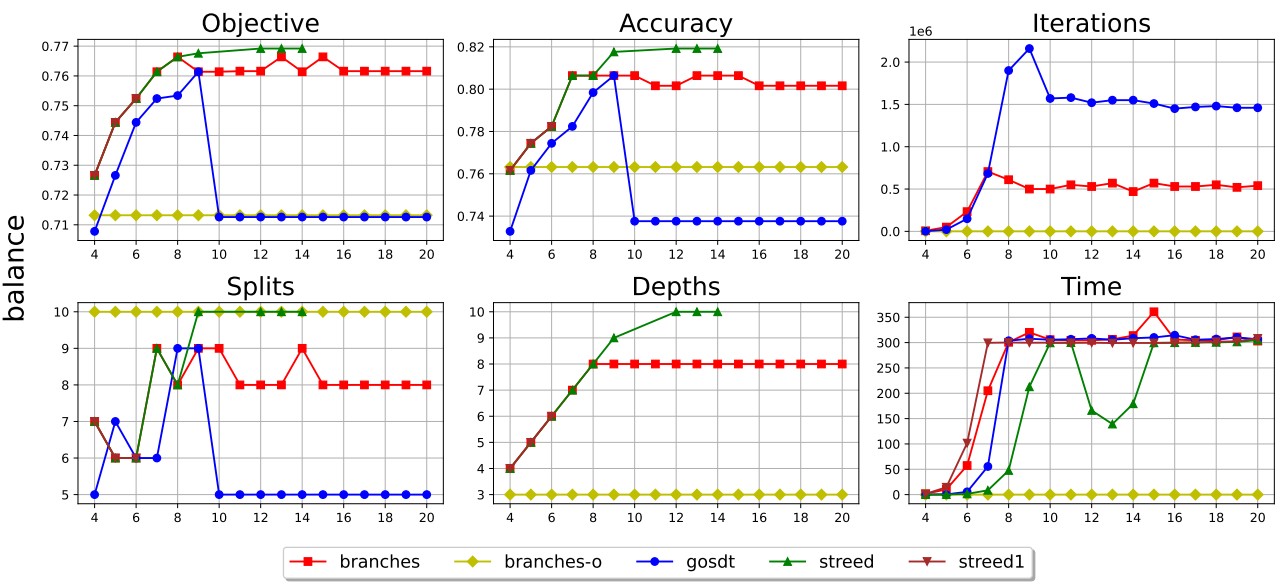

*Figure 39.* Depth analysis for balance.

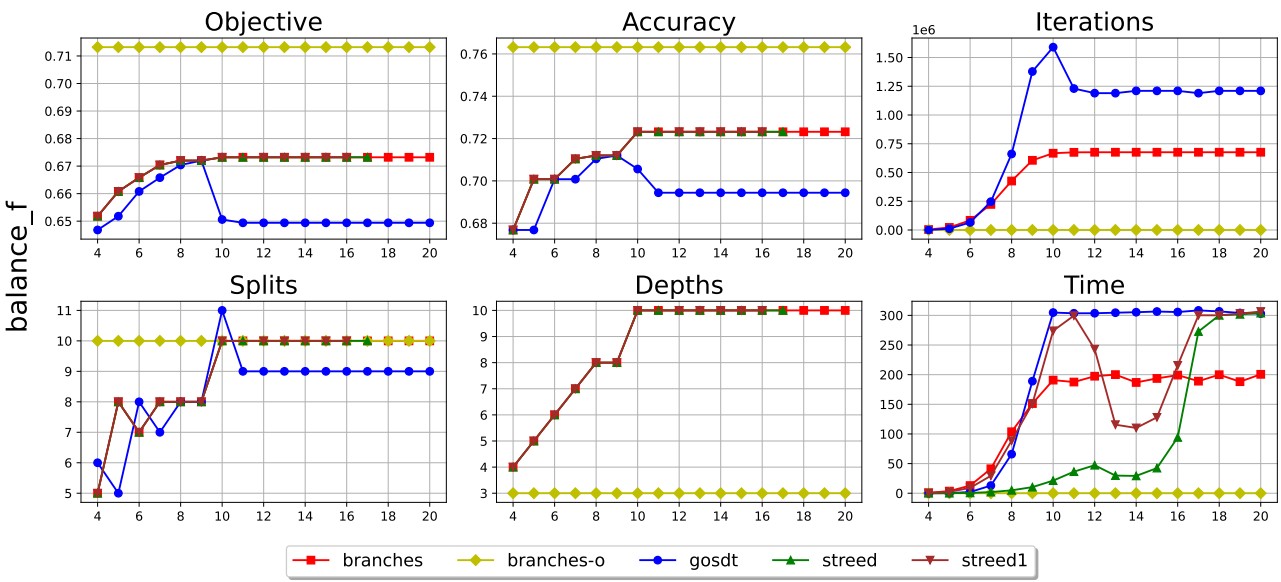

*Figure 40.* Depth analysis for balance-f.

