# OpenReview forum: "Branches: Efficiently Seeking Optimal Sparse Decision Trees via AO*"
_ICML.cc/2025/Conference — ICML 2025 poster_

### Official Review · Reviewer_qYmf · 2025-03-10

**Overall Recommendation:** 3

**Summary:**

The paper introduces BRANCHES, a search method for decision trees. Using an AND/OT graph formulation for the decision tree search, the method relies an OA*-like exploration strategy with purification bounds for the heuristic.
The introduced method provably recovers the optimal decision tree and the efficiency compares favorably with respect to the efficiency bound from OSTD in the literature. The method shows significantly improved empirical performance and efficiency an a wide range of UCI classification task.

**Claims And Evidence:**

I did not find any problematic claims.

**Essential References Not Discussed:**

The related work section focuses on DFS, BFS, and AO* approaches. I believe there is also a line of work applying MCTS to find optimal decision trees:

* Online Learning of Decision Trees with Thompson Sampling; Chaouki et al. (2024) [This reference is already included, but it is not discussed how it relates to the proposed method]

* Learning decision trees through Monte Carlo tree search: An empirical evaluation; Nunes et al. (2020).

**Experimental Designs Or Analyses:**

I checked the experiment in Section 5 and did not find an issue with it.

**Methods And Evaluation Criteria:**

* Analyzing the method in terms of optimality and efficiency makes sense. I also appreciated the numerical comparison of the efficiency bound to the one from OSDT in order to give some perspective on how they compare to each other.
* The method is empirically evaluated on standard classification tasks from the UCI repository (11 in total) and compared to three baselines (including a relatively recent one from 2024). Using accuracy/number of splits/time (and number of iterations, where applicable) as evaluation criteria also makes sense.

**Other Comments Or Suggestions:**

I think it could help to add a remark for which nodes are the "AND" nodes and which nodes are the "OR" nodes.

The notation is a bit unusual (not wrong, though) in some cases:
* Section 3, first paragraph: I think the following notation is a bit confusing: $D=\ {X_m, Y_m\}_{m=1}^n$. Letting the index run from $m=1,...,M$ or from $n=1,...,N$ might be a bit easier to parse.
* Denoting the transitions in a MDP with $F$ instead of $T$ is also a bit unusual.

**Other Strengths And Weaknesses:**

I appreciated the relatively detailed problem formulation which was helpful even for readers without extensive background on decision trees.

**Questions For Authors:**

-

**Relation To Broader Scientific Literature:**

* the optimization objective (including the penalty term) is the same as Bertsimas & Dunn (2017),  Chaouki et al. (2024), Hu et al. (2019) and in case of binary classification, Lin et al. (2020).
* the work is most closely related to Lin et al. (2020), but a detailed list of differences is given in the Appendix (namely: additional support for ordinal encoding, additional complexity analysis, improved empirical performance, value estimate update only along selection path, multiple local priority queues instead of a global one).

**Theoretical Claims:**

I did not check the proofs in Appendix G.

---

> ### Author Rebuttal · Authors · 2025-03-31
>
> Thank you for very much for your reviews, please find our response below:
>
> ## Missing literature
> - Chaouki et al. (2024)'s TSDT algorithm is tailored to online classification where a data stream is observed instead of a batch of data, which is different from the batch setting we consider. Due to the online consideration, TSDT does not have a natural termination condition like Branches and GOSDT, it has to keep adjusting the prior distributions of the node values until reaching a prespecified number of iterations. For these two reasons we decided not to compare with TSDT. However, we thank you for this suggestion, and we can definitely add such a comparison in the appendix for illustration purposes.
> - Nunes et al. (2020) do not solve the sparsity problem (minimising $\mathcal{H}_\lambda$).  In addition, we are not aware of a public code for the authors' algorithm, which prevents us from directly comparing with their algorithm. Nevertheless, (Table 4, Nunes et al. 2020) seems to indicate that the algorithm is significantly slower than the state of the art as it runs in hours. For example, car-
> evaluation takes 3 hours and 19 minutes and tic-tac-toe takes 4 hours and 59 minutes. Branches and
> GOSDT terminate and are optimal in under 2 minutes in both of these experiments.
>
> ## AND/OR representation
> Thank you for this remark. In fact, in section 3.3, we state that we follow the hypergraph convention in (Nilsson, 2014, Section 3.1), which replaces the notions of AND/OR nodes with nodes and connectors. The reason we chose this convention over the AND/OR nodes is to make the graph illustration more compact. With AND/OR nodes, Figure 1 would not be able to fit in the paper. We note that both conventions are equivalent. With AND/OR nodes, OR nodes would represent states (branches) and AND nodes would represent actions (split actions and the terminal action). Furthermore, the search space of branches is represented in several papers with the hypergraph convention (albeit without specifying a link to AND/OR search), e.g:
>
> - "Aglin, G., Nijssen, S., and Schaus, P. (2020). Learning optimal decision trees using caching branch-and-bound search. In Proceedings of the AAAI conference on artificial intelligence, volume 34, pages 3146–3153." Figure 2.
> - "Nijssen, S. and Fromont, E. (2007). Mining optimal decision trees from itemset lattices. In Proceedings of the 13th ACM SIGKDD international conference on Knowledge discovery and data mining, pages 530–539." Figure 2.
> - "Nijssen, S. and Fromont, E. (2010). Optimal constraint-based decision tree induction from
> itemset lattices. Data Mining and Knowledge Discovery, 21:9–51." Figure 1.
>
> We thank you for this suggestion, we can try to include an equivalent representation of Figure 1 in terms of AND/OR nodes in the Appendix.
>
> ## Notation
> Thank you for this recommendation. In fact, we chose $F$ for transitions instead of $T$ to avoid confusion because we already use $T$ substantially with sub-DTs and DTs. The reason we chose $F$ specifically is because it is used in this context in some literature, e.g.:
> - "Learning Depth-First Search: A Unified Approach to Heuristic Search in Deterministic and Non-Deterministic Settings, and its application to MDPs" in section **Models**.

---

> > ### Comment · Reviewer_qYmf · 2025-04-04
> >
> > Thank you for your reply. My questions were addressed. I don’t object acceptance.

---

### Official Review · Reviewer_EG8X · 2025-03-16

**Overall Recommendation:** 3

**Summary:**

The paper presents Branches, a new approach for computing optimal decision trees by formulating the problem as AND/OR graph search and proposing an AO*-type algorithm that solves the problem. The proposed approach learns non-binary trees (i.e., trees with multiway splits) for non-binary features. The author provide a theoretical characterization of the algorithm complexity and experimental evaluation against three popular optimal decision tree baselines.

## after rebuttal
I thank the authors for their response. I think the paper would benefit from a more detailed discussion on interpretability (as noted in the author response, the number of branches in multiway splits can definitely impact interpretability). I also think a more detailed comparison of testing accuracy including approaches focused on continuous features that are very common is important (very granular discretization will lead to many categories which may hurt performance or interpretability). Overall, I maintain my evaluation.

**Claims And Evidence:**

- See specific concerns below regarding the experimental evaluation and analysis.
- In addition, the claim of interpretability requires further justification: in particular, different from previous work this work proposes a non-binary tree, i.e., trees with multiway splits. It is not clear whether they should be perceived as equally interpretable to people and there is no clear discussion on this.

**Essential References Not Discussed:**

N/A

**Experimental Designs Or Analyses:**

See relevant points under "methods and evaluation" above.

**Methods And Evaluation Criteria:**

There are several concerns regarding the experimental evaluation setting:
- The reporting of the results seem to focus on *training* accuracy rather than *test* accuracy (it is not clearly stated what accuracy is reported, but likely training accuracy as there is no mention of splitting the dataset to train and test set). Given that some algorithms time out (TO) and some reach different optimal solutions (as indicated by different objectives while both algorithms have run to completion, perhaps due to the setting of max depth), it is particularly useful to compare test accuracy in addition to training accuracy.
- No comparison with simple baselines like CART.
- No comparison with anytime approaches for optimal trees like Blossom [1].
- Benchmark dataset: many of the benchmarks on the list (monks and tic-tac-toe which together account for half of the results table) are synthetic.
- Most datasets have low-dimensionality. The only one whose dimensionality is above 100 is mushroom where it seems Murtee and STreeD are significantly faster.
- No analysis for dataset with continuous variables and in particular comparison with approaches designed for such datasets like Quant-BnB [2].
- Its not clear why running times and splits that are the lowest are sometimes not in bold font (if the bold font is based purely on objective, then the only thing that should be bold is the objective)

[1] Demirović, Emir, Emmanuel Hebrard, and Louis Jean. "Blossom: an anytime algorithm for computing optimal decision trees." International Conference on Machine Learning. PMLR, 2023.

[2] Mazumder, Rahul, Xiang Meng, and Haoyue Wang. "Quant-BnB: A scalable branch-and-bound method for optimal decision trees with continuous features." International Conference on Machine Learning. PMLR, 2022.

**Other Comments Or Suggestions:**

N/A

**Other Strengths And Weaknesses:**

Strengths:
- Novel approach for optimal decision trees based on AND/OR graphs and AO*-style algorithm.
- Experiments show significant gains in performance compared to the baselines.
- Theoretical characterization of complexity is provided

Other weaknesses:
- No discussion of tie-breaking in search (e.g., if multiple actions have similar value in Eq. 15) and its impact on performance (also relevant for steps like "choose one of them arbitrarily", p.6).
- It would be useful to provide a brief description of the base AO* algorithm.

**Questions For Authors:**

See above for my concerns.

**Relation To Broader Scientific Literature:**

Overall the paper includes a reasonable review of previous work, with some notable works missing (examples provided above). It provides a new approach for optimal decision trees which is an active research area with significant interest in recent years.

**Theoretical Claims:**

I did not carefully check the correctness of the proofs in the supplementary material.

---

> ### Author Rebuttal · Authors · 2025-03-31
>
> Thank you very much for your review, please find below our response:
>
> ## Interpretability of multi-way splits
> - Thank you for raising this point. The interpretability of DTs is due to their simple decision rules, it is not specific to binary DTs. On the other hand, we recognise that DTs where each node has a large number of children can become less interpretable than binary DTs. We alleviate this issue by penalising the number of splits, which leads to DTs with small number of splits and hence more interpretablity.
> - In Appendix E, we directly compare ordinal encoding with binary encoding in terms of interpretability, especially when we introduce the notion of collapse. We argue in these examples that DTs that stem from ordinal encoding can be more interpretable than binary encoded DTs. This is especially the case when comparing Figure 7 and Figure 8; and also Figure 12 and Figures 9 and 10.
> - In addition, Branches is not restricted to ordinal encoding, it can be applied to binary encoding in similar fashion to the state of the art. In which case, if we have a good binary encoding that we think yields interpretable DTs, we can choose to use Branches with it. The benefits of Branches compared to the other algorithms are still satisfied in this case.
>
> ## Methods And Evaluation Criteria:
> - Yes, Table 2 reports the training accuracy with no train/test split. The objective being to find the most accurate DT with least complexity (most sparse DT). Quoting (Lin et al.2020; Section 5)[2]: "Learning theory provides guarantees that training and test accuracy are close for sparse trees". However, we recognise that a direct comparison of test performance would be insightful as well. In Appendix H.2, we compare Branches with CART and DL8.5 in terms of Pareto fronts (accuracy vs number of splits) within a 10-fold cross-validation. The reason we restrict this comparison to Branches, CART and DL8.5 is to compare different types of DT construction algorithms (Branches seeks optimal sparse DTs, CART seeks DTs greedily, and DL8.5 seeks optimal DTs subject to a hard constraint on depth), comparing Branches with GOSDT and STreeD in this context would yield the same solutions when they terminate.
> - We compared with CART in Appendix H.2.
> - Blossom does not solve the problem of sparsity that Branches, GOSDT and STreeD solve. It rather seeks optimal DTs subject to hard constraint on depth in similar fashion to DL8.5. Nevertheless, we included an illustrative comparison with these types of DTs in Appendix H.2, where we chose the popular DL8.5 algorithm. We note that DL8.5 is anytime as well.
> - Indeed Branches struggles with highly dimensional data such as mushroom, but we showed that its handling of ordinal encoding alleviates this issue significantly, with a fast termination in only $0.15s$ for mushroom-o. Moreover, in Appendix H.4, we investigated the reason STreeD performs exceptionally well on mushroom, we found that it is mainly due to the depth 2 solver, a technique introduced in Demirovic et al.2022 [1]. This leads us to believe that a future incorporation of the depth 2 solver in Branches could further improve its current performance.
> - Branches handles categorical features. Any type of discretisation preprocessing can be applied to numerical features before feeding the dataset to Branches. This is similar to other algorithms such as GOSDT, STreeD and MurTree, with the additional benefit that Branches handles multiway splits and thus does not necessitate binary preprocessing. We did not compare with Quant-BnB because it is specifically tailored to continuous features, and it does not solve the sparsity problem. Rather, Quant-BnB seeks optimal DTs subject to a hard constraint on depth that is either 2 or 3. Branches and the algorithms we compare with can find optimal sparse DTs of higher depths.
> - We make text bold based on the objective $\mathcal{H}_\lambda$ first. Then we compare accuracy, splits and runtimes of the methods yielding the highest objective, with the corresponding bold text.
>
> ## Other weaknesses:
> - Thank you very much, this is a very interesting point. In appendix D.1, page 14, we provide a tie-break strategy "There is an additional benefit to storing -value_complete. When there are multiple split actions attribute that maximise value, then we prioritise the one maximising value_complete.". We will update the main paper to refer to this tie-break strategy. Thank you for this recommendation.
> - For space concerns, we refer the reader to (Nilsson, 2014, Section 3.2) for the base AO*.
>
> [1] Demirovic, E., Lukina, A., Hebrard, E., Chan, J., Bailey, J., Leckie, C., Ramamohanarao, K., and Stuckey, P. J. Murtree: Optimal decision trees via dynamic programming and search. Journal of Machine Learning Research, 23(26):1–47, 2022.
>
> [2] Lin, J., Zhong, C., Hu, D., Rudin, C., and Seltzer, M. Generalized and scalable optimal sparse decision trees. In International Conference on Machine Learning, pp. 6150–6160. PMLR, 2020.

---

### Official Review · Reviewer_TyUY · 2025-03-22

**Overall Recommendation:** 3

**Summary:**

The paper considers the problem of learning an optimal decision tree for a given dataset. Specifically, the DT learning problem is formulated as a heuristic search problem over an AND/OR graph representing the space of all possible DTs. Consequently, an efficient best-first search algorithm (aka AO* search) called Branches is developed to find an optimal DT in terms of splits. An empirical evaluation is carried out on standard machine learning datasets from the UCI repository. The results demonstrate that the proposed AO* search algorithm outperforms existing state-of-the-art approaches based on depth-first Branch-and-Bound search.

**Claims And Evidence:**

The claims are supported by experimental results.

**Essential References Not Discussed:**

The related work seems to be addressed well.

**Experimental Designs Or Analyses:**

The experimental evaluation is sound.

**Methods And Evaluation Criteria:**

The evaluation makes sense

**Other Comments Or Suggestions:**

See previous section.

**Other Strengths And Weaknesses:**

Main strengths:
-------------------

The paper considers an important yet quite challenging problem in machine learning and AI, namely learning optimal decision trees for given datasets. Despite its difficulty, the problem has many real-world applications and therefore more efficient algorithms for solving it are warranted. DTs are interpretable and therefore well suited for situations where the model's decisions must be explained such is healthcare applications.

The empirical evaluation is sound and is conducted in a principled manner. The results are presented in a relatively clear manner and therefore it is fairly easy to get the pig picture and appreciate the good performance of the proposed method.

Main weakness:
------------------

I found the quality of the presentation quite poor. The presentation of the method is quite dense and it is not easy to follow the details. I think sections 3.1 and 3.2 need a good running example that would illustrate the technical details described such as the branches and sub-DTs.

Section 3.2 and especially Section 4 are not easy to follow because the presentation mixes concepts common to the RL literature such as policies, actions and value functions, with concepts common to the heuristic search community such as search nodes, node expansion, node value update etc.

Since the main contribution of the paper is a heuristic best-first search I suggest adopting a description closer to the search community. Specifically, I think it's important to describe clearly the search space in terms of OR nodes and AND nodes, as well as the values associated with the nodes, what these values represent and the way they are computed during search. As far as I can see, an OR nodes maximises the values of its children, while an AND node combines the values of its children by summation. The connector representation from Fig 1 is not very common and therefore is not easy to digest. Instead, I would represent the OR and the AND nodes explicitly. Also it is important to articulate clearly what the solution graph represents and perhaps illustrate it with an example.

The observation that AO*-like algorithms are more efficient than depth-first branch and bound algorithms (of course at the expense of using additional memory) is well known in the heuristic search community. Therefore, the experimental results are not very surprising.

**Questions For Authors:**

1. Regarding the heuristics used, is the proposed purification bound heuristic admissible?

**Relation To Broader Scientific Literature:**

The related work seems to be addressed well.

**Theoretical Claims:**

The theoretical claims appear to be sound.

---

> ### Author Rebuttal · Authors · 2025-03-31
>
> Thank you very much for your reviews, please find our response below:
>
> ## Presentation:
> Thank you for this feedback. We have included Figure 2 in the Appendix for the purpose of illustrating the notions of branches and sub-DTs. Do you have any recommendation that would improve this figure's quality? Thank you.
>
> ## Mixed concepts from the RL and heuristic search community
> RL and heuristic search share many common notions and terminology and we do not think that notions like policies, actions and values are exclusive to RL. In fact, several heuristic search papers employ these concepts such as:
>
> - "LAO*: A heuristic search algorithm that finds solutions with loops".
> - "Learning Depth-First Search: A Unified Approach to Heuristic Search in Deterministic and Non-Deterministic Settings, and its application to MDPs".
>
> The RL community also employs many concepts that are traditionally linked to heuristic search. The notions of node expansion, update... are extensively employed in the context of Monte Carlo Tree Search, e.g "Browne, C. B., Powley, E., Whitehouse, D., Lucas, S. M., Cowling, P. I., Rohlfshagen, P., Tavener, S., Perez, D., Samothrakis, S., and Colton, S. (2012). A survey of Monte Carlo Tree Search methods. IEEE Transactions on Computational Intelligence and AI in games, 4(1):1–43".
>
> These unified concepts are very helpful for us in deriving the proofs of our theoretical results. If we failed to define clearly some of these notions, we kindly ask if you could refer us to the ones in question so that we incorporate the necessary modifications. Thank you very much.
>
> ## AND/OR Search Graph
> In section 3.3, we state that we follow the hypergraph convention in (Nilsson, 2014, Section 3.1). The reason we chose this convention over the AND/OR nodes is to make the graph illustration more compact. With AND/OR nodes, Figure 1 would not be able to fit in the paper. We note that both conventions are equivalent, with AND/OR nodes, OR nodes would represent states (branches) and AND nodes would represent actions (split actions and the terminal action). Furthermore, the search space of branches is represented in several papers with the hypergraph convention (albeit without specifying a link to AND/OR search), e.g:
>
> - "Aglin, G., Nijssen, S., and Schaus, P. (2020). Learning optimal decision trees using caching branch-and-bound search. In Proceedings of the AAAI conference on artificial intelligence, volume 34, pages 3146–3153." Figure 2.
> - "Nijssen, S. and Fromont, E. (2007). Mining optimal decision trees from itemset lattices. In Proceedings of the 13th ACM SIGKDD international conference on Knowledge discovery and data mining, pages 530–539." Figure 2.
> - "Nijssen, S. and Fromont, E. (2010). Optimal constraint-based decision tree induction from
> itemset lattices. Data Mining and Knowledge Discovery, 21:9–51." Figure 1.
>
> Thank you for the recommendation regarding a solution graph. A solution graph is the DT of a policy, for example, the graph constituted of the red connectors in figure 1 is a solution graph. We can add this remark to the paper, thank you.
>
> ## Observation regarding AO* vs DFS:
> Indeed, but to our knowledge, this advantage has not been explored for seeking optimal sparse decision trees. The contribution of our paper was to explore this and develop Branches.
>
> ## Questions
> 1- Yes, Proposition 4.2 proves that the purification bound is admissible as it always overestimates the optimal value of a node (overestimation as opposed to the traditional underestimation because we are in a formulation based on objective maximisation instead of cost minimisation). Moreover, Theorem 4.3 proves the optimality of Branches upon termination.

---

> > ### Comment · Reviewer_TyUY · 2025-04-02
> >
> > Thanks for your response. It definitely clarified my concerns.

---

### Official Review · Reviewer_EY1f · 2025-03-24

**Overall Recommendation:** 2

**Summary:**

This paper presents "BRANCHES," a novel algorithm for learning optimal decision trees (DTs) by integrating Dynamic Programming (DP) and Branch & Bound (B&B) techniques. The study addresses the trade-offs in existing methods—where some approaches provide efficient DP strategies but lack strong pruning bounds, and others excel in pruning but sacrifice DP efficiency. The authors introduce a new analytical pruning bound, called the "Purification Bound," to enhance search space reduction. Empirical evaluations demonstrate that BRANCHES surpasses state-of-the-art methods in speed, optimality, and iteration efficiency while supporting non-binary features. Theoretical analysis confirms its computational superiority.

**Claims And Evidence:**

See Strengths And Weaknesses.

**Essential References Not Discussed:**

Many peer works that can handle with non-binary features and large datsets are missing:
[1] McTavish, H., Zhong, C., Achermann, R., Karimalis, I., Chen, J., Rudin, C., & Seltzer, M. (2022, June). Fast sparse decision tree optimization via reference ensembles. In Proceedings of the AAAI conference on artificial intelligence (Vol. 36, No. 9, pp. 9604-9613).
[2] Mazumder, R., Meng, X., & Wang, H. (2022, June). Quant-BnB: A scalable branch-and-bound method for optimal decision trees with continuous features. In International Conference on Machine Learning (pp. 15255-15277). PMLR.
[3] Hua, K., Ren, J., & Cao, Y. (2022). A scalable deterministic global optimization algorithm for training optimal decision tree. Advances in Neural Information Processing Systems, 35, 8347-8359.

Author should cite and compare with the newest literature to validate the claimed benefits in their work.

**Experimental Designs Or Analyses:**

See Strengths And Weaknesses, and Essential References Not Discussed.

**Methods And Evaluation Criteria:**

See Strengths And Weaknesses.

**Other Comments Or Suggestions:**

While BRANCHES performs well, its complexity may still pose challenges for extremely high-dimensional datasets. The study does not compare BRANCHES to ensemble methods such as Random Forests, which could provide additional insights into practical performance trade-offs. The reliance on Python may hinder adoption in environments where execution speed is critical.

Overall, this work presents a significant contribution to interpretable machine learning by improving decision tree optimization. However, addressing its limitations—particularly scalability, native numerical feature support, and parallelization—would further strengthen its impact. Minor clarifications on the empirical benchmarking methodology and additional comparisons with alternative optimization techniques would be beneficial.

**Other Strengths And Weaknesses:**

Strengths
The paper successfully integrates the strengths of both methodologies, providing an effective pruning strategy while maintaining computational efficiency. This new analytical bounding strategy significantly improves search space reduction, leading to faster convergence. The authors offers rigorous theoretical proofs and extensive empirical evaluations demonstrating the superiority of BRANCHES over existing methods.

Weaknesses
1. The method currently only supports categorical data, necessitating pre-processing for numerical datasets.
2. The Python-based implementation, while effective, lags behind C++-based state-of-the-art implementations in execution speed.
3. Although BRANCHES is inherently parallelizable, the current implementation does not exploit multithreading capabilities.

**Questions For Authors:**

1. How does BRANCHES handle large-scale datasets compared to existing methods in terms of memory usage and computational time?
2. Would extending the method to numerical features via native handling (rather than discretization) significantly impact its efficiency?
3. How does BRANCHES perform when tested on real-world applications beyond benchmark datasets?

**Relation To Broader Scientific Literature:**

Related to a new methods for optimal decision tree.

**Theoretical Claims:**

See Strengths And Weaknesses.

---

> ### Author Rebuttal · Authors · 2025-03-31
>
> Thank you very much for your reviews, please find below our response.
> ## Missing Literature:
>
> - (McTavish et al. 2022): We cite these authors, in fact we use their implementation of GOSDT as mentioned in Appendix H. We also directly quote them in Appendix H.4. However, we only compared with GOSDT and not with the additional guided guesses that the authors introduced. The reason is that this loses the optimality guarantee (with respect to $\mathcal{H}_\lambda$) of GOSDT upon termination, and our main objective in this paper is to compare methods satisfying this optimality guarantee, hence the choice of STreeD, MurTree and GOSDT. Furthermore, the guesses strategy is not exclusive to GOSDT, it can be incorporated within Branches search strategy as well in similar fashion to how the authors incorporated it in GOSDT and DL8.5. For this reason, it makes sense to compare the base algorithms and to consider an incorporation of the guesses strategy in the future for Branches.
> - (Mazumder et al.2022) is specifically tailored to continuous features. Moreover, the main objective of this work, which is (1), does not incorporate sparsity concerns in terms of actively seeking to minimise the complexity of the solution while simultaneously maximising its accuracy. Quant-BnB rather optimises its objective subject to a hard constraint on depth as either $2$ or $3$. This is different from our work, which aligns more with literature that jointly optimise accuracy and sparsity via the objective $\mathcal{H}_\lambda$.
> - (Hua et al. 2022) is also specifically tailored to continuous features and large scale applications as demonstrated by the costly experiments of 1000 cores and two hours run-times. Our experiments on the other hand are less costly, we run them on a personal Machine (2,6 GHz 6-Core Intel Core i7) as stated in Appendix H and for 5 minutes, which makes them easily reproducible.
>
> ## Weaknesses:
> 1- Indeed, Branches necessitates a discretisation of numerical features, this is common among many works in the literature of DT optimisation. Moreover, contrary to the state of the art, Branches is not restricted to binary encodings but can deal with ordinal encodings as well.
>
> 2, 3 - Indeed, we have mentioned these limitations in Section 6 as avenues for future work.
>
> ## Comparison with Random Forest:
> Ensemble methods such as Random Forest focus on performance in terms of accuracy (for example) but forgo interpretability. The objective of this paper is to jointly optimise for both concerns in similar fashion to papers from this literature such as:
> -  Demirovic, E.,Lukina, A., Hebrard, E., Chan, J., Bailey,J., Leckie, C., Ramamohanarao, K., and Stuckey, P. J.Murtree: Optimal decision trees via dynamic programming and search. Journal of Machine Learning Research, 23(26):1–47, 2022.
> - Hu, X., Rudin, C., and Seltzer, M. Optimal sparse decision trees. Advances in Neural Information Processing Systems, 32, 2019.
> - Lin, J., Zhong, C., Hu, D., Rudin, C., and Seltzer, M. Generalized and scalable optimal sparse decision trees. In International Conference on Machine Learning, pp. 6150–6160. PMLR, 2020.
>
> For this reason, works from this literature generally do not compare with ensemble methods.
>
> ## Questions:
> 1- Being a BFS (Best First Search) method, Branches is more memory consuming than DFS methods such as MurTree and STreeD. On the other hand, its informed BFS strategy allows it to achieve optimality faster thus alleviating the issue. This is a general trade-off between BFS and DFS, and it is thus valuable to devise an BFS strategy that terminates optimally quickly before consuming too much memory. This is what we achieved with Branches, Theorem 4.4 and Table 1 analyse its time complexity (which is directly linked to its memory complexity as the search graph grows with each iteration) and show its superiority compared to the literature. Furthermore, Table 2 shows that the number of iterations to termination of Branches is significantly smaller than GOSDT (a BFS method as well), which allows it to terminate in smaller runtimes even though Branches is currently implemented in Python and GOSDT in C++ . The memory concerns of BFS and DFS are discussed in Section 2.
>
> 2- Perhaps, but we have to keep in mind the optimality concerns with respect to sparsity as well. It is unclear whether a native handling of numerical features would keep this guarantee.
>
> 3- As shown with the mushroom dataset, high-dimensional data can be hard to deal with as the degree of each node in the search graph becomes too large. On the other hand, Branches seems to work well on large datasets in terms of rows.

---

### Decision · Program_Chairs · 2025-05-01

**Decision:**

Accept (poster)

**Comment:**

The work deals with the theoretically hard task to find good decision tree models. The devised algorithm is anytime and guaranteed to return an optimal solution when it terminates. The work provides some complexity guarantees and empirical study about it. The problem is formulated as an AND/OR graph search which is attacked via an AO*-type algorithm. The resulting approach is not fully compared against other ideas in the literature (though the authors try to answer to this point in the rebuttal, it is apparent from the discussions that both theoretical and empirical comparisons could have been better conducted), and currently has some limitations in the type of data it can handle (or requires some sort of discretisation, even if that is common practice in optimisation for this purpose). The presentation is thought to be jammed and could be significantly improved, and the description could be more in line with previous work on search. All in all, this is a borderline case.